# Biosensor-Enhanced Organ-on-a-Chip Models for Investigating Glioblastoma Tumor Microenvironment Dynamics

**DOI:** 10.3390/s24092865

**Published:** 2024-04-30

**Authors:** Gayathree Thenuwara, Bilal Javed, Baljit Singh, Furong Tian

**Affiliations:** 1School of Food Science and Environmental Health, Technological University Dublin, Grangegorman Lower, D07 H6K8 Dublin, Ireland; gayathreethenuwara57@gmail.com (G.T.); bilal.javed@tudublin.ie (B.J.); 2Institute of Biochemistry, Molecular Biology, and Biotechnology, University of Colombo, Colombo 00300, Sri Lanka; 3Nanolab Research Centre, FOCAS Research Institute, Technological University Dublin, Camden Row, D08 CKP1 Dublin, Ireland; 4MiCRA Biodiagnostics Technology Gateway, Technological University Dublin (TU Dublin), D24 FKT9 Dublin, Ireland; baljit.singh@tudublin.ie

**Keywords:** glioblastoma, biosensor, microenvironment, organ-on-a-chip, real-time monitoring, cellular dynamics, precision medicine, diagnostic strategies, therapeutic approaches

## Abstract

Glioblastoma, an aggressive primary brain tumor, poses a significant challenge owing to its dynamic and intricate tumor microenvironment. This review investigates the innovative integration of biosensor-enhanced organ-on-a-chip (OOC) models as a novel strategy for an in-depth exploration of glioblastoma tumor microenvironment dynamics. In recent years, the transformative approach of incorporating biosensors into OOC platforms has enabled real-time monitoring and analysis of cellular behaviors within a controlled microenvironment. Conventional in vitro and in vivo models exhibit inherent limitations in accurately replicating the complex nature of glioblastoma progression. This review addresses the existing research gap by pioneering the integration of biosensor-enhanced OOC models, providing a comprehensive platform for investigating glioblastoma tumor microenvironment dynamics. The applications of this combined approach in studying glioblastoma dynamics are critically scrutinized, emphasizing its potential to bridge the gap between simplistic models and the intricate in vivo conditions. Furthermore, the article discusses the implications of biosensor-enhanced OOC models in elucidating the dynamic features of the tumor microenvironment, encompassing cell migration, proliferation, and interactions. By furnishing real-time insights, these models significantly contribute to unraveling the complex biology of glioblastoma, thereby influencing the development of more accurate diagnostic and therapeutic strategies.

## 1. Introduction

Glioblastoma multiforme (GBM) stands as the most prevalent, aggressive, and lethal primary brain tumor in adults, characterized by high vascularity and malignancy [1,2]. The recent World Health Organization (WHO) update in brain tumor classification in 2021 marks a pivotal shift by incorporating genotypic markers alongside established histological criteria. In the current classification of glioblastoma, the identification of a specific single nucleotide polymorphism in the isocitrate dehydrogenase (IDH) gene serves as a key determinant, distinguishing between wild-type and mutant variations. Historically, the diagnosis of glioblastomas relied on histological features, encompassing both IDH-mutated (10%) and IDH wild-type (90%) tumors, each manifesting markedly distinct biological characteristics and prognoses. However, under the WHO CNS5 classification, glioblastomas are now exclusively attributed to IDH wild-type tumors, marking a significant departure from the prior system [3].

Patient prognosis in GBM has remained persistently unfavorable for the past three decades, primarily due to the limitations of existing therapies such as surgical resection followed by concurrent radiation therapy and temozolomide (TMZ) [4]. The inadequacies in current therapeutic strategies are exacerbated by the high failure rates observed during clinical trials, largely attributed to imperfect models that hinder the accurate prediction of efficacy and toxicity in humans. This predicament is particularly pronounced in GBM, where no successful therapy has significantly improved survival since the introduction of temozolomide two decades ago [5,6,7]. GBM poses significant therapeutic challenges due to its inherent treatment resistance and propensity for recurrence. Despite multimodal therapies, GBM often relapses, underscoring the complexity of its tumor microenvironment (TME) [8]. This niche comprises a diverse array of cell types, including endothelial cells, neurons, astrocytes, oligodendrocytes, and resident immune cells such as microglia, as well as infiltrating immune cells like tumor-associated macrophages and tumor-infiltrating lymphocytes (TILs). Additionally, noncellular components such as signaling molecules, exosomes, extracellular matrix (ECM) constituents, and ECM remodeling enzymes contribute to the dynamic milieu of the TME. Such complexity is pivotal in shaping cancer cell survival and responses to therapy [9,10]. GBM cells play a central role in orchestrating TME dynamics through intricate signaling networks, modulating processes essential for tumor growth, invasion, and resistance to therapy [11,12]. These cells communicate with their surrounding microenvironment via various mechanisms, including soluble factors like cytokines, chemokines, and growth factors. Furthermore, GBM cells utilize exosomes, gap junctions, circulating tumor cells, tunneling nanotubes, cell-free DNA, and horizontal DNA transfer to interact with neighboring cells, both tumor and normal, thereby promoting tumor progression and therapy resistance [13,14,15].

The recognition of the pivotal role played by the tumor microenvironment (TME) in the development and progression of GBM has led to a renewed focus on therapeutic targeting. Novel treatment strategies aim to disrupt the intricate communication networks between tumor and normal cells within the TME [16]. However, to effectively develop and evaluate these therapies, reliable preclinical models that recapitulate the complexity of the GBM TME are essential [17]. Recent advancements in understanding the GBM TME have highlighted its highly heterogeneous and dynamic nature. Alterations in cellular composition, cell-to-cell interactions, and metabolic products, alongside other chemical factors such as pH and oxygen levels, profoundly influence TME dynamics. GBM cells possess the remarkable ability to reprogram their microenvironment, exploiting microenvironmental elements to promote rapid proliferation, invasion, migration, and survival, thereby contributing to treatment resistance. Moreover, emerging evidence suggests novel mechanisms of interaction between GBM cells and the TME, including neuron–glioma interfaces and neurotransmitter-based interactions. These discoveries shed light on additional layers of complexity within the GBM TME, revealing further opportunities for therapeutic intervention [18].

Accordingly, due to the complex microenvironment of GBM, it poses significant challenges in oncology research, which traditional in vitro models struggle to replicate accurately. Several studies have underscored that inter-tumor and intra-tumor heterogeneity in GBM significantly contribute to unsatisfactory outcomes in clinical and pre-clinical trials. Effective targeting of GBM heterogeneity demands an understanding of the drivers behind sub-clonal variation, such as vascularity, hypoxia, and inflammation. Advanced in vitro GBM models, encompassing both GBM tumors and normal brain tissues, offer a promising avenue to achieve this insight. However, traditional Petri-dish-based assays fall short in capturing the complexity of tumors, thus limiting their potential for determining predictive functional biomarkers [18,19,20].

Organ-on-a-chip (OOC) technology has emerged as a revolutionary approach to address these limitations. OOC enables the simulation of human functional units, including tissues and organs, ex vivo on microscopic cell and tissue culture platforms. By incorporating the fundamental components necessary for functional units, such as multicellular components, extracellular matrix (ECM), and physicochemical microenvironmental factors, OOC compensates for the shortcomings of previous cell culture methods. It offers numerous advantages, including three-dimensional (3D) dynamic culture, controlled physicochemical stimulation, low cost, high throughput, and high reliability. Moreover, when combined with imaging instruments, OOC enables real-time monitoring of cell biology changes, facilitating the recording of cell behavior alterations during disease states and responses to drugs [21,22,23,24].

In recent years, OOC technology, integrated with microfluidics and 3D bioprinting, has been leveraged to model the GBM TME, giving rise to “GBM-on-a-chip” platforms. These platforms replicate the functional units of GBM tumors in vitro, replicating the cellular composition and anatomical structure of both the target tumor and normal brain tissues. By precisely regulating complex factors within the GBM TME in a spatiotemporal, controllable manner, GBM-on-a-chip provides bionic support at the cellular and tissue levels. Consequently, it has been extensively employed to investigate biological mechanisms and therapeutics in GBM, offering promising applications in personalized precision medicine and immunotherapy [25,26,27,28]. Microfluidic devices offer precise control over fluid flow and microenvironmental conditions, allowing researchers to recreate key aspects of the GBM microenvironment, such as nutrient gradients, hypoxia, and immune cell interactions. Bioprinting techniques enable the precise deposition of GBM cells, stromal cells, and extracellular matrix components within microfluidic channels, resulting in three-dimensional GBM-on-a-chip models that closely resemble in vivo tumor architecture and function [29,30,31,32].

The application of GBM-on-a-chip platforms is diverse and promising, with researchers utilizing these models to investigate GBM progression mechanisms, identify novel drug candidates, and develop personalized therapeutic approaches. Moreover, GBM-on-a-chip platforms hold great potential for advancing GBM immunotherapy by providing a platform for studying immune cell interactions within the tumor microenvironment and evaluating the efficacy of immunomodulatory therapies. Despite their potential, GBM-on-a-chip platforms face challenges, including the need for improved biomimetic designs, validation against in vivo models, and scalability for high-throughput applications. Future advancements in microfluidics, bioprinting, and tissue engineering will further enhance the capabilities of GBM-on-a-chip platforms for studying GBM biology and developing innovative therapeutic strategies [33,34]. 

Biosensor-enhanced GBM-on-a-chip platforms represent a cutting-edge advancement in the study of the TME dynamics of GBM. These platforms integrate biosensors into the microfluidic devices or 3D bioprinted constructs, allowing real-time monitoring of various parameters critical for understanding GBM progression and response to therapy. One of the key advantages of biosensor-enhanced GBM-on-a-chip platforms is their ability to provide continuous and precise measurements of important biochemical and biophysical parameters within the TME. For example, biosensors can monitor pH levels, oxygen tension, glucose concentration, and lactate production, reflecting metabolic activity and microenvironmental changes associated with tumor growth and progression. This real-time data acquisition enables researchers to capture dynamic alterations in the TME, such as metabolic shifts, nutrient gradients, and hypoxic conditions, which are known to influence GBM aggressiveness and therapeutic resistance [35,36]. Furthermore, biosensors integrated into GBM-on-a-chip platforms can facilitate the study of cell–cell and cell–matrix interactions within the TME. For instance, biosensors capable of detecting specific biomarkers or signaling molecules can provide insights into paracrine signaling events between GBM cells, stromal cells, and immune cells. By monitoring the secretion of cytokines, growth factors, and extracellular vesicles in real-time, researchers can elucidate the complex interplay between different cell types in the TME and identify potential therapeutic targets [37]. Moreover, biosensor-enhanced GBM-on-a-chip platforms offer a valuable tool for drug screening and personalized medicine approaches. By continuously monitoring cellular responses to pharmacological agents, biosensors can help identify optimal drug dosages, treatment regimens, and drug combinations that effectively target specific pathways or vulnerabilities within the GBM TME. This dynamic and personalized approach to drug testing can potentially improve the efficacy of therapeutic interventions and reduce the likelihood of drug resistance. Collaboration between researchers, clinicians, and industry partners will be essential for translating findings from GBM-on-a-chip models into clinical applications to improve patient outcomes [33,38]. 

The aim of this review is to critically evaluate the utility and potential of biosensor-enhanced organ-on-a-chip (OOC) models in studying the complex dynamics of the glioblastoma tumor microenvironment (TME). Specifically, the review focuses on elucidating cellular interactions, understanding heterogeneity within the TME, and exploring the therapeutic implications of these findings. Through this comprehensive assessment, the goal is to provide insights into the effectiveness of OOC technology in advancing GBM research and therapy, ultimately contributing to improved patient outcomes in the fight against this devastating disease.

## 2. Glioblastoma Tumor Microenvironment

The glioblastoma (GBM) tumor microenvironment (TME) is characterized by significant polymorphism and heterogeneity, contributing to its multifaceted complexity. It encompasses a diverse array of malignant cells, stromal cells, tissue-resident cell types, immune cells, extracellular matrix (ECM) components, and soluble factors such as cytokines and chemokines. Bidirectional interactions between these cellular components and the TME are crucial for maintaining normal tissue homeostasis and driving tumorigenesis [39]. The interactions of tumor cells with stromal cells play a pivotal role in influencing disease progression and patient outcomes. For instance, the secretion of cytokines by microglia and tumor-associated macrophages (TAMs) contributes to the establishment of an immunosuppressive environment within GBM [39]. Additionally, cancerous cells within the GBM TME engage in interactions with neoplastic cells through suppressive receptors and gangliosides, thereby enhancing tumor immune-escaping capabilities [40]. The ECM within the GBM TME serves as a dynamic scaffold for intra-tumoral niches, which form from interactions between different tumor cells (including infiltrating, proliferative, or stem cell populations) and noncancerous immune cells. These interactions lead to the formation of complex regions within the tumor, characterized by variations in cellular composition and microenvironmental conditions. These regions may include proliferative tumor cell clusters, perivascular areas surrounding the vasculature, necrotic or peri-necrotic zones, and hypoxic or peri-hypoxic regions [41]. The subsequent sections will delve into the comprehensive analysis of the extracellular matrix (ECM), immune components, neural constituents, chemical constituents, and stem cell populations associated with glioblastoma (Figure 1).

Illustration depicting the intricate tumor microenvironment of glioblastoma, highlighting key structural components including the extracellular matrix (ECM), immune cells, neural constituents, chemical constituents, and stem cell populations. The ECM provides structural support and signaling cues for tumor progression, while immune components interact dynamically with tumor cells, influencing both anti-tumor immunity and immunosuppressive mechanisms. Neural constituents may contribute to tumor growth and invasion through neurovascular interactions. Hypoxic conditions and acidity in the necrotic sections create a hostile microenvironment that promotes tumor aggressiveness and therapy resistance. Additionally, stem cell populations, such as glioblastoma stem cells, play a crucial role in tumor initiation, progression, and therapeutic resistance. Understanding the complex interplay between these components is essential for developing targeted therapies against glioblastoma (created with Biorender).

### 2.1. Extracellular Matrix

The brain extracellular matrix (ECM) presents distinct features, comprising a complex array of matrix molecules, including fibronectin, laminins, tenascins, hyaluronic acid (HA), and chondroitin sulfate proteoglycan (CSPG) [42]. Collagen’s presence in the brain is mainly confined to blood vessels and the glia limitans, unlike other body tissues. The primary assembly of the adult brain ECM is believed to consist of a hyaluronan-lectican-tenascin R-complex [43]. Understanding ECM composition and architecture presents opportunities for the development of innovative therapeutic strategies aimed at overcoming the barriers to effective treatment in GBM [44]. 

A recent study by Yan and his colleagues in 2021 shed light on this aspect, demonstrating heightened expression levels of HA, hyaluronic acid synthase 3 (HAS3), and the HA receptor CD44 correlated inversely with the prognosis of glioma patients [45]. Integrins, transmembrane glycoproteins, play a pivotal role in facilitating cell-to-ECM interactions by binding to various ligands such as collagen, fibronectin, laminin, and tenascin-C (TN-C) [46,47]. Furthermore, TN-C is prominently associated with blood vessels, which proves advantageous for glioma cells as they rely on a steady supply of nutrients from circulation [42,48]. 

In vitro studies have demonstrated that glioblastoma cell lines with TN-C knockdown exhibit increased adhesion to extracellular matrix (ECM) components, facilitated by the upregulation of the FAK (focal adhesion kinase) pathway [49]. Furthermore, in mouse xenograft models, tumors derived from TN-C knockdown cells display reduced invasiveness, with tumor cells confined to more clearly defined tumor borders compared to control tumors of similar size, suggesting that TN-C modulates glioma cell invasion and migration through the ECM without influencing cell proliferation [50]. Investigations by Zhang and his colleagues in 2019 suggest that TN-C expression in glioma tissue may be promoted by the interleukin-33 (IL-33)-ST2-nuclear factor kappa B (NFκB) pathway [51]. It indicates the multifaceted role of TNC in facilitating GBM invasion and highlights potential therapeutic targets for inhibiting TNC-mediated invasion pathways in GBM [52,53].

Fibronectin (FN) enhances the adherence of glioma stem-like cells in a concentration-dependent manner, mediated by increased levels of matrix metallopeptidases. In a study conducted by Yu et al. in 2018, pre-coated fibronectin (FN) was found to lead to increased adherence of glioblastoma stem cells (GSCs). The activation of the focal adhesion kinase/paxillin/AKT signaling pathway was implicated in FN-mediated modulation of GSCs. FN suppressed p53-mediated apoptosis and upregulated P-glycoprotein expression, rendering GSCs more resistant to alkylating agents like carmustine [54,55,56,57].

Glioblastoma cells exhibit a chemotactic migratory response towards fibronectin, with migration rates correlating with fibronectin concentration [58]. In an experimental animal model, fibronectin knockdown tumors exhibited a mean survival advantage of 23 days over wild-type tumors. Furthermore, brain samples from animals bearing fibronectin-knockdown tumors showed delayed Treg recruitment. These findings collectively suggest that fibronectin serves as a pivotal mediator of glioma progression, as its inhibition leads to the delay of both tumor progression and immunosuppression [59]. Recent investigations have also implicated GBP2, an interferon-inducible large GTPase, in the induction of fibronectin expression via the Stat3 pathway [60]. These findings collectively highlight the regulatory role of fibronectin in glioblastoma invasion dynamics, suggesting its potential as a prognostic biomarker given its association with poor clinical outcomes [61].

Fibulin-3, an essential glycoprotein within the ECM framework of GBMs, is implicated in driving tumor progression by modulating Notch and NF-κB signaling pathways [62]. Despite its limited distribution in the body and absence in the adult brain, fibulin-3 exhibits significantly heightened expression levels within GBMs, where it assumes a pivotal role in ECM-mediated signaling processes. This upregulation endows Fibulin-3 with new functions as an autocrine/paracrine activator of Notch and NF-κB signaling pathways, consequently amplifying GBM invasion, vascularization, and the survival of tumor-initiating cell populations. Importantly, elevated Fibulin-3 levels correlate with unfavorable patient survival and serve as a marker for regions of active tumor progression within GBM [63].

Laminins, abundant components of the glioma microenvironment, predominantly localize to the basal lamina of blood vessels, particularly in regions of brain–tumor interface. In vitro investigations reveal that glioma cell lines exhibit enhanced adhesion and invasion characteristics in the presence of laminins [64]. Specifically, laminin exposure prompts glioma cells to develop F-actins, robust stress fibers, and increased pseudopodia formation, facilitating cell adhesion and invasiveness. During glioma progression, laminin-9 (alpha4beta2gamma1) transitions to laminin-8 (alpha4beta1gamma1), which is implicated as a primary mediator of glioblastoma invasion [65,66]. 

To assess the feasibility of this approach and optimize treatment conditions, an in vitro model was established. Human glioblastoma multiforme cell lines M059K and U-87MG were cocultured with normal human brain microvascular endothelial cells (HBMVECs) for this purpose. Western blot analysis and immunohistochemistry confirmed that antisense treatment effectively inhibited laminin-8 protein synthesis. Antisense oligonucleotides targeting both α4 and β1 chains of laminin-8 significantly blocked invasion of cocultures through Matrigel. On average, invasion was reduced by 62% in cocultures of U-87MG with HBMVEC and by 53% in cocultures of M059K with HBMVEC. These results suggest that laminin-8 may contribute to glioma progression and recurrence not only as part of the neovascularization process but also by directly enhancing the invasive potential of tumor cells [67].

Normal brain tissue ECMs typically exhibit low collagen content, whereas the glioma microenvironment, particularly around vessels, demonstrates a significant increase in collagen deposition [68]. Specifically, the upregulation of collagen alpha-2(I) chain (COL1A2), collagen type III–VI collagen, and type XVI mRNA in GBMs compared to normal brain tissue correlates with poor progression-free survival and overall survival [69,70,71]. Furthermore, the study showed that the lysyl oxidase inhibitor β-aminopropionitrile disrupts collagen structure within the tumor microenvironment. These findings underscore the significance of the tumor microenvironment, particularly the role of collagen structure, in tumor angiogenesis and the progression of brain tumors [72].

Various families of receptors and pathways are implicated in collagen-mediated GBM invasion. Prolyl-4-hydroxylase subunit 2 (P4HA2), a collagen modification enzyme, exhibits higher transcriptional levels in glioma samples compared to normal brain tissue, correlating with glioma grading and patient survival [73]. Endo180 (CD280), a collagen-binding receptor overexpressed in GBMs, facilitates collagen internalization and is critical in glioma cell invasion into the ECM. Through promoter analysis studies, it was demonstrated that this heightened expression is, in part, mediated via TGF-beta signaling [74]. HSP47, a chaperone protein for collagen, is significantly overexpressed in GBMs and promotes GBM stem-like cell survival through modulation of the tumor microenvironment via the TGF-β pathway [75].

Matrix metalloproteinases (MMPs) are a group of zinc-dependent endopeptidases that play crucial roles in degrading various components of the extracellular matrix (ECM) through integrin mediation. They are involved in tissue structural changes, cell proliferation, and cell migration [76]. In GBM, at least 23 members of the MMP family have been identified, and GBM cells secrete various MMPs to degrade ECM proteins such as fibronectin, laminins, collagen, and gelatin, promoting cell migration and releasing activated proteins through cleavage. Specific subgroups of MMPs, including MMP-1, -2, -7, -9, -11, -12, -14, -15, and -25, have been closely associated with glioma grading and GBM development [77]. High levels of MMP-9 and MMP-2 are particularly linked to higher tumor grades, reduced response to chemotherapy, and worse survival outcomes [78,79,80,81,82]. Additionally, uPA/uPAR interaction with integrin receptors leads to downstream signaling activation, including focal adhesion kinase (FAK), extracellular signal-regulated kinase (ERK), and Src pathways, facilitating cell migration. Other signaling pathways implicated in MMP activation and GBM invasion include the Sonic Hedgehog signaling pathway, which increases MMP-9/-2 expression via the PI3K/AKT pathway, and IL-17A, which may induce MMP-9/-2 overexpression through PI3K/AKT signaling [83]. 

In summary, the ECM and its constituents emerge as critical determinants in the pathogenesis of GBM and influence therapeutic responses. The intricate interplay between tumor cells and the ECM underscores its multifaceted role in GBM progression. Comprehensive comprehension of these complex interactions provides crucial insights into devising targeted therapeutic approaches aimed at surmounting the formidable challenges posed by the ECM microenvironment in GBM management. The ECM orchestrates various cellular processes integral to GBM pathogenesis, including cell adhesion, migration, invasion, and therapeutic resistance. Recognition of the pivotal role of the ECM in GBM pathophysiology opens avenues for the development of targeted therapeutic strategies. Strategies aimed at inhibiting ECM-mediated cell adhesion, migration, and invasion hold promise for attenuating tumor aggressiveness and improving patient outcomes in GBM. In essence, elucidating the intricate dynamics between GBM cells and the ECM offers a foundation for the advancement of precision medicine approaches tailored to counteract the deleterious effects of the ECM microenvironment. Leveraging these insights may ultimately lead to the development of more effective therapeutic interventions capable of mitigating the challenges posed by the ECM in GBM treatment and ultimately enhancing patient survival. Table 1 illustrates the roles of ECM components in GBM pathogenesis.

### 2.2. The Immune Composition within the Glioma Microenvironment

GBM exhibits a complex interaction with the immune system. Within the tumor microenvironment, a diverse array of immune cells and signaling molecules play crucial roles in tumor progression and therapeutic responses. Microglia, the resident immune cells of the brain, along with glioma-associated macrophages (GAMs), constitute a significant portion of the tumor cell population. These innate immune cells, alongside infiltrating neutrophils, contribute to the immunosuppressive milieu within GBM. Additionally, adaptive immune cells, including tumor-infiltrating lymphocytes (TILs), regulatory T cells (Tregs), and dendritic cells (DCs), modulate immune responses within the tumor microenvironment. The immune landscape of GBM is further shaped by cytokines, chemokines, and immune checkpoint molecules, such as programmed cell death protein 1 (PD-1) and cytotoxic T-lymphocyte-associated protein 4 (CTLA-4), which regulate immune cell activation and function. GBM cells express tumor-associated antigens, which can be targeted by the immune system. However, the immunosuppressive microenvironment of GBM, characterized by factors such as interleukins, tumor necrosis factor-alpha (TNF-α), transforming growth factor-beta (TGF-β), and C-C motif chemokine ligands (CCLs), presents significant challenges for effective anti-tumor immune responses. Despite these challenges, immunotherapeutic strategies targeting immune checkpoints, tumor-associated antigens, and immune cell function show promise for GBM treatment [84,85]. Understanding the complex interaction between tumor cells and the immune system within the GBM microenvironment is critical for developing effective immunotherapies. Recent advancements in single-cell analytical techniques have revealed the intricate interplay between tumor cells, immune effectors, and host cells in the glioma microenvironment. Factors such as metabolic stress, hypoxia, and damage-associated molecular patterns contribute to the immunosuppressive milieu in GBM. Therefore, therapeutic strategies targeting both glioma immunogenicity and the immunosuppressive microenvironment hold promise for improving treatment outcomes. By studying immune components of GBM, we can gain insights into the mechanisms underlying tumor progression and immune evasion, facilitating the development of biosensor-enhanced GBM chip models. The subsequent sections will explore the immune composition of the glioblastoma microenvironment, focusing on the characterization of microglia and glioma-associated macrophages, neutrophils, dendritic cells, and tumor-infiltrating lymphocytes, among others (Figure 2).

#### 2.2.1. Microglia and Glioma-Associated Macrophages (GAMs)

Microglia and glioma-associated macrophages (GAMs) are predominant immune cell populations within the microenvironment of primary gliomas, situated within the central nervous system (CNS) milieu. These myeloid cell lineages have been extensively studied, revealing their diverse origins, phenotypes, and functional plasticity [86]. GAMs constitute a considerable portion, up to 30–40%, of the bulk tumor mass in GBM and predominate lymphocyte infiltration, highlighting their importance in GBM initiation and progression. Various factors released by glioma cells can attract GAMs to tumor sites. Subsequently, GAMs can exert multiple pro-tumorigenic activities. Accumulating studies consistently demonstrate the pivotal role of factors released by GAMs in promoting the proliferation and invasion of GBM [87].

The glioma microenvironment profoundly influences the behavior of microglia and GAMs, which exhibit dynamic phenotypes. Under different cytokine and chemokine conditions, macrophages can polarize into M1 or M2 phenotypes, with M1 macrophages exerting immune-supportive and anti-cancer functions in contrast to M2 macrophages. Despite the complexity of GAM phenotyping, these myeloid lineages typically exhibit an immunosuppressive profile in the glioma microenvironment. GAMs produce anti-inflammatory cytokines (such as IL-4, IL-10, and TGF-β), tumor-promoting factors (including IGF-1, EGF, and PDGF), angiogenesis mediators (like VEGF and IL-8), and metabolic disruptors (e.g., ARG1 and IDO) [88,89,90]. The abundance and polarization of GAMs are influenced by various factors, including the isocitrate dehydrogenase (IDH) mutation status of GBM. In IDH wild-type GBM, characterized by a higher prevalence and severity, there is an increased presence of macrophages expressing CD11b+ and CD45+, which exhibit an activated state associated with anti-inflammatory responses and are correlated with poorer patient outcomes compared to IDH-mutant GBM, where microglial concentrations are higher, fostering a pro-inflammatory milieu [91,92].

Microglia and microglia-conditioned medium have been implicated in promoting the invasion of glioma in vitro, suggesting a role for substances released by GAMs [93]. Conversely, oligodendroglia and endothelial cells only exhibit weak stimulation of glioma cell motility. This motility-promoting activity is attenuated in glioma cells when transforming growth factor (TGF)-β is knocked down, indicating a dependence on microglia-derived TGF-β for glioma invasion. Furthermore, GBM invasion promoted by TGF-β is associated with upregulated integrin expression. TGF-β also induces the expression of matrix metalloproteinase (MMP)-2 and suppresses tissue inhibitor of metalloproteinases (TIMP)-2, leading to accelerated extracellular matrix breakdown [94,95]. MMPs facilitate not only the invasion of GBM into the brain parenchyma but also GBM proliferation. In response to factors released by GBM cells, membrane type 1 metalloprotease (MT1-MMP) expression is upregulated in GAMs but not in tumor cells. Notably, microglia release TGF-β, which triggers the release of pro-MMP2 from GBM. Pro-MMP2 is subsequently activated by GAM-expressed MT1-MMP. The toll-like receptor (TLR) adapter protein MyD88 or p38 deletion inhibits MT1-MMP expression and GBM proliferation, suggesting that GAMs’ TLR and p38 MAPK pathways mediate the high expression of MT1-MMP and increased proliferation capacity [96].

Pleiotrophin (PTN)-PTPRZ1 paracrine signaling supports GBM malignant proliferation. GAMs secrete abundant PTN, which binds to its receptor PTPRZ1, stimulating GBM proliferation. In a study conducted by Shi and colleagues in 2017 [97], it was reported that GAMs secrete abundant pleiotrophin (PTN), which stimulates glioma stem cells (GSCs) through its receptor PTPRZ1, thereby promoting malignant growth of glioblastoma (GBM) through PTN–PTPRZ1 paracrine signaling. The expression of PTN correlates with the infiltration of CD11b+/CD163+ TAMs and indicates a poor prognosis for GBM patients. Co-implantation of M2-like macrophages (MLCs) enhanced GSC-driven tumor growth, but silencing PTN expression in MLCs attenuated their pro-tumorigenic activity. The PTN receptor PTPRZ1 is preferentially expressed in GSCs and also serves as a predictor of poor prognosis in GBM. Disrupting PTPRZ1 abrogated GSC maintenance and its tumorigenic potential. Furthermore, blocking the PTN–PTPRZ1 signaling using shRNA or an anti-PTPRZ1 antibody significantly suppressed GBM tumor growth and prolonged animal survival. The study revealed a critical molecular crosstalk between TAMs and GSCs mediated by PTN–PTPRZ1 paracrine signaling, supporting the malignant growth of GBM [97].

The CCL2/CCR2/IL-6 loop promotes GBM invasion. Glioma-derived CCL2 acts on microglia, triggering IL-6 production, which in turn promotes GBM invasion. GAMs highly express CCL8, promoting pseudopodia formation in GBM cells [98]. 

Furthermore, colony-stimulating factor-1 (CSF-1) secreted by GBM acts as a chemoattractant for GAMs, facilitating M2-like activation and GBM proliferation [99,100]. GAMs contribute to GBM-associated angiogenesis by promoting abnormal vessel formation. Depleting GAMs reduces GBM vessel density, indicating their importance in tumor angiogenesis [101]. GAMs isolated from gliomas overexpress proangiogenic factors such as VEGF and CXCL2. The interaction of RAGE with its ligands promotes tumor angiogenesis, driven by GAMs. CSF-1 increases IGFBP1 expression, promoting angiogenesis [101,102,103].

In a study conducted by Zhang and colleagues in 2019, it was demonstrated that chemokine (C-C motif) ligand 8 (CCL8) is highly expressed by GAMs and contributes to pseudopodia formation by GBM cells. The presence of CCL8 in the glioma microenvironment promotes the progression of tumor cells. Moreover, CCL8 induces invasion and stem-like traits in GBM cells, with CCR1 and CCR5 identified as the main receptors mediating CCL8-induced biological behavior. Additionally, CCL8 dramatically activates ERK1/2 phosphorylation in GBM cells, and blocking GAM-secreted CCL8 using neutralized antibodies significantly decreases the invasion of glioma cells [104]. 

Therapeutic approaches targeting GAMs in GBM encompass a range of strategies, including depletion, repolarization, enhancing phagocytosis, and reducing recruitment. TAM depletion strategies have been explored extensively. Depletion methods such as CD11b-HSVTK-mediated microglia depletion using ganciclovir have shown promising results in preclinical models, reducing vessel density and tumor volume in GBM-bearing mice [101,105,106]. However, the selective depletion of resident microglia or total myeloid cell populations underscores the complex role of GAMs in tumor progression, with studies indicating both tumor-promoting and inhibitory effects upon depletion [101]. The timing of GAM depletion may also impact efficacy, with early-stage GBM patients potentially benefiting more from this approach [107]. 

Repolarizing GAMs from a pro-tumoral M2-like phenotype to an anti-tumoral M1-like phenotype is another strategy. Inhibiting factors such as colony-stimulating factor-1 (CSF-1) or its receptor, CSF-1R, have shown promise in preclinical models, leading to reduced tumor growth and prolonged survival [99,108]. Combination therapies targeting additional pathways, such as phosphatidylinositol 3-kinase (PI3K) and the insulin-like growth factor-1 (IGF-1) receptor, have demonstrated efficacy in overcoming resistance to CSF-1R inhibition [109]. Enhancing the phagocytosis of TAMs presents another avenue for intervention. Inhibiting the CD47-SIRPα axis, which acts as an anti-phagocytic signal, has shown promise in preclinical models, leading to increased tumor phagocytosis and reduced tumor burden. Combining CD47 blockade with temozolomide (TMZ) further enhances phagocytosis and immune responses, offering a synergistic therapeutic approach [110,111].

Reducing TAM recruitment is crucial for inhibiting tumor progression. Targeting chemokine axes such as CX3CL1/CX3CR1 and SDF-1/CXCR4 has shown efficacy in preclinical models, leading to inhibited TAM recruitment and tumor growth [112]. Despite promising preclinical results, translating these findings into clinical applications remains challenging. Clinical trials targeting TAMs have shown mixed results, highlighting the need for further research to optimize therapeutic strategies and improve outcomes for GBM patients.

Clinical trials are currently underway to explore the efficacy of various agents, either alone or in combination therapy, to exploit GAMs as therapeutic targets and ultimately improve outcomes in GBM patients. One such clinical trial, registered as NCT02323191, investigated the combination of Emactuzumab (RG7155), a therapeutic anti-colony-stimulating factor-1 receptor (CSF-1R) antibody, with atezolizumab, a programmed cell death-1 ligand (PD-L1)-blocking monoclonal antibody (mAb). This phase I study demonstrated a significant objective response rate (ORR) alongside a manageable safety profile, suggesting the potential efficacy of this combination in GBM treatment. Additionally, the stromal cell-derived factor-1 (SDF-1)/C-X-C motif chemokine 12 (CXCL12) axis plays a crucial role in the recruitment of GAMs. In line with this, a pilot phase I/II trial (NCT01977677) focused on assessing the safety profile and optimal dosage of Plerixafor, a CXCR4 inhibitor, when administered following radiation therapy plus temozolomide (TMZ). This trial aimed to evaluate the efficacy of Plerixafor in treating patients with newly diagnosed high-grade glioma, leveraging its ability to disrupt TAM recruitment mediated by the SDF-1/CXCL12 axis. These clinical trials underscore the growing interest and efforts in targeting GAMs as a therapeutic strategy in GBM treatment. By elucidating the complex interactions within the tumor microenvironment and developing targeted interventions, such approaches hold promise for improving outcomes and advancing the management of GBM.

#### 2.2.2. Neutrophils

Neutrophils, a subset of myeloid-derived suppressor cells, exhibit similar immunosuppressive properties to tumor-associated macrophages, hindering the function of tumor-specific effector T cells. Their primary function involves triggering inflammatory responses through various mechanisms, including phagocytosis, the production of reactive oxygen species (ROS) and nitrogen species, and the release of cytotoxic granules [113]. However, in the context of GBM, neutrophils contribute to oncogenic processes such as tumor initiation, proliferation, and dissemination through a pro-tumorigenic feedback loop. They transfer myeloperoxidase granules to tumor cells, augmenting ROS production and lipid peroxidase accumulation, leading to necrosis and the subsequent recruitment of more neutrophils [113,114]. Additionally, neutrophils promote angiogenesis and suppress the functions of macrophages, dendritic cells (DCs), and natural killer (NK) cells, thereby dampening the immune response and facilitating tumor cell extravasation. In the GBM microenvironment, several cytokines and chemokines, including IL-8, play significant roles in neutrophil recruitment. Glioma cells produce IL-8, a potent neutrophil chemoattractant, under the influence of cytokines such as IL-1 and TNF-α [115]. Additionally, high-mobility group box 1 (HMGB1), derived from neutrophil extracellular traps (NETs), binds to the receptor for advanced glycation end products (RAGE) in glioma tissue, activating the NF-κB signaling pathway and promoting IL-8 secretion, thereby enhancing neutrophil infiltration [116]. Other mediators, such as IL-1β and the long noncoding RNA LINC01116, also contribute to neutrophil recruitment by glioma cells [117].

Once recruited to the GBM microenvironment, neutrophils acquire unique phenotypes and functionalities. Neutrophils infiltrating glioma tissues have been associated with tumor grade and progression. In GBM, neutrophils play a role in upregulating the expression of the S100A4 protein, which mitigates the mesenchymal phenotype and contributes to acquired resistance to anti-VEGF therapy [118,119]. Inhibition of S100A4 using small hairpin RNA (shRNA) has been shown to enhance the efficacy of bevacizumab treatment and improve overall survival in preclinical models [119]. Several therapeutic strategies targeting neutrophils in GBM are under investigation. These include inhibiting neutrophil recruitment, blocking their immunosuppressive functions, and utilizing neutrophils as drug delivery vehicles. Furthermore, recent research has explored the use of neutrophils as a drug delivery system in conjunction with chimeric antigen receptor T (CAR-T) cells in vivo [120]. These findings underscore the multifaceted role of neutrophils in GBM progression and highlight their potential as therapeutic targets or delivery vehicles for novel treatment modalities. Additionally, strategies to inhibit neutrophil extracellular trap (NET) formation or to enhance the efficacy of existing immunotherapies by targeting neutrophils are being explored. The occurrence of NETs within tumors of the central nervous system (CNS) has been infrequently documented in scientific literature. However, recent investigations have unveiled the presence of NETs within grade IV glioma tissues, discerned through immunostaining techniques targeting myeloperoxidase (MPO) and citrullinated histone H3 (CitH3). Substantially augmented levels of NETs were discerned in high-grade glioma tissues relative to their low-grade counterparts. Moreover, functional analyses have unveiled the involvement of NETs in the proliferation and invasive behaviors of glioblastoma multiforme (GBM) cells, facilitated by the interaction between high mobility group box 1 (HMGB1) and the receptor for advanced glycation end products (RAGE). This interaction serves to instigate the activation of the NF-κB signaling cascade, thereby modulating cellular processes pertinent to GBM progression [116]. Furthermore, exploiting the natural ability of neutrophils to penetrate the blood–brain barrier (BBB) for drug delivery holds potential for the development of novel therapies for GBM. Neutrophils loaded with anticancer drugs or engineered to deliver therapeutic agents directly to tumor sites offer a promising avenue for targeted GBM therapy. In summary, understanding the multifaceted roles of neutrophils in the GBM microenvironment is crucial for the development of effective immunotherapeutic strategies. Targeting neutrophils holds promise for improving treatment outcomes and overcoming the challenges associated with GBM therapy.

#### 2.2.3. Dendritic Cells

Dendritic cells (DCs) play pivotal roles in immune responses, orchestrating the activation and regulation of various immune effector cells. Within normal brain parenchyma, DCs are typically absent but are localized in vascular-rich compartments such as the choroid plexus and meninges, suggesting potential migratory routes for peripheral DCs into the central nervous system (CNS) [121,122]. In pathological conditions, DCs migrate to the brain and spinal cord via afferent lymphatics or high endothelial venules. While the specific role of DCs in GBM pathogenesis is still evolving, they likely participate in recognizing and presenting tumor antigens to initiate T cell-mediated responses [123]. Myeloid DCs, with the capacity to differentiate into Langerhans cells or interstitial dendritic cells, play pivotal roles in this process.

DCs within the tumor microenvironment (TME) undergo modulation by various molecules that hinder their activation and drive them toward a suppressive phenotype. These molecules include vascular endothelial growth factor (VEGF), prostaglandin E2 (PGE2), interleukin-10 (IL-10), and colony-stimulating factor-1 (CSF-1) [124]. Regulatory DCs within the TME can trigger regulatory T cell (Treg) activation and dampen cytotoxic T lymphocyte (CTL) recruitment to the tumor site by secreting suppressive cytokines like IL-10 and transforming growth factor-beta (TGF-β) [125,126,127]. Similarly, plasmacytoid DCs (pDCs) within the TME may exhibit an immunosuppressive phenotype characterized by reduced type I interferon production and diminished expression of costimulatory molecules, exacerbating immune evasion by tumors. Exposure of DCs to GBM antigens induces an immunosuppressive state characterized by elevated interleukin-10 expression and reduced levels of co-stimulatory molecules such as CD80, CD86, and interleukin-12. These alterations in the DC phenotype contribute to the immune evasion strategies employed by GBM cells [128].

Recent insights into the molecular pathways of DCs have identified potential therapeutic targets. For example, overexpression of nuclear factor erythroid 2-related factor (Nrf) in DCs induced by the GBM microenvironment hampers DC maturation and effector T-cell activation. Conversely, inhibition of Nrf2 pathways restores DC maturation and cytokine secretion [129]. Additionally, the m6A-binding protein YTHDF1 has been implicated in an immunosuppressive phenotype in DCs. Inhibition of YTHDF1 enhances cross-presentation of tumor antigens and primes CD8+ T cells in vivo [130].

DC-based immunotherapy, focusing on dendritic cell vaccines (DCVs), has shown promise in other cancers by eliciting tumor-specific immune responses through optimal tumor antigen presentation. In DCV therapy, DCs are harvested from peripheral blood or induced ex vivo from monocytes, pulsed with tumor antigens, and reintroduced into the patient to initiate a tumor-specific T cell-mediated response. Preclinical investigations in GBM have highlighted the potential efficacy of DCVs in inducing anti-tumor responses, with various methodologies aimed at optimizing their efficacy in GBM treatment [131]. These studies underscore the potential of DCVs in bolstering anti-tumor immunity and represent a promising avenue for further exploration in GBM therapy. Despite recent advances in understanding the molecular differentiation and pathways of DCs, there remains a need for comprehensive elucidation of the distinct roles of DCs in generating anti-tumor immune responses—particularly within the context of gliomas—as well as further delineation of mechanisms underlying tumor-mediated immunosuppression through DC phenotypic modulation.

#### 2.2.4. Natural Killer Cells

Natural killer (NK) cells, characterized by the expression of CD56 and lack of CD3, represent a potent subset of lymphoid cells proficient in antigen-independent immune surveillance against pathogens and stressed cells via “missing-self” recognition. These cells also demonstrate innate anti-tumor activity. However, in glioblastoma multiforme (GBM), NK cells are notably sparse among tumor-infiltrating immune cells, predominantly comprising the CD56^dim^CD16^−^ subtype [132]. Recent single-cell analyses have reaffirmed their scarcity in primary gliomas, with immature CD16^−^ NK cells predominating in IDHwt tumors, while cytotoxic CD16^+^ NK cells are more prevalent in IDH1mut tumors and brain metastases [133]. Furthermore, NK cell functionality is profoundly modulated by both classic and nonclassic human leukocyte antigen class I (HLA-I) molecules, predominantly through interactions with inhibitory killer immunoglobulin-like receptors. Despite the potential loss of heterozygosity in HLA-I genes within gliomas, complete deletion of these genes remains infrequent. The glioma tumor microenvironment (TME) further exacerbates NK cell suppression, as evidenced by the secretion of transforming growth factor-beta (TGFβ) by various cellular constituents, leading to downregulation of the NKG2D-activating receptor on NK cells isolated from GBM patients [134]. Consequently, the precise role of NK cells in anti-glioma surveillance remains a subject of ongoing inquiry. Nevertheless, adoptive NK cell therapy has shown promise in exerting cytotoxic effects against glioma cells, including glioma stem cells, in preclinical models. Recent investigations have identified potential therapeutic targets aimed at enhancing NK cell function by mitigating specific inhibitory mechanisms, thereby offering novel avenues for therapeutic intervention in GBM [135,136].

Several clinical trials are underway investigating the use of natural killer (NK) cell-based immunotherapies against GBM. One approach involves chimeric antigen receptor (CAR) NK cell therapy, where NK cells are genetically engineered to express CARs targeting specific antigens present on glioma cells. For instance, in trial NCT03383978, NK-92 cells modified to express anti-MUC1 CARs are being administered via intracranial injection to target MUC1-positive gliomas. Similarly, in trial NCT02839954, autologous NK cells are being infused intravenously after genetic modification to express NK-92/5.28. z cells targeting HER2-positive glioblastomas (GBMs) that have recurred or are refractory to treatment. Another approach is adoptive NK cell therapy, where patients receive infusions of autologous NK cells to enhance their anti-tumor immune response. In trial NCT00909558, autologous NK cells are administered intravenously to patients with gliomas, aiming to harness the cytotoxic capabilities of NK cells against tumor cells. In trial NCT04254419, autologous NK cells are injected directly into the tumor site to target malignant gliomas, while in trial NCT01875601, autologous NK cells combined with recombinant human interleukin-15 (rhIL-15) are infused intravenously to treat advanced solid tumors, including gliomas. Furthermore, trials like NCT02100891 are investigating the use of haploidentical hematopoietic cell transplantation (HCT) to enhance the efficacy of autologous NK cell therapy. In this trial, patients with high-grade gliomas receive autologous NK cells in combination with HLA haploidentical HCT to potentiate the anti-tumor immune response. Additionally, lymphokine-activated killer (LAK) cell therapy is being explored in trials such as NCT00814593, where autologous peripheral blood mononuclear cells (PBMCs) are cultured ex vivo to generate LAK cells for intra-tumoral injection in patients with primary GBM. Similarly, in trials like NCT00003067 and NCT00331526, LAK cells combined with aldesleukin are injected directly into the tumor site to treat primary, recurrent, or refractory malignant gliomas. Moreover, trial NCT00005813 investigates the use of LAK cells combined with the MDX477 bispecific antibody for intra-tumoral injection in patients with GBM expressing epidermal growth factor receptor (EGFR), aiming to enhance the targeting of tumor cells with bispecific antibodies. These trials collectively represent significant efforts to harness the potential of NK cell-based immunotherapies for the treatment of glioma, with various strategies being explored to enhance anti-tumor efficacy and patient outcomes.

#### 2.2.5. Lymphocytes

In the context of gliomas, there is a notable scarcity of lymphocyte infiltration, including natural killer (NK) and T cells, despite the potential for T lymphocytes to migrate into brain tumors, in contrast to the abundance of myeloid cells [85]. Tumor-infiltrating lymphocytes (TILs) serve as representative markers for the predominant presence of CD8+ cytotoxic T lymphocytes (CTLs), CD3+ T cells, and CD4+/FoxP3+ regulatory T cells (Tregs) within the glioblastoma (GBM) tumor microenvironment (TME). Primary gliomas typically have a limited lymphocyte presence, constituting only a small fraction of the immune cell population. CD8+ cytotoxic T cells and CD4+ helper T cells are observed in GBM [137]; however, the functionality of T cells is frequently compromised due to factors such as senescence, tolerance, anergy, or exhaustion [138]. The increased infiltration of CD3+ and CD8+ T cells into the GBM TME correlates with advanced tumor grade and is associated with improved disease prognosis and favorable post-operative treatment outcomes [132]. Conflicting observations regarding the clinical significance of T-cell infiltration in gliomas have been reported, thus obscuring the precise extent to which TILs can effectively initiate anti-tumor immune responses. Moreover, gliomas demonstrate poor responsiveness to immune checkpoint inhibitors (ICIs), despite the significant genetic and epigenetic heterogeneity within tumor lineages. While ICIs function by counteracting the exhaustion phenotype of cytotoxic T cells expressing the programmed cell death protein 1 (PD1) receptor, the efficacy of these agents is limited in gliomas. This resistance to ICIs may be attributed to intrinsic factors within cancer cells as well as extracellular milieu influences. Gliomas typically exhibit low immunogenicity and a reduced tumor mutational burden, resulting in fewer immunogenic neoantigens available for T-cell recognition [139,140]. Furthermore, the limited accessibility of tumor antigens to DCs within the brain parenchyma further hampers the priming of naive T cells, exacerbating the challenges associated with eliciting effective anti-tumor immune responses in gliomas.

To overcome the immunosuppressive barriers presented by the glioma TME, various immunotherapeutic strategies have been proposed, including dendritic cell vaccines, STING agonists, oncolytic viruses, and epigenetic modulators. These approaches aim to enhance glioma immunogenicity and promote cytotoxic T-cell activity, offering potential synergistic benefits when combined with immune checkpoint blockade. Additionally, chimeric antigen receptor (CAR) T-cell therapies hold promise for targeting specific tumor antigens, albeit with challenges related to intra-tumoral heterogeneity, antigen escape, and T-cell dysfunction. Mechanistic insights into the dysfunction of CAR T cells and other antigen-specific effector T cells continue to inform the development of novel therapeutic approaches aimed at overcoming resistance mechanisms and enhancing anti-tumor immunity in gliomas [141,142]. Table 2 delineates the various aspects of immune cell types within the glioma microenvironment, elucidating their respective roles, mechanisms/functions, and targeted therapeutic approaches.

### 2.3. Neural Components of the Glioblastoma Microenvironment

GBMs and other high-grade gliomas engage in intricate interactions with neural elements, leading to cellular and network-level alterations. Neural communication within the GBM TME emerges as a critical factor, where GBM cells integrate into the brain’s neural network to promote tumor growth and survival [143,144,145]. GBM’s invasive behavior involves connexin-43-mediated communication between glioma cells via gap junctions and microtubules. Studies have demonstrated that the transfer of micro ribonucleic acid (RNA), such as miR-19b, from glioma cells to astrocytes through connexin-43 plaques promotes GBM-astrocyte communication and facilitates tumor invasion into the brain parenchyma [146]. Furthermore, microtubules facilitate functional connections between glioma cells and astrocytes by propagating intracellular calcium signaling, which is crucial for cell proliferation and apoptosis resistance. Elevated intracellular calcium levels may lead to glutamate release, contributing to excitotoxicity and tumor progression [147].

Astrocytes, integral components of the blood–brain barrier (BBB), play essential roles in regulating brain fluid, metabolism, and blood homeostasis. GBM disrupts the astrocyte-BBB relationship by displacing astrocyte end-foot processes, enabling tumor invasion into uninhabited areas [148]. GBM cells secrete receptor activator of nuclear factor kappa beta ligand (RANKL), activating the nuclear factor kappa-light-chain-enhancer of activated B cells (NF-κB) pathway in astrocytes, promoting tumor invasiveness [149]. Additionally, sonic hedgehog (SHH) protein released by GBM cells activates GLI family transcription factors in nearby astrocytes, regulating cellular proliferation and differentiation [150]. Reactive astrocytes surrounding GBM lesions produce connective tissue growth factor (CTGF), promoting tumor growth, migration, and invasion. Inhibition of CTGF shows promise as a therapeutic strategy [151,152]. Furthermore, astrocytes release C-C motif ligand 20 (CCL20), activating the NF-κB pathway and hypoxia-inducible factor 1α (HIF-1α), contributing to GBM proliferation [153]. Moreover, GBM cells manipulate the extracellular matrix (ECM) of astrocytes to favor cell proliferation by inhibiting p53 expression (Figure 3).

Figure 3 presents a schematic representation illustrating the reciprocal communication between astrocytes and GBM cells within the tumor microenvironment. GBM disrupts the astrocyte–blood–brain barrier (BBB) interface, facilitating tumor invasion through the displacement of astrocyte end-foot processes. GBM cells secrete factors such as receptor activator of nuclear factor kappa beta ligand (RANKL) and sonic hedgehog (SHH) protein, activating nuclear factor kappa-light-chain-enhancer of activated B cells (NF-κB) signaling in astrocytes, thereby promoting tumor invasiveness. Reactive astrocytes surrounding GBM lesions produce connective tissue growth factor (CTGF), fostering tumor growth, migration, and invasion, suggesting CTGF inhibition as a potential therapeutic avenue. Moreover, astrocytes release C-C motif ligand 20 (CCL20) under hypoxic conditions, activating NF-κB and hypoxia-inducible factor 1α (HIF-1α) in GBM cells, thus contributing to GBM proliferation. Additionally, tumor-associated astrocytes (TAAs) promote glioma invasion via signaling pathways including NF-κB, interleukin (IL)-6/JAK/STAT, and SHH signaling, facilitated by connexin 43 (Cx43)-mediated gap junction formation. This bidirectional crosstalk ultimately supports GBM progression, emphasizing potential targets for therapeutic intervention.

The interplay between GBM cells and healthy glial cells constitutes a complex and bidirectional communication network. A bidirectional model has unveiled the mechanisms by which glioma cells modulate glial cells and regulate key signaling pathways, including extracellular signal-regulated kinase (ERK), protein kinase B (Akt), and c-Jun N-terminal kinase (JNK), through paracrine interactions mediated by the secretion of various proteins. GBM cells release a spectrum of proteins, including insulin-like growth factor-binding protein 2 (IGFBP2), myeloid-derived growth factors, and metalloproteinase inhibitor 2, which act as signaling mediators in the microenvironment [154]. These proteins play pivotal roles in altering the behavior of neighboring glial cells and facilitating tumor progression. Activation of the JNK and ERK signaling pathways has been linked to the evasion of apoptosis and the regulation of cancer cell proliferation, contributing to GBM aggressiveness. Meanwhile, the Akt pathway, in conjunction with neuroligin-3 (NLGN3) and the phosphoinositide 3-kinase (PI3K)-mammalian target of the rapamycin (mTOR) pathway, exerts profound effects on cell survival and growth regulation within the tumor microenvironment [155,156]. In vitro studies conducted by Gao et al. have provided compelling evidence supporting the targeting of specific signaling pathways as a potential therapeutic strategy for GBM. Particularly, the Akt inhibitor SC66 has emerged as a promising candidate for mitigating GBM progression by exerting profound effects on cell proliferation and apoptosis induction. They elucidated the remarkable inhibitory effects of SC66 on the proliferation of U87 and U251 GBM cells, concurrent with its ability to impede epithelial–mesenchymal transition (EMT)-mediated cell migration and invasion. Furthermore, SC66 exhibited a propensity to induce apoptosis in GBM cells while causing cell cycle arrest at the G0/G1 phase. Notably, the investigation unveiled the concentration-dependent downregulation of the AKT signaling pathway by SC66, indicating its mechanism of action. Additionally, treatment with SC66 resulted in a significant reduction in β-catenin nuclear translocation, indicative of its impact on β-catenin-mediated transcriptional activity. This observation was further supported by the suppression of T-cell factor/lymphoid enhancer factor (TCF/LEF) activity as demonstrated by the luciferase reporter assay. Importantly, the study demonstrated that augmenting β-catenin activity through IM12 could counteract the inhibitory effects of SC66 on GBM cell proliferation and metastasis [155].

The interaction between GBM and neurons remains an area of limited exploration, yet recent studies utilizing patient-derived glioma tumors have provided intriguing insights. These investigations have revealed that the presence of neuronal-regulated programmed death ligand 1 (PD-L1) is associated with a more favorable disease prognosis compared to GBM-regulated PD-L1. PD-L1 functions by binding to the programmed cell death protein 1 (PD-1) receptor, facilitating immune evasion by GBM cells.

In vitro studies have suggested that GBM cells promote their survival and proliferation through the activation of the intrinsic PD-L1 signaling pathway. Specifically, the binding of PD-L1 to the Ras protein has been shown to activate the extracellular signal-regulated kinase epithelial–mesenchymal transition (Erk-EMT) downstream signaling pathway, thereby promoting malignancy [157]. Notably, GBM cells have the capacity to induce the PD-L1 signaling pathway by secreting various factors, including the epidermal growth factor receptor (EGFR), interferon-α receptor, interferon-γ receptor, and toll-like receptor [158].

Regarding oligodendrocytes, which are responsible for myelinating nerves in the central nervous system (CNS), recent studies have implicated them in the upregulation of GBM invasiveness via the angiopoietin-2 signaling pathway. Angiopoietin-2 growth factor binds to the angiopoietin-1 receptor as well as integrins αvβ3, αvβ5, and α5β1 on endothelial cells (ECs), consequently inducing tumor angiogenesis and growth [159]. In vitro experiments utilizing anti-angiopoietin-2 neutralizing antibodies have demonstrated a decrease in GBM motility, providing further evidence of enhanced GBM invasiveness facilitated by oligodendrocytes. The study conducted by Kawashima et al. in 2019 investigated the interaction between stromal cells and GBM cells. The impact of stromal cells, specifically oligodendrocytes or fibroblasts, on the invasive potential of GBM cells was assessed through wound-healing assays and invasion assays. The results revealed that oligodendrocytes, in contrast to fibroblasts, significantly augmented the migratory and invasive capabilities of GBM cells. Furthermore, analysis of the conditioned medium obtained from oligodendrocytes demonstrated elevated levels of angiopoietin-2. Subsequent experiments showed that angiopoietin-2 significantly enhanced the motility of GBM cells. Moreover, the motility-enhancing effect of the conditioned medium derived from oligodendrocytes was markedly attenuated in the presence of an anti-angiopoietin-2-neutralizing antibody [160].

Paracrine interactions play a significant role in the communication between GBM cells and the nervous system. This communication involves the exchange of signals mediated by brain-derived neurotrophic factor (BDNF) and NLGN3 proteins, which contribute to tumor growth and survival. The regulation of BDNF involves its conversion from pro-BDNF to mature-BDNF, which can occur intracellularly through prohormone convertases or extracellularly via plasmin and matrix metalloproteinases (MMPs). While pro-BDNF exhibits anti-proliferative and anti-migratory effects, mature BDNF promotes cell proliferation, migration, and resistance to apoptosis [161]. In GBM, there is an increased production of mature BDNF compared to lower-grade gliomas, which correlates with the aggressiveness and invasive nature of GBM. MicroRNAs miR-210 and miR-489-3p have been identified as regulators of BDNF expression, with their overexpression leading to downregulation of BDNF and inhibition of cell proliferation, migration, and invasion [162,163].

Neuroligin, a synaptic cell surface protein, mediates trans-synaptic signaling, with NLGN3 being the isoform implicated in the GBM tumor microenvironment. Upon cleavage from neurons and precursor oligodendrocytes, NLGN3 induces transcriptional changes, including the upregulation of synapse-related genes in glioma cells. This upregulation ultimately promotes GBM proliferation and tumor growth by activating the PI3K-mTOR pathway. Experimental evidence supports the notion that GBM cell lines grow faster in the presence of NLGN3 compared to control conditions without NLGN3. Additionally, GBM recurrence is hypothesized to occur in specific brain regions, such as the basal ganglia, corpus callosum, and thalamus, where NLGN3 expression is elevated [164,165]. Overall, understanding the dynamics of these interactions within the glioblastoma microenvironment is essential for developing effective therapeutic strategies against this aggressive brain tumor. Table 3 outlines the interactions of neural components in the GBM microenvironment: functions, mechanisms, and significance.

### 2.4. Chemical Constituents in the GBM Microenvironment

In addition to the widely recognized role of genetic and epigenetic alterations in cancer pathogenesis, emerging evidence highlights the significance of metabolic reprogramming and interactions between stromal and tumor cells in driving tumor progression. Tumor cells undergo profound metabolic changes to sustain their rapid proliferation and survival, adapting to the demanding energy requirements associated with their aggressive growth [166]. One prime example of this metabolic reprogramming is observed in GBM, where glioma cells exhibit heightened uptake of glucose and other nutrients to fuel the robust biomass production characteristic of this malignancy. However, the metabolic landscape within the tumor microenvironment is far from homogeneous. Variations in the tumor vasculature, particularly due to the heterogeneous nature of neo-angiogenesis, can lead to disparities in nutrient and oxygen supply to different regions of the tumor. Consequently, intra-tumoral differences in vascularization directly impact the availability of oxygen within the tumor mass, exerting profound effects on the metabolic properties and energy utilization of cancer cells. Such fluctuations in metabolic substrates and oxygen levels contribute to the emergence of differential metabolic signatures among tumor cells within the same lesion.

These variances in metabolic profiles are further compounded by disparities in physiological factors, such as extracellular pH and oxygen concentration, between normal and cancerous cells within the tumor microenvironment. Dysregulation of these factors has been intricately linked to various aspects of tumor biology, including tumor progression, immunosuppression, therapy resistance, and metastasis.

The interplay between metabolic reprogramming, intra-tumoral heterogeneity, and microenvironmental factors underscores the complexity of tumor biology and highlights the dynamic nature of cancer metabolism. Moreover, these insights shed light on potential therapeutic vulnerabilities that can be targeted to disrupt tumor growth and improve treatment outcomes in GBM. By elucidating the intricate interplay between metabolic alterations and tumor microenvironmental cues, researchers aim to develop novel therapeutic strategies that exploit the vulnerabilities inherent in cancer metabolism while minimizing the impact on normal tissues [167,168]. In the subsequent sections, we will delve into the intricate mechanisms and consequential outcomes attributed to hypoxia and acidosis within the tumor microenvironment of GBM.

#### 2.4.1. Hypoxia

Hypoxia, characterized by low oxygen levels, is a well-established hallmark of tumor biology, particularly evident in GBM, one of the most hypoxic tumor types [169]. Among all the organs in the human body, the brain necessitates the highest oxygen supply. Despite constituting a mere 2% of total body weight, the brain consumes a staggering 20% of the body’s oxygen. Oxygen levels within the brain exhibit variability across different regions. For instance, the midbrain typically maintains an oxygen level of approximately 0.5%, whereas the pia, a delicate membrane covering the brain’s surface, experiences a higher oxygen concentration of around 8% [170]. In the context of brain tumors, research indicates distinctive oxygen environments. It has been observed that brain tumors harbor oxygen levels averaging 1.25%. Notably, the peritumoral area, the region surrounding the tumor, displays a slightly elevated oxygen concentration of about 2.5%. These variations in oxygen levels within the brain and tumor microenvironments have profound implications for tumor biology, including tumor growth, invasion, and response to therapy [171].

Within the GBM microenvironment, hypoxia exerts multifaceted effects on tumor biology and pathology, significantly impacting various aspects such as vascularization, invasiveness, drug resistance, and the modulation of anti-tumor immune responses. Importantly, hypoxia-driven processes such as invasion and angiogenesis are strongly associated with a poor prognosis in GBM patients. Additionally, hypoxia-mediated resistance to standard therapies, including chemotherapy and radiation, presents a significant challenge in the clinical management of GBM. GBM tumors exhibit distinctive characteristics, including necrotic foci, pseudopalisades, and microvascular hyperplasia. Pseudopalisades are believed to form as a consequence of tumor cells migrating away from hypoxic regions to establish the invasive leading edge. These migrating cells contribute to the formation of microvascular hyperplasia by organizing into tuft microaggregates surrounding blood vessels, ultimately leading to the development of glomeruloid bodies [172]. Such features are predominantly driven by hypoxia-induced angiogenesis. For instance, heightened expression of vascular endothelial growth factor (VEGF) stimulates excessive proliferation of endothelial cells, resulting in the formation of abnormal and leaky blood vessels that are prone to disruption [173]. This aberrant vasculature within the GBM microenvironment hampers the delivery of oxygen, drugs, and immune cells [174]. Furthermore, hypoxia arising from the dysfunctional vasculature promotes tumor cell invasion, which poses a significant challenge for GBM therapies.

Hypoxia plays a pivotal role in GBM progression, impacting various cellular processes through HIF-mediated transcriptional regulation. For instance, hypoxia upregulates the expression of carbonic anhydrase 9 (CA9), contributing to pH regulation in the tumor microenvironment [175]. Epithelial-to-mesenchymal transition (EMT) and metastasis-related genes, including recombination signal binding protein for immunoglobulin kappa J (RBPJ), zinc finger E-box-binding homeobox 1 (ZEB1), and Twist-related protein 1 (TWIST1), are also influenced by hypoxia via HIF-1α-dependent mechanisms [176,177,178]. Furthermore, hypoxia-induced activation of focal adhesion kinase (FAK) via integrin signaling facilitates GBM cell migration [179]. Procollagen-lysine 2-oxoglutarate 5-dioxygenase 2 (PLOD2), another hypoxia-regulated gene, promotes extracellular matrix remodeling and invasiveness in GBM cells [180]. Additionally, hypoxia stabilizes the epidermal growth factor receptor variant III (EGFRvIII) protein, enhancing its interaction with integrin β3 and promoting GBM cell invasiveness [181]. Moreover, hypoxia orchestrates adaptive responses in GBM cells, including modulation of protein translation and activation of stress response pathways. Hypoxia-induced translational suppression, mediated by protein kinase R (PKR)-like endoplasmic reticulum kinase (PERK) and the mechanistic target of rapamycin complex 1 (mTORC1), selectively regulates gene expression to favor cell survival under oxygen-deficient conditions [182,183]. Concomitantly, hypoxia triggers the unfolded protein response (UPR), involving activation of ATF4 and other UPR effectors, to maintain protein homeostasis [184]. Mitochondrial functions are also perturbed by hypoxia, affecting electron transport chain (ETC) activity and metabolic reprogramming in GBM cells [185].

The presence of an immunosuppressive microenvironment is a defining characteristic of GBM, presenting numerous interconnected barriers that hinder an effective anti-tumor immune response. Within the niche of glioma stem cells (GSCs), hypoxic and inflammatory responses intersect to shape the immune microenvironment, influencing both GSCs and infiltrating immune cells. Lymphocytes frequently encounter hypoxic conditions during their development and migration through peripheral tissues [185]. Stimulation of the T-cell receptor leads to the synthesis and stabilization of HIF-1α, particularly in hypoxic environments [186]. In vitro studies have shown that hypoxia can enhance the cytotoxic activity of CD8+ T lymphocytes and promote interferon-gamma secretion by CD4+ T cells [186,187]. However, oxygen deprivation also hampers the development, proliferation, and expression of inflammatory cytokines by cytotoxic T lymphocytes (CTLs) [186]. In contrast to differentiated tumor cells, GSCs have been observed to selectively inhibit the proliferation of activated T cells, although the precise mechanisms remain incompletely understood [55]. One potential mechanism involves the secretion of galectin-3, a molecule that binds to mature tumor-specific T cells within the tumor microenvironment, inducing apoptosis in these T cells [188]. Studies in galectin-3 knockout mice have shown enhanced tumor-free survival and increased effector T-cell functions, suggesting a role for galectin-3 in immune modulation within the GBM microenvironment. While the exact source of galectin-3 (tumor cells versus the microenvironment) remains unclear, its expression has been associated with hypoxic regions within the tumor [188]. Furthermore, the hypoxic tumor microenvironment of GBM plays a crucial role in orchestrating immunomodulation by recruiting immunosuppressive cells such as tumor-associated macrophages (TAMs), myeloid-derived suppressor cells (MDSCs), and regulatory T cells (Tregs). These cells exert their immunosuppressive effects through the secretion of factors and the expression of surface molecules that engage inhibitory receptors on effector immune cells. GSCs, commonly found in hypoxic niches within the tumor, interact closely with these immunosuppressive cells, influencing their recruitment and function [189,190].

Targeting hypoxia and HIF signaling pathways presents a promising therapeutic strategy for GBM. Ongoing clinical trials investigating the use of hypoxia-targeting agents, such as MBM-02 (Tempol) (NCT04874506) and belzutifan (NCT02974738), to inhibit HIFs offer hope for improved treatment outcomes in GBM patients. By understanding the intricate mechanisms underlying hypoxia-mediated tumor progression and therapy resistance, researchers can develop novel therapeutic interventions aimed at disrupting hypoxia signaling and enhancing the efficacy of existing treatment modalities in GBM.

#### 2.4.2. Acidosis

Tumor acidosis, characterized by an acidic microenvironment surrounding cancer cells, is a prominent feature of various malignancies, including GBM. This acidic milieu stems from metabolic adaptations and interactions within the TME, exerting significant effects on cellular processes and fostering an immunosuppressive environment that promotes treatment resistance. In GBM, metabolic reprogramming, particularly the Warburg effect, drives tumor acidosis by promoting aerobic glycolysis, leading to the accumulation of metabolic byproducts such as lactic acid and H+ ions. Magnetic resonance spectroscopic imaging of GBM confirms the acidic extracellular pH attributed to heightened lactic acid production [191,192,193,194].

Tumor acidosis in GBM plays a crucial role in inducing stem cell-like properties in non-stem cell tumors, facilitating an invasive phenotype marked by elevated expression of hypoxia-inducible factors (HIF)-1α and HIF-2α. Additionally, it influences cytoskeletal dynamics, cell adhesion, motility, and invasiveness while altering interactions between GBM cells and TME components, including endothelial cells, astrocytes, microglia/macrophages, and the extracellular matrix [195].

Cancer cells deploy protective mechanisms to counteract the acidic TME, enhancing the expression of proteins like LAMP2 to protect cellular membranes and upregulating autophagy-related and anti-apoptotic proteins like ATG5 and BCL-2, respectively, to ensure survival and promote aggressiveness [193,196]. Furthermore, tumor acidosis contributes to the immunosuppressive nature of the GBM microenvironment, impairing immune cell function and promoting tumor progression. It also provides a sanctuary for dormant tumor cells, supports circulating tumor cell survival and invasion, and enhances resistance to chemotherapy and radiotherapy [197]. Recent studies underscore the metabolic plasticity of GBM cells, revealing subsets with both glycolytic and oxidative phosphorylation phenotypes, suggesting potential therapeutic vulnerabilities. Targeting tumor acidosis, alongside standard-of-care therapy, emerges as a promising strategy, with ongoing clinical trials investigating the efficacy of such approaches [198].

Preclinical studies suggest that temozolomide-induced up-regulation of carbonic anhydrase (CA), mediated by the proto-oncogene BCL-3, contributes to therapy resistance in GBM cells. Conversely, acetazolamide (ACZ), a CA inhibitor, sensitizes patient-derived GBM cells to temozolomide. A Phase I clinical trial evaluated ACZ in adjuvant temozolomide treatment for newly diagnosed MGMT-methylated malignant glioma. Given the acidic microenvironment prevalent in GBM, characterized by elevated lactic acid production and alterations in energy utilization, the addition of ACZ to adjuvant temozolomide treatment presents a novel approach. Twenty-four patients (23 GBM, 1 Grade III IDH-mutant astrocytoma) participated, receiving ACZ from days 1 to 21 of each temozolomide cycle. No regimen-limiting toxicity was observed, and adverse events were minimal. Median progression-free survival (PFS) was 18.8 months, and median overall survival (OS) was 25.0 months, with a 2-year OS rate of 68.2%. Secondary endpoint analysis, including the 6-month objective response rate (ORR) and BCL-3 biomarker assessment, is pending. Initial findings suggest ACZ addition is well-tolerated and may improve PFS and 2-year OS. These results advocate for further investigation in randomized, placebo-controlled trials [199].

These results highlight the importance of considering tumor acidosis in therapeutic strategies for GBM and advocate for continued investigation in randomized, placebo-controlled trials to validate these observations. Table 4 illustrates the interplay of hypoxia and acidosis in GBM with its key aspects and implications 

### 2.5. Glioblatoma Stem Cells

Glioblastoma stem cells (GSCs) represent pivotal entities within the heterogeneous landscape of GBM, predominantly localized within specialized microenvironments termed perivascular and hypoxic niches. Their interaction with the TME profoundly influences disease progression and recurrence dynamics. The precise spatial distribution of GSCs within the GBM tumor microenvironment governs their biological behavior and aggressiveness. Notably, GSCs dwelling in peritumoral regions exhibit relatively subdued aggressiveness compared to their counterparts situated within the tumor core [200]. However, they manifest augmented resistance to standard therapeutic modalities such as temozolomide (TMZ) and radiation therapy (RT), highlighting the therapeutic challenges posed by GSCs [201,202].

Distinguishing GSC populations via biomarkers allows for targeted interventions, shedding light on the pathophysiological processes underlying cerebral nervous system tumors. In recent years, significant progress has been made in elucidating various putative GSC markers, including CD133 (PROM-1), Nestin, LGR5, B23 (NPM1), and GPD1. These markers, when functionally correlated with GSC maintenance, offer valuable insights into GBM progression and treatment [202].

Direct targeting of GSCs is essential for developing effective therapeutic strategies for GBM. Understanding and blocking key signaling pathways and receptors associated with GSC maintenance and oncogenesis is crucial for advancing targeted therapies. Notable pathways include Notch, Wnt, Sonic Hedgehog (SHH), and Receptor Tyrosine Kinase (RTK) pathways, each playing a significant role in GSC proliferation, maintenance, and resistance to standard treatments like chemotherapy and radiotherapy [202]. Inhibiting the Notch signaling pathway, which regulates cell fate determination and differentiation, is a promising strategy [203]. Inhibitors such as γ-secretase inhibitors (GSIs), such as RO4929097, have shown efficacy in blocking Notch signaling both in vitro and in vivo, offering potential therapeutic benefits [204]. The SHH signaling pathway, critical for normal neural stem cell fate determination and maintenance, also contributes to GSC oncogenesis and self-renewal. Inhibiting this pathway could disrupt GSC properties and enhance sensitivity to chemotherapy, particularly TMZ [205,206,207,208]. Receptor Tyrosine Kinase (RTK) pathways, including Epidermal Growth Factor Receptor (EGFR), Platelet-Derived Growth Factor Receptor (PDGFR), and Vascular Endothelial Growth Factor Receptor (VEGFR), are implicated in GSC expansion, metastasis, and survival. Targeting these receptors with inhibitors presents opportunities for therapeutic intervention [202].

Promoting GSC differentiation represents another avenue for therapeutic intervention. Agents such as bone morphogenic proteins (BMPs), microRNAs (miRNAs), and graphene oxide (GO) have demonstrated the ability to induce GSC differentiation, sensitizing tumors to chemotherapy and reducing tumorigenicity [209,210,211]. Virotherapy, particularly oncolytic virotherapy (OV), holds promise as a therapeutic approach [212]. Oncolytic viruses can selectively replicate within tumor cells, leading to their lysis. Strategies involving stem cell-mediated delivery of oncolytic viruses have shown potential for targeting GSCs and preventing GBM recurrence [213]. Indirect targeting of GSCs involves addressing the microenvironmental niches that support their survival and proliferation, including the perivascular and hypoxic niches. These approaches aim to disrupt the supportive microenvironmental cues essential for GSC survival and proliferation, enhance the efficacy of existing therapies, and improve patient outcomes [202].

Understanding the molecular underpinnings governing GSC-mediated immunosuppression and tumor progression opens novel avenues for therapeutic exploration, potentially enhancing clinical outcomes for GBM patients. Targeted immunotherapeutic interventions aimed at disrupting these intricate interactions hold promise for curtailing tumor growth and metastatic dissemination.

### 2.6. Deciphering Glioblastoma Complexity: Insights from Organ-on-Chip Modeling

GBM poses a formidable challenge in oncology, marked by limited therapeutic progress and persistently poor patient outcomes despite extensive research endeavors. A crucial contributing factor to treatment resistance and unfavorable prognosis in GBM stems from its profound heterogeneity, which manifests both between tumors (inter-tumor heterogeneity) and within individual tumors (intra-tumor heterogeneity). This intricate landscape presents significant obstacles to formulating effective therapeutic approaches, as interventions targeting specific molecular pathways or cellular subtypes may only exert efficacy against a subset of tumor cells, leaving others unaffected [2].

GBM is intricately intertwined with its TME. GBM cells exhibit the capability to release cell signaling molecules that modulate the TME, thereby fostering tumor angiogenesis and inducing immune tolerance. Conversely, immune cells within the TME exert influence over GBM cell growth and development. Furthermore, components within the non-tumor elements of the TME play a crucial role in facilitating GBM cell proliferation and invasion. The presence of a GBM TME augments the capacity for GBM cell proliferation, migration, and immune evasion, thereby promoting GBM progression. A correlation exists between the genetic makeup of tumors and the complexity of their surrounding microenvironment [9,10].

Given the unsatisfactory outcomes of current GBM treatments, there is a pressing imperative for thorough investigation into the mechanisms governing GBM development within the TME, with particular emphasis on elucidating the interplay between TME complexity and tumor genetics. Such endeavors are essential for identifying novel therapeutic targets and devising innovative treatment strategies for GBM. Recognizing the urgent need to decipher the drivers of GBM heterogeneity and their implications for tumor behavior and treatment response, there has been a surge in the utilization of advanced in vitro models that replicate the TME. These models aim to capture the intricate interplay between tumor cells, stromal cells, immune cells, and the extracellular matrix (ECM), all of which contribute to GBM progression, invasion, and therapy resistance [214,215].

Organ-on-a-chip (OOC) technology has emerged as a potent tool among various in vitro model systems, offering the capability to replicate crucial aspects of tissue architecture, function, and microenvironmental cues in a controlled laboratory setting. Leveraging OOC platforms provides several advantages over conventional cell culture methods. These include the ability to simulate dynamic physiological conditions, integrate multiple cell types in spatially defined arrangements, and monitor cellular responses in real-time. Such attributes make OOC models particularly well-suited for investigating complex diseases like GBM, where the intricate interplay between diverse cell populations and microenvironmental factors profoundly influences disease progression and treatment outcomes. The transformative nature of OOC technology lies in its ability to replicate vital organ functions and microenvironments on a chip, drawing from principles of microfluidics, cell biology, and tissue engineering. Initially pioneered by Huh et al., the development of a lung-on-a-chip marked the inception of OOC, mimicking the alveolar capillary interface of the human lung for toxicological investigations. This seminal work underscored the immense potential of OOC in biomedical research, leading to the creation of various OOC platforms, including heart-on-a-chip, gut-on-a-chip, liver-on-a-chip, and more, with some combining multiple organ models to simulate intricate organ interactions [216,217,218].

OOC devices offer numerous advantages, such as miniaturization, reduced reagent consumption, and precise control over the cellular microenvironment. They excel at recapitulating human cellular microenvironments and interactions, surpassing conventional two-dimensional cell culture methods. Unlike animal models, OOCs mitigate ethical concerns and species variations, making them promising for human disease studies, drug development, and personalized medicine. In recent years, OOC models have gained momentum in elucidating the dynamics of the GBM TME and evaluating novel therapeutic strategies. Microfluidics-based OOC platforms have been engineered to replicate critical features of GBM tumors, such as vascularization, hypoxia, and immune cell infiltration. Simultaneously, advancements in 3D bioprinting techniques have facilitated the development of sophisticated multicellular models that closely mimic the structural and genetic characteristics of patient tumors [219].

Extrinsic and intrinsic targets within the complex landscape of GBM biology stand as crucial focal points for advanced tumor-on-chip models. Extrinsic targets encapsulate the dynamic elements of the tumor microenvironment, encompassing a spectrum ranging from extracellular matrix (ECM) components to immune cells like macrophages and T cells, as well as stromal cells such as fibroblasts and endothelial cells. Additionally, cytokines, chemokines, angiogenic factors, and the influence of hypoxic conditions and acidic pH levels contribute to the intricate milieu. Furthermore, the involvement of extracellular vesicles, growth factors like VEGF and EGF, and matrix metalloproteinases (MMPs) further enriches this environment. Intrinsic targets delve into the intricate web of intra- and intercellular interactions and functions fundamental to GBM progression. These encompass pivotal pathways such as oncogenic signaling (e.g., EGFR, PI3K/AKT, and MAPK), tumor suppressor genes (e.g., p53 and PTEN), DNA repair mechanisms, and epigenetic modifications, including DNA methylation and histone modifications. Moreover, the orchestration of cell cycle regulators (e.g., cyclins and CDKs), apoptotic pathways, cellular metabolism, and the presence of cancer stem cells (CSCs) significantly influence disease dynamics. Additionally, factors such as drug efflux pumps (e.g., ABC transporters) and immune checkpoint molecules (e.g., PD-1 and CTLA-4) play critical roles in therapeutic response and resistance. Integrating these multifaceted targets into GBM-on-chip models offers a sophisticated platform for in-depth investigations into tumor progression, drug efficacy, and the mechanisms underlying therapeutic resistance. This approach not only enhances our understanding of GBM biology but also paves the way for the development of more effective treatment strategies tailored to combat this aggressive form of brain cancer (Figure 4).

## 3. GBM-on-Chip Models

### 3.1. Microfluidic GBM-on-Chip Models

Microfluidic methods offer versatile approaches for developing organ-on-chip (OOC) models, enabling the recreation of complex physiological microenvironments and interactions between different cell types. These methods facilitate the construction of miniature systems that mimic the structure and function of human organs, providing valuable platforms for studying disease mechanisms, drug responses, and tissue regeneration [220]. The step-by-step process for preparing a glioblastoma tumor microenvironment is illustrated using traditional microfluidic chips (Figure 5).

One commonly utilized microfluidic method is soft lithography, particularly replica molding, which is widely employed for fabricating microfluidic devices due to its simplicity and versatility. This technique involves creating a master mold using photolithography and replicating the mold with elastomers such as polydimethylsiloxane (PDMS). By patterning microchannels and chambers on PDMS substrates, researchers can construct OOC devices with precise control over fluid flow and cell culture conditions [221].

Another method, micromolding in capillaries (MIMIC), utilizes capillary action to fabricate microfluidic channels within a hydrogel matrix. By injecting liquid precursor solutions into a capillary tube and allowing them to polymerize, researchers can create intricate networks of microchannels embedded within a 3D hydrogel scaffold. This approach is particularly suitable for constructing OOC models with perfusable vascular networks and multicellular architectures [222]. Additionally, 3D bioprinting enables the precise deposition of biomaterials, cells, and growth factors to create complex 3D tissue constructs. By integrating microfluidic components into bioprinting systems, researchers can fabricate OOC models with spatially defined cell distributions and heterogeneous tissue architectures. This approach allows for the generation of personalized OOC platforms tailored to specific disease states or patient-derived samples.

Droplet-based microfluidic platforms offer unique capabilities for high-throughput cell encapsulation, culture, and analysis. By generating monodisperse droplets containing cells, hydrogels, and biochemical cues, researchers can create microscale environments conducive to cell growth, differentiation, and interaction. These droplet-based systems enable the creation of OOC models with precise control over cellular microenvironments and the ability to perform multiplexed assays and screenings [223].

Furthermore, integrating organoid cultures with microfluidic devices allows for the development of more physiologically relevant OOC models. Organoids, self-organizing 3D cell aggregates derived from stem cells or patient samples, can be incorporated into microfluidic systems to recreate organ-specific architectures and functions. This approach enables the study of organoid behavior under dynamic flow conditions, as well as the investigation of intercellular interactions and responses to external stimuli [224].

Microfluidic methods play a pivotal role in advancing the study of GBM through the development of organ-on-chip (OOC) models. These models serve as innovative platforms for replicating the intricate microenvironment of brain tumors and investigating crucial aspects of GBM biology, including disease progression, drug responses, and potential therapeutic strategies (Figure 3). The utilization of microfluidic techniques in GBM OOC models enables researchers to achieve a higher level of physiological relevance and control, leading to more accurate and insightful findings [225]. One of the primary microfluidic methods employed in GBM OOC models is the development of tumor-on-a-chip platforms. These microfluidic devices are meticulously designed to mimic the complex tumor microenvironment by incorporating features such as 3D culture scaffolds, perfusable vascular networks, and gradients of oxygen and nutrients. This allows for the co-culture of GBM cells with other cell types found in the tumor microenvironment, such as endothelial cells, immune cells, and stromal cells. Through these platforms, researchers can delve into the intricate intercellular interactions and tumor-stroma crosstalk that contribute to GBM progression and therapeutic resistance. Additionally, microfluidic BBB models are integrated into GBM OOC models to replicate the unique vascular properties of the brain and study drug transport dynamics across the blood–brain barrier (BBB). By culturing brain endothelial cells within microfluidic devices with integrated perfusable channels, researchers can investigate the impact of BBB dysfunction on GBM progression and evaluate novel drug delivery strategies for targeted therapy. These models provide valuable insights into the challenges of delivering therapeutic agents to GBM tumors and facilitate the development of more effective treatment approaches [226]. Microfluidic Chemotaxis Assays are another essential component of GBM OOC models, enabling researchers to study the chemotactic migration of GBM cells in response to soluble factors present in the tumor microenvironment. By precisely controlling the spatial distribution of chemoattractants such as growth factors, cytokines, and chemokines within gradient-generating channels, researchers can elucidate the underlying mechanisms driving GBM cell invasion and metastasis. Moreover, these assays serve as valuable platforms for screening potential anti-migratory drugs and identifying novel therapeutic targets for GBM treatment.

Furthermore, organotypic brain slice cultures in microfluidic chambers provide a physiologically relevant ex vivo model for studying GBM tumor behavior within a controlled microenvironment. By culturing organotypic brain slices derived from GBM patient samples or animal models, researchers can maintain the viability and functionality of brain tissue over extended periods of time. Integration of microfluidic perfusion systems enables continuous monitoring of GBM cell interactions with surrounding brain tissue, offering insights into tumor invasion and progression mechanisms. Microfluidic Drug Screening Platforms represent another critical application of microfluidic methods in GBM research, facilitating high-throughput screening of potential therapeutics. By culturing GBM cells in microfluidic chambers and exposing them to libraries of small molecules or therapeutic agents, researchers can assess drug efficacy, toxicity, and pharmacokinetics in a more physiologically relevant context. These platforms expedite the identification of promising drug candidates for further preclinical evaluation, ultimately accelerating the development of novel treatment strategies for GBM [227,228].

Numerous studies have been conducted utilizing microfluidic chips in the context of glioblastoma research. These studies leverage the precise control and manipulation of fluids at the microscale to recreate the intricate microenvironment of glioblastoma tumors. 

Shi and colleagues, in 2023, revealed the critical need for preclinical models that effectively incorporate the complexity of the tumor microenvironment and the BBB structure and function for the treatment of glioma. As most anti-glioma drug candidates struggle to permeate the BBB, the development of such models is urgently required. In response to this need, the researchers constructed an in vitro BBB–glioma microfluidic chip model. This model featured primary human brain microvascular endothelial cells, pericytes, astrocytes, and glioma cells, recapitulating the high barrier function characteristic of the human BBB and the glioma microenvironment. The BBB unit within the BBB–glioma microfluidic chip (referred to as the BBB–U251 chip) demonstrated selective permeability to fluorescein isothiocyanate isomer-dextran (FITC-dextran) with varying molecular weights, as well as to three model drugs exhibiting different permeability behaviors across the BBB. These findings affirmed the functionality of the barrier within the glioma model. Subsequently, six potential anti-glioma components sourced from traditional Chinese medicine were introduced into the blood channel of the chip, and their permeation was quantified using high-performance liquid chromatography combined with ultraviolet detection (HPLC-UV). The permeated drugs were then delivered directly to 3D-cultured glioma cells (U251) within the chip to assess drug efficacy. Notably, the permeability coefficients of the drugs closely mirrored in vivo data obtained from traditional Transwell models. Moreover, the efficacy of the drugs on U251 cells within the BBB-U251 chip was significantly attenuated due to the presence of the BBB. These findings underscored the importance of considering the BBB when developing new anti-glioma drugs [229].

In a recent study conducted by Bae and colleagues, the phenomenon of vessel co-option (VC) was explored, highlighting its distinction from angiogenesis, wherein tumor cells grow towards existing blood vessels. VC allows tumor cells to access comparatively higher levels of nutrients and oxygen from these vessels. Despite its clinical significance, VC has been relatively understudied compared to angiogenesis due to challenges associated with longitudinally observing VC in vivo and the absence of suitable VC models in vitro. Traditionally, a needle template method has been employed to create blood vessels and simulate angiogenesis by forming microchannels in hydrogel using needles. However, this method had not been previously applied to mimic VC. The study presented the development of VC on a chip utilizing the needle template method. Through this innovative approach, the researchers investigated the effect of the distance between spheroids and blood vessels on VC induction. GBM spheroids were seeded at distances of 50 and 250 μm from preformed blood vessels on the VC chip. Remarkably, irrespective of the distance, cancer cells from the spheroids exhibited a tendency to grow towards the blood vessels without penetrating them, indicative of VC-like behavior in GBM cells. These findings underscore the potential of the chip model to recapitulate VC in GBM, offering a promising platform for further research into the dynamics of VC and its implications for tumor microenvironment interactions. The study contributes to advancing our understanding of VC mechanisms and may have implications for the development of therapeutic strategies targeting VC in GBM [230].

In a study conducted by Hosni and their research team in 2023, they introduced a straightforward and robust microfluidic device capable of maintaining GBM spheroids (U87 cells) for a minimum duration of 7 days. Through RNA sequencing analysis, it was demonstrated that spheroids cultured within the microfluidic environment exhibited heightened proliferative activity compared to those cultivated in static plate cultures. Additionally, these spheroids downregulate genes associated with cell adhesion, thereby potentially offering valuable insights into the metastatic process. The study also included a comparative analysis of siRNA-mediated gene knockdown targeting PRMT2 and RAB21 genes across different culture systems, including 2D monolayer cultured cells, static spheroid cultures, and spheroids sustained within the microfluidic device. The results, as measured by quantitative PCR, revealed a significant reduction in gene expression across all culture platforms. While the most efficient knockdown was observed in cells grown in 2D monolayer culture followed by static spheroid culture, the microfluidic device still achieved an impressive approximately 40% knockdown efficiency [231].

Dou et al. (2020) focused on the development of a sophisticated microfluidic chip to generate a controllable stiffness gradient and orthogonal chemical stimulation, aimed at investigating the behaviors of glioma cells. The chip integrated fibronectin-conjugated polyacrylamide (PAA) hydrogel within the cell culture chamber, exhibiting a longitudinal stiffness gradient ranging from approximately 1 kPa to 40 kPa. Additionally, lateral diffusion-based chemical stimulation was induced through circumambient microchannel arrays. The study focused on examining the synergistic effect of epidermal growth factor (EGF) stimulation and hydrogel stiffness gradient on U87-MG cell migration. Through cell migration tracing, it was observed that hydrogel stiffness could promote cell chemotaxis, while the EGF gradient could accelerate cell migration. Moreover, analysis of cell morphology revealed typical cell spreading, increased aspect ratios, and decreased circularity in response to a stiffer substrate, with these effects plateauing at a certain stiffness level. Furthermore, the study observed that the content of intracellular reactive oxygen species (ROS) on the hydrogel’s soft end was enhanced by approximately 2-fold compared to that on the hydrogel’s stiff end. This finding suggests a potential link between substrate stiffness and intracellular ROS levels, indicating a role for mechanical cues in cellular responses. The enhancement of substrate stiffness on cell chemotaxis demonstrated in this study holds significant implications for in vitro model simulation and tissue engineering applications. By providing insights into the interplay between mechanical and biochemical cues in glioma cell behavior, this research contributes to a deeper understanding of tumor progression mechanisms and may inform the development of novel therapeutic strategies targeting the tumor microenvironment [232].

Olubajo et al. (2020) assembled a device, fabricated using a photolithographic process, consisting of two layers of glass bonded together to enclose a tissue chamber and a network of microchannels enabling continued tissue perfusion. The results of their study revealed that a total of 128 tissue biopsies from 33 patients were sustained in microfluidic devices for an average duration of 72 h. The analysis indicated that tissue viability, measured with Annexin V and propidium iodide, was 61.1% in tissue maintained on-chip, compared with 68.9% for fresh tissue analyzed at the beginning of the experiments. Additional biomarkers, including lactate dehydrogenase absorbance and trypan blue exclusion, corroborated the viability of the tissue maintained on-chip. Histological appearances remained unchanged during the tissue maintenance period, and immunohistochemical analysis of Ki67 and caspase 3 showed no significant differences compared with fresh tissues. A trend observed indicated that tumors associated with poorer outcomes, such as recurrent tumors and Isocitrate Dehydrogenase (IDH) wildtype, displayed higher viability on-chip than tumors linked with improved outcomes, such as low-grade gliomas, IDH mutants, and primary tumors [233].

In the study conducted by Liu et al. (2017), a three-dimensional tumor-microvascular structure was simulated on a microfluidic chip to investigate the effects of antioxidants on malignant glioma cells in vitro. The research involved the construction of a 3D hydrogel containing a lumen, facilitating the co-culture of endothelial cells and glioma cells to mimic the tumor microvascular environment. A macroporous gelatin transglutaminase (TG) hydrogel was prepared with biological and mechanical properties suitable for cell culture and nutrient refreshment. To recreate a vessel structure, U87 cells were dispersed in the TG-gelatin hydrogel, while HUVEC cells were seeded in the lumen of the hydrogel. Three typical antioxidants—α-lipoic acid, catechins, and ascorbic acid—were selected to investigate their effects on glioma cells in the simulated tumor microenvironment. The results demonstrated that the HUVEC cells forming the vessel exhibited transportation and penetrable functions for antioxidants from the lumen to the glioma cells. Furthermore, the antioxidants displayed higher selectivity towards U87 cells compared to HUVEC cells, with α-lipoic acid exhibiting particularly strong antioxidant capacity. These findings offer valuable insights into the potential therapeutic effects of antioxidants in the context of glioma treatment within a simulated tumor microenvironment [234].

Further, in a study by Fan and colleagues, a novel three-dimensional (3D) brain cancer chip was developed, composed of photo-polymerizable poly(ethylene) glycol diacrylate (PEGDA) hydrogel, aimed at facilitating drug screening for brain cancer. Unlike microfluidic devices made of poly(dimethylsiloxane) (PDMS), this chip could be produced within seconds of photolithography, eliminating the need for silicon wafers, replica molding, and plasma bonding. The researchers cultured glioblastoma cells (U87) on the chip, which formed 3D brain cancer tissues. Utilizing this GBM chip, they conducted combinatorial treatments with pitavastatin and irinotecan. The results indicated that the chip enabled high-throughput formation of GBM cancer spheroids, simultaneous administration of multiple drugs, and parallel testing of drug responses on a large scale [235].

Han and colleagues devised a microfluidic chip comprising 488 hexagonal chambers connected with two microchannels, designed for convenient long-term cell culture with daily resupply of doxorubicin (DOX) and culture media, aimed at elucidating the mechanisms underlying drug-resistant GBM cells. Within these chambers, U87 cells were uniformly distributed, exhibiting a notable decrease in cell numbers near the DOX channel by day 3 post-treatment. By day 5, approximately three-quarters of the chambers were empty, indicative of cellular depletion, followed by cell repopulation observed by day 7, signifying the emergence of resistance and the migration of resistant cells to regions with higher DOX concentrations. Molecular analysis conducted on the cells after 7-day growth on the chip, employing exome sequencing and transcriptome sequencing techniques, unveiled significant findings. Notably, mutations in the CHD1 (chromodomain helicase DNA binding protein 1) gene, a known target of epirubicin (the 4′-epi-isomer of DOX), and the filamin-A gene were identified. The loss of filamin-A was found to promote DOX resistance by regulating the influx and efflux of topoisomerase II poisons. Moreover, overexpression of aldo-keto reductases (AKRs), specifically AKR1B1, AKR1C1, AKR1C2, and AKR1C3, which convert DOX to doxorubicinol, resulted in substantially reduced cytotoxicity, diminished DNA-binding activity, localization to extranuclear lysosomes, and activation of NF-κB through regulation of NOD1 (nucleotide-binding oligomerization domain containing 1) activity mediated by three mutated genes—CARD6 (caspase activation and recruitment domain family, member 6), NSD1 (nuclear receptor binding SET domain protein 1), and NLRP13 (NOD-like receptor family, pyrin domain containing 13). This cascade of events led to the overexpression of chemokines and proinflammatory cytokines [236].

Shao and colleagues conducted a study investigating a novel microfluidic chip-based approach for analyzing exosomal mRNA associated with drug resistance in glioblastoma. Monitoring drug efficacy in GBM presents a significant clinical challenge due to the impracticality of serial re-biopsy of primary tumors. Key enzymes like MGMT (O6-methylguanine DNA methyltransferase) and APNG (alkylpurine-DNA-N-glycosylase) are pivotal in repairing temozolomide-induced DNA damage, with their tissue levels inversely linked to treatment effectiveness. However, current clinical analysis methods rely heavily on promoter methylation studies of tumor biopsy material collected during initial surgery, posing limitations in real-time assessment. In response, the researchers developed a microfluidic chip tailored to analyze mRNA levels of MGMT and APNG in enriched tumor exosomes isolated from blood samples. Results indicated a strong correlation between the exosomal mRNA levels of these enzymes and their counterparts in parental cells. Notably, fluctuations in these levels were observed during the treatment course of seven patients. These findings suggest that the proposed method, subject to validation on a larger patient cohort, could offer a promising avenue for predicting drug responses in GBM patients. This approach presents a non-invasive alternative for assessing treatment efficacy, potentially enhancing clinical management strategies for this challenging disease [237].

Overall, the integration of microfluidic technologies with GBM research holds tremendous promise for advancing our understanding of this devastating disease and developing more effective therapeutic interventions. By recapitulating key features of the GBM microenvironment in vitro, microfluidic OOC models provide valuable tools for translational research and personalized medicine approaches aimed at improving patient outcomes in the fight against GBM.

### 3.2. Bioprinting GBM-on-a-Chip

In recent years, there has been a discernible upsurge in the utilization of bioprinting methodologies for the fabrication of intricate GBM models. These models serve as a platform for mimicking the complex tumor TME in a controlled in vitro milieu. Through the simultaneous deposition of diverse cell types and ECM mimetic materials onto substrates compatible with cellular growth, bioprinting facilitates the generation of GBM-on-a-chip models capable of replicating the heterogeneous TME, including the establishment of vascular networks. The step-by-step procedure of 3D bioprinting was illustrated. Bioprinting simulates the tumor microenvironment typical of glioblastoma (Figure 6).

Researchers have demonstrated the feasibility of fabricating GBM-on-a-chip models using bioprinting techniques, showcasing their potential for elucidating the biological mechanisms underlying GBM and for conducting preclinical investigations of therapeutic interventions. Tang and colleagues employed extrusion-based bioprinting to create a GBM tumor encapsulated within a hydrogel system containing macrophages, thereby establishing a bionic GBM TME for investigating the impact of infiltrating immune cells on GBM cell behavior and drug responses. Their study focused on recurrent glioblastoma, where macrophages/microglia play a significant role in tumor progression. To elucidate the function of macrophages within a 3D context, the researchers compared the growth patterns of GSCs alone or in combination with astrocytes and neural precursor cells within a hyaluronic acid-rich hydrogel, with or without the presence of macrophages. The bioprinted constructs incorporating macrophages successfully recapitulated patient-derived transcriptional profiles, providing insights into factors predictive of patient survival, stemness maintenance, invasion potential, and drug resistance. Moreover, employing whole-genome CRISPR screening with these bioprinted complex systems allowed for the identification of unique molecular dependencies in GSCs, particularly when compared to traditional sphere culture methods. These multicellular bioprinted models offer a scalable and physiologically relevant platform for investigating various aspects of glioblastoma, including drug sensitivity, cellular crosstalk, invasion mechanisms, context-specific functional dependencies, and immunological interactions within a species-matched neural environment [238]. In a study conducted by the same research group in 2021, a rapid 3D bioprinting technique was utilized to fabricate a GBM model replicating native biophysical heterogeneity. The research focused on developing biomimetic tri-regional GBM models, comprising tumor regions, acellular ECM regions, and an endothelial region with stiffness patterns corresponding to the GBM stroma, pathological or normal brain parenchyma, and brain capillaries, respectively. These models employed patient-derived GBM cells, human endothelial cells, and hyaluronic acid derivatives to establish a species-matched and biochemically relevant microenvironment. The study revealed that biophysical cues play pivotal roles in various tumor cell behaviors and angiogenic potentials, influencing different molecular subtypes of GBM. The stiffer models were observed to be enriched in the mesenchymal subtype, exhibiting diffuse invasion of tumor cells and inducing protruding angiogenesis, resulting in higher drug resistance to temozolomide. Conversely, the softer models exhibited enrichment in the classical subtype and supported expansive cell growth. The three-dimensional bioprinting technology utilized in this study facilitated rapid, flexible, and reproducible patient-specific GBM modeling with biophysical heterogeneity. This approach provides a tunable system for future studies to effectively interrogate GBM disease mechanisms and screen drug compounds [239]. 

Yi et al. (2019) developed a bioprinted human glioblastoma-on-a-chip for the identification of patient-specific responses to chemoradiotherapy. The aim was to create patient-specific ex vivo models of human tumors that accurately mimic the pathological characteristics and complex ecology of native tumors, thereby aiding in determining the most suitable cancer treatment for individual patients. The researchers demonstrated that bioprinted reconstituted glioblastoma tumors, comprising patient-derived tumor cells, vascular endothelial cells, and decellularized extracellular matrix from brain tissue, arranged in a compartmentalized cancer-stroma concentric-ring structure sustaining a radial oxygen gradient, successfully recapitulated the structural, biochemical, and biophysical properties of native tumors. Furthermore, the study revealed that the glioblastoma-on-a-chip model replicated clinically observed patient-specific resistances to treatment with concurrent chemoradiation and temozolomide. Importantly, the model enabled the identification of drug combinations associated with enhanced tumor killing. The patient-specific tumor-on-a-chip model holds promise for identifying effective treatments for glioblastoma patients resistant to standard first-line treatment [25].

Additionally, 3D bioprinting was employed to develop a vascularized GBM-on-a-chip model that replicated the pathophysiological conditions of GBM tumors and their surrounding vascular microenvironment, providing insights into the mechanical regulation of GBM by gravitational forces. According to Silvani et al. (2021), an innovative hybrid in vitro vascularized GBM-on-a-chip model is presented as a strategic integration of microfluidics and 3D bioprinting technologies. The system can recreate the compartmentalized brain tumor microenvironment, comprising the functional blood–brain barrier (BBB) and the adjacent 3D perivascular tumor niche, by selectively mimicking physiological shear stress and cell–cell, cell–matrix mechanical interactions. The GBM-on-a-chip model was evaluated under simulated microgravity (µG) conditions as a form of mechanical unloading, showing a significant cellular morphological and mechanotransduction response, thereby indicating that gravitational forces play an important role in glioblastoma mechanical regulation. The proposed GBM-on-a-chip represents a meaningful biological tool for further research in cancer mechanobiology and pre-clinical approaches to brain tumor therapy [240].

Heinrich et al. investigated the pivotal role of GBM-associated macrophages (GAMs) in the progression and invasiveness of GBM, a phenomenon not fully elucidated. Moreover, the absence of relevant in vitro models to emulate their interactions prompted the development of novel 3D-bioprinted mini-brains comprising glioblastoma cells and macrophages. This model serves as a tool to explore the interactions between these cell types and to evaluate therapeutics targeting this interaction. The study demonstrated that within the mini-brains, glioblastoma cells actively recruit macrophages and polarize them into a GAM-specific phenotype, corroborating clinical relevance to transcriptomic and patient survival data. Additionally, it was observed that macrophages induce glioblastoma cell progression and invasiveness within the mini-brains. Furthermore, the efficacy of therapeutics targeting the GAM-tumor cell interaction was assessed, revealing reduced tumor growth and increased sensitivity to chemotherapy upon inhibition of this interaction. The study envisions the utilization of this 3D-bioprinted tumor model to enhance the comprehension of tumor biology and for the evaluation of novel cancer therapeutics [241].

Furthermore, the emergence of patient-specific GBM-on-a-chip models holds promise for personalized medicine approaches in GBM treatment. These models, generated using patient-derived GBM cells and compatible bioinks, offer a representation of the GBM TME compared to conventional GBM cell lines. For instance, researchers have successfully bioprinted GBM-on-a-chip models using patient-derived cells, demonstrating their utility in evaluating tumor-killing drug candidates and screening for effective therapies, particularly for GBM patients resistant to standard drug regimens. Despite these advancements, challenges persist in developing bioinks that accurately recapitulate the complexity of the GBM TME. Nevertheless, ongoing efforts in bioink formulation hold the potential to refine GBM-on-a-chip models further, ultimately reducing reliance on animal models for preclinical studies. Neufeld et al. utilized 3D bioprinting to create a perfusable GBM model capable of replicating various in vivo features of GBM, including growth kinetics, invasiveness, and genetic characteristics, for the purpose of drug response testing. The heterogenic tumor microenvironment (TME) was emulated using a fibrin-based GBM bioink containing patient-derived GBM cells, astrocytes, and microglia. Furthermore, perfusable blood vessels were simulated employing a sacrificial bioink coated with brain pericytes and human umbilical vein endothelial cells (HUVECs). These 3D bioprinted models showcase the promising potential of advanced biomanufacturing techniques in GBM research. Compared to GBM cell lines such as U87, which have been criticized for their inability to accurately represent the genetic and molecular characteristics of GBM in patients, GBM models bioprinted using patient-derived cells offer increased credibility and personalization [242].

Amereh et al. (2023) explored the influence of matrix stiffness on the invasion behavior of human glioblastoma tumoroids, considering the well-acknowledged correlation between ECM stiffness and GBM tumor progression. A 3D-printed tumor-on-a-chip model was employed to culture human glioblastoma tumoroids, facilitating the assessment of stiffness effects on tumor progression. To induce variations in collagen matrix stiffness, different concentrations of collagenase were incorporated, leading to an inhomogeneous collagen concentration. To delve into the underlying mechanisms governing GBM invasion, an in-silico hybrid mathematical model was utilized to predict tumor evolution within an inhomogeneous environment, enabling the study of multiple dynamically interacting variables. This model comprised a continuum reaction–diffusion model for tumoroid growth and a discrete model for single-cell migration into the surrounding tissue. Results delineated two distinct invasion patterns of tumoroids in response to collagenase concentration: ring-type and finger-type patterns. Furthermore, higher collagenase concentrations correlated with increased invasion lengths, affirming the significant influence of matrix stiffness on tumor behavior. The congruence between experimental findings and model predictions underscores the efficacy of this approach in investigating the impact of diverse extracellular matrix characteristics on tumor growth and invasion [243].

Overall, biosensor technology holds great promise for advancing our understanding of GBM biology, improving early diagnosis, guiding personalized treatment strategies, and monitoring therapeutic responses. The fabrication techniques for GBM-on-chip models are systematically analyzed with respect to their advancements and limitations in Table 5. The platforms utilized for generating glioblastoma (GBM)-on-chip models are systematically compared, delineating their characteristics, advantages, and disadvantages in Table 6. Further research and development in this field is warranted to harness the full potential of biosensors in revolutionizing GBM diagnosis and management, ultimately leading to improved patient outcomes.

## 4. Biosensor Technology in Glioblastoma Research

Biosensor technology represents a vital tool in the field of GBM research, offering innovative approaches for real-time monitoring and diagnosis. Biosensors are structured with distinct components, comprising the sensitive segment, the transducer or detector module, and the signal processor. The sensitive portion typically involves a biologically derived element responsible for interacting with the target analyte. This interaction triggers a signal, which is then transduced by the detector component. The transducer converts this signal into a quantifiable and interpretable output, allowing for the measurement of the analyte’s concentration or presence. Finally, the signal processor refines and presents the received signal in a format that is comprehensible and accessible to the user, facilitating further analysis and interpretation [244]. In GBM, various biomarkers, including nucleic acids, proteins, microRNAs (miRNAs), circulating tumor cells (CTCs), extracellular vesicles, and tumor tissues, are commonly utilized for detection [245,246,247]. Early detection of GBM is crucial for timely intervention and improved patient outcomes [248]. These biomarkers play critical roles in understanding tumor progression and guiding therapeutic strategies (Figure 7).

Conventional diagnostic methods for GBM, including computed tomography (CT) scans, magnetic resonance imaging (MRI), and positron emission tomography (PET) scans, play a crucial role in the initial detection and characterization of these primary brain tumors. CT scans provide widespread availability, faster scanning times, and lower cost compared to MRI, although they expose patients to radiation and have limited soft tissue visualization capabilities. On the other hand, MRI offers high sensitivity and resolution for detecting glioblastomas, aiding in surgical guidance and treatment planning [249,250,251]. Positron emission tomography (PET) has emerged as a pivotal tool in the diagnosis, prognosis, and monitoring of glioblastomas, providing insights into the biological aspects of these tumors. PET imaging using radiotracers related to glucose metabolism and amino acid transport offers valuable information for non-invasive grading, differential diagnosis, tumor delineation, and treatment planning. Recent advancements in PET imaging using radiolabeled amino acids have led to recommendations for their essential use in the diagnostic assessment of brain tumors [252,253,254]. While conventional diagnostic methods provide valuable information for detecting and characterizing GBMs, there is a growing need for novel biosensor methods to complement these techniques. Biosensor technology offers the potential for sensitive, specific, and non-invasive detection of molecular biomarkers associated with GBM, addressing the limitations of conventional imaging methods. These biosensors can detect biomarkers indicative of tumor presence, progression, and response to treatment, aiding in early diagnosis, personalized treatment strategies, and therapeutic monitoring [2].

Biosensor technology offers a promising solution by providing sensitive and specific detection of GBM biomarkers in real-time. By leveraging the specificity of biomolecules and physicochemical transducers, biosensors can convert biological signals into measurable outputs, such as electrical or optical signals [255]. Various types of biosensors have been developed for GBM detection, each offering unique advantages in terms of sensitivity, specificity, and ease of use. In recent years, significant progress has been made in identifying molecular biomarkers that play key roles in GBM diagnosis and treatment. For instance, mutations in isocitrate dehydrogenase (IDH) and the methylation status of O6-methylguanine-DNA methyltransferase (MGMT) have emerged as crucial indicators of GBM prognosis. Telomerase reverse transcriptase (TERT) promoter mutations are currently a subject of investigation due to their implication in telomere maintenance within GBM. Among other noteworthy molecular biomarkers are B7-H3, chondroitin sulfate proteoglycan-4 (CSPG4), carbonic anhydrase-IX (CAIX), GD2, human epidermal growth factor receptor 2 (HER2), interleukin 13 receptor alpha 2 (IL13R α2), matrix metalloproteinase (MMP2), trophoblast-cell-surface antigen 2 (TROP2), and 1p/19q co-deletion. Additionally, ATRX mutations, EGFR (epidermal growth factor receptor) alterations, CD70, CD147, CDKN2A deletions, exosomes, cfDNA (cell-free DNA), ctDNA (circulating tumor DNA), and CTCs (circulating tumor cells) stand out as significant markers. These markers collectively offer valuable insights into the genetic and molecular landscape of gliomas, thereby influencing both diagnostic approaches and prognostic predictions. These biomarkers provide valuable insights into the genetic and molecular characteristics of GBM, guiding personalized treatment strategies [256,257,258,259,260,261,262,263,264,265]. Colorimetric biosensors have gained attention for their simplicity and cost-effectiveness in detecting tumor markers associated with GBM. These biosensors utilize color changes triggered by catalytic reactions or enzymatic conversions to detect specific biomolecules. Gold nanoparticles (GNPs) have been particularly promising in detecting glioma-associated biomarkers, offering high sensitivity and specificity in identifying GBM-specific markers. The integration of novel nanomaterials into colorimetric assays continues to enhance their detection sensitivity and simplify assay methodologies. A recent study introduced a novel colorimetric peptide nucleic acid loop-mediated isothermal amplification (CPNA-LAMP) technique for rapid and extraction-free detection of the IDH1 mutation in GBM samples. This technique selectively amplifies the DNA sequence containing the IDH1-R132H mutation, enabling swift detection within a timeframe of under 1 h without the need for nucleic acid extraction. The CPNA-LAMP method demonstrated high specificity and accuracy in detecting the IDH1 mutation, showcasing its potential as a valuable tool for GBM diagnosis and prognosis assessment [266].

Electrochemical biosensors represent versatile tools capable of detecting electrochemical reactions and precisely measuring changes occurring at the electrode surface. These biosensors rely on modulating the number of transported ions on the electrode surface, establishing a direct relationship between analyte concentration and the resulting electrochemical signal. In the context of cancer diagnosis, electrochemical biosensors offer immense potential for detecting tumor biomarkers with high sensitivity and specificity. Recent studies have highlighted the potential of electrochemical biosensors for the early diagnosis of GBMs.

A study was conducted to introduce a novel and highly sensitive analytical method for the comprehensive profiling of circulating exosomes directly sourced from the serum plasma of individuals diagnosed with GBMs. This pioneering approach involved the utilization of target-specific metal nanoparticles for labeling exosomes, coupled with a miniaturized integrated magneto-electrochemical sensing system for detection. Remarkably, this integrated platform exhibited superior detection sensitivity when compared to existing methods, enabling the discrimination of GBM-derived exosomes from those originating from non-tumor host cells. Moreover, the study demonstrated the potential utility of circulating GBM exosomes in analyzing primary tumor mutations and predicting treatment-induced alterations [267].

In a 2023 study by Wang et al., a multivalent aptamer nanoscaffold cytosensor was developed for detecting glioma circulating tumor cells (CTCs) during epithelial–mesenchymal transition (EMT), addressing the challenge of accurately identifying CTCs undergoing this transition, which is crucial in glioma invasion and metastasis. Early detection and monitoring of EMT-CTCs are essential for glioma diagnosis and prognosis; however, the dynamic changes in CTC biomarkers during EMT present significant detection challenges. To tackle this issue, the researchers devised a multivalent aptamer nanoscaffold-based electrochemical cytosensor (MAS-cytosensor) tailored to detect EMT-CTCs efficiently. The MAS consists of two chains, each featuring a specific aptamer detector, a region for DNA self-assembly, and a foothold for interface anchoring. Upon encountering target CTCs, the bisaptamer detector on the MAS accurately identifies and captures the CTCs, facilitating their immobilization on the electrode surface and generating electrochemical signals. The MAS-cytosensor exhibited remarkable sensitivity in detecting EMT-CTCs, with a detection limit of 6 cells/mL in buffer solution. Additionally, it allows downstream analysis after CTC release with high viability. This cytosensor provides a dependable detection solution for CTCs regardless of their EMT status, offering an efficient method for studying the role of post-EMT CTCs in clinical applications and metastasis mechanisms [268].

In a study conducted by Yang and colleagues in 2022, the detection of methylation levels in the O6-methylguanine-DNA methyltransferase (MGMT) gene emerged as a critical endeavor for the diagnosis, prognosis, and treatment of glioblastoma. To address this need, the researchers proposed an electrochemical immunosensor tailored for the detection of MGMT gene methylation. The methodology involved the application of a graphene oxide-magnetic nanoparticles-β-cyclodextrin (GO-Fe3O4-β-CD) nanocomposite, which was coated onto an electrode surface. Subsequently, anti-5-methylcytosine (5mC) antibodies were immobilized on the modified electrode through host–guest interaction with β-CD. This immobilized antibody exhibited specific recognition and direct capture of 5mC on DNA containing the MGMT promoter sequence (MGMT-DNA). Evaluation of the methylation level in MGMT-DNA was facilitated by utilizing Ru(NH_3_)_6_^3+^ as an electrochemical signal indicator. The redox recycling of Ru(NH_3_)_6_^3+^ was initiated by Fe(CN)_6_^3−^, leading to a further amplified current response. Remarkably, the method achieved an impressive detection limit as low as 0.0825 pM.

Finally, the effectiveness of this electrochemical approach was demonstrated through its successful application in the analysis of methylated MGMT-DNA in biological samples. Furthermore, the method exhibited versatility by enabling the detection of other methylation modifications, such as N6-methyladenosine in RNA [269].

In 2019, Scoggin and their research group developed an enzyme-based electrochemical biosensor probe with sensitivity to detect astrocytic versus glioma uptake of glutamate in real time in vitro. Glutamate, a crucial excitatory neurotransmitter in the central nervous system, plays a vital role in regulating various brain functions, such as thought, movement, and memory. Dysregulation of glutamate signaling is linked to severe neuropathological conditions like epilepsy and glioma, a form of brain cancer. While current neurochemical probes offer some means of detecting glutamate signals, there is a significant technology gap in continuously and in real-time measuring glutamate dynamics from multiple brain locations. To address this gap, the researchers developed an enzymatic glutamate microbiosensor in the form of a ceramic-substrate-enabled platinum microelectrode array. This biosensor can continuously measure changes in glutamate concentration from multiple recording sites in real time. Notably, it exhibits nearly four-fold higher sensitivity to glutamate compared to previously reported enzymatic sensors. Analysis of glutamate dynamics recorded by the microbiosensor in cultured astrocytes (control condition) and glioma cells (pathological condition) effectively distinguished normal from impaired glutamate uptake, respectively. These findings underscore the potential of the developed glutamate microbiosensor array as a valuable tool for monitoring and understanding glutamate signaling in both normal and pathological conditions [270].

In another study, researchers developed a 3D enzymatic electrochemical sensor integrated with collagen hydrogel to monitor lactate levels in cells cultured within the matrix. Lactate oxidase immobilized on the sensor’s outer surface facilitated the oxidation of lactate, producing H_2_O_2_ as a byproduct. Prussian blue nanoparticles electrodeposited on carbon nanotubes adsorbed to the electrode provided excellent and stable electrocatalytic performance for H_2_O_2_ detection, enabling accurate lactate quantification. The 3D sensor was successfully integrated with collagen hydrogel, with C6 glioma cells seeded within the matrix. This integration allowed for the real-time monitoring of lactate released from the glioma cells. The platform provided an effective detection tool for exploring the biochemical responses within 3D cell culture systems, offering insights relevant to understanding glioblastoma biology and response to treatment [271].

Further, another research group developed an aptamer electrochemical sensor designed for the rapid and efficient detection of glioblastoma multiforme cells (GMCs). The sensor was constructed by activating carboxyl groups following the modification of 4-carboxyphenyl diazonium salt. Subsequently, the aptamer, specifically recognizing GMC, was immobilized on the electrode surface through the formation of an amide bond between the amino and carboxyl groups present on the aptamer. Detailed investigations were conducted on the conditions influencing the sensing capabilities of the aptamer sensor. The sensor’s current exhibited a proportional relationship with GMC concentration within the range of 1 × 10^7^ to 1 × 10^2^ cells/mL. Additionally, the sensor demonstrated reasonable specificity in identifying GMC [272].

In a 2020 study led by Sun and colleagues, they developed an electrochemical biosensor utilizing Zr-based metal–organic frameworks (Zr-MOFs) for detecting GBM-derived exosomes, offering practical applications. The proposed electrochemical biosensor, sensitive and label-free, incorporates a peptide ligand capable of specifically binding to the human epidermal growth factor receptor (EGFR) and EGFR variant III mutation (EGFRvIII), both overexpressed on GBM-derived exosomes. Simultaneously, Zr-MOFs encapsulated with methylene blue can adhere to the exosome surfaces due to the interaction between Zr4+ and the intrinsic phosphate groups outside of the exosomes. The exosome concentration is directly quantified by monitoring the electroactive molecules inside the MOFs, ranging from 9.5 × 10^3^ to 1.9 × 10^7^ particles/µL, with a detection limit of 7.83 × 10^3^ particles/µL. The proposed biosensor has the ability to differentiate GBM patients from healthy groups, showcasing its significant potential for early clinical diagnosis [273].

In a recent study, a label-free electrochemical biosensing system was developed for the sensitive detection of epidermal growth factor receptor (EGFR), a protein frequently overexpressed in various cancers, including lung cancer, breast cancer, and GBM. This innovative system utilized magnetic Fe_3_O_4_/α-Fe_2_O_3_@Au nanocomposites as signal amplifiers and for the immobilization matrix of specific probes (ssDNA-APT). The Fe_3_O_4_/α-Fe_2_O_3_@Au/ssDNA-APT biosensors were constructed through Au–S binding on this matrix and were immobilized onto a magnetic glassy carbon electrode (MGCE) using magnetic-induced self-assembly. The performance of the EGFR detection system was evaluated using differential pulse voltammetry (DPV), demonstrating a wide linear range from 0.1 ng/mL to 1000 ng/mL (R^2^ = 0.9947), with a limit of detection (LOD) of 0.18 ng/mL. Additionally, the biosensing system was assessed for repeatability, stability, selectivity, and its ability for real sample analysis. With its simplicity and reliability, this biosensing system holds promise as a valuable tool for clinical EGFR detection, with implications for various cancers, including GBM. Its potential for point-of-care (POC) testing makes it particularly appealing for rapid and accurate diagnosis and monitoring of GBM and other cancers, contributing to improved patient outcomes [274].

Optical biosensors represent a versatile class of analytical tools that find utility across various biosensing applications, owing to their facile operational characteristics, capacity for multiplexed analyte detection, and compatibility with automated microfluidic systems [275]. Functioning on the principle of monitoring the interaction between a recognition element and the target analyte, optical biosensors generate quantifiable signals indicative of this interaction [276]. These sensors, alternatively termed light-based biosensors, detect alterations in specific light wavelengths, utilizing transducers such as luminescence, fluorescence, colorimetric, or interferometric techniques. Variations in wavelengths or surface plasmon resonance (SPR) triggered by analyte recognition are subsequently converted into electrical or digital readouts employing optical transducers. In the realm of GBM, optical biosensors offer notable advantages for diagnostic, prognostic, and therapeutic monitoring applications. Capitalizing on the distinct biomarkers expressed in GBM cells, optical biosensors enable rapid and sensitive detection of GBM-related molecules, facilitating early disease diagnosis and the design of personalized treatment regimens. Moreover, integration of optical biosensors into intraoperative imaging systems empowers surgeons to visualize tumor margins in real-time during surgical resection, thereby enhancing the precision and completeness of tumor removal while mitigating damage to adjacent healthy brain tissue. Furthermore, optical biosensors serve as invaluable tools for monitoring therapeutic responses in GBM patients by detecting changes in molecular markers indicative of treatment efficacy or resistance. Offering real-time, non-invasive assessment of disease progression and treatment outcomes, optical biosensors hold significant promise for enhancing patient management strategies in GBM. Fluorescence resonance energy transfer (FRET)-based biosensors and surface plasmon resonance (SPR)/localized surface plasmon resonance (LSPR) biosensors stand out as key optical technologies in the realm of glioblastoma multiforme (GBM) diagnosis and treatment. FRET-based biosensors offer a unique capability to monitor protein-protein interactions within GBM cells [277,278]. By leveraging the transfer of energy between fluorescent molecules, these biosensors provide valuable insights into tumor invasion and progression, shedding light on critical cellular mechanisms. On the other hand, SPR and LSPR biosensors represent a label-free and real-time detection approach for GBM biomarkers. Harnessing the distinctive properties of surface plasmons, these biosensors detect molecular interactions at the nanoscale level, facilitating the precise characterization of GBM-related molecules. Their high sensitivity and specificity make them powerful tools in the identification and analysis of biomarkers associated with GBM [279,280,281].

In a prospective study involving 35 patients with GBM undergoing surgical treatment, a nanostructured gold biosensor was utilized ex vivo to differentiate between tumor tissue and peritumoral tissue. Each patient provided paired samples of tumor and peritumoral tissue, which were analyzed individually on the biosensor surface to determine the difference in their refractive indices. The optical biosensor, based on plasmonic nanostructures, offered a label-free approach to detecting the variations in refractive index upon interaction with the tissue samples. Histopathological analysis confirmed the tumor and non-tumor origins of each tissue sample. The results revealed that the refractive index values obtained from the tissue imprints were significantly lower in peritumoral samples compared to tumor samples. The biosensor demonstrated a high discriminatory capacity between the two tissues, as indicated by the receiver operating characteristic (ROC) curve analysis, with an area under the curve of 0.8779. An optimal refractive index cut-off point of 0.003 was determined using the Youden index, resulting in a sensitivity of 81% and a specificity of 80%. Overall, the plasmonic-based nanostructured biosensor proved to be a label-free system capable of accurately distinguishing between tumor and peritumoral tissue in real-time during surgery for GBM patients [282].

Photonic crystals (PhCs) have emerged as powerful tools for optical data processing and sensing, particularly in the realm of brain tissue analysis. Nouman et al. (2020) devised a 1D PC sensor specifically tailored for detecting brain lesions, showcasing remarkable sensitivity and a notable figure of merit (FOM) of 6.16 × 10^7^ (1/RIU) [283]. Building upon this foundation, Asuvaran and Elatharasan (2021) introduced 2D PhC sensors with enhanced quality factors, enabling precise differentiation between normal and abnormal brain tissues [284]. Subsequently, Mohammed et al. (2023) demonstrated significant advancements with PhC sensors boasting heightened sensitivity, effectively detecting a wide range of brain tissues and tumors, with detection limits spanning from 10^−5^ to 10^−6^ RIU [285].

Metamaterial-inspired biosensors have garnered significant attention in recent years due to their potential for fast and cost-effective detection in the terahertz (THz) frequency range. In a study conducted by Zhang and colleagues, a novel label-free biosensing approach for molecular classification of glioma cells is proposed. The biosensor design integrates cut wires and split ring resonators to achieve polarization-independent electromagnetic-induced transparency (EIT) at THz frequencies. Simulation results indicate that the EIT-like resonance exhibits changes in both resonance frequency and magnitude in response to alterations in analyte properties, a phenomenon elucidated using coupled oscillator model theory. Theoretical assessment demonstrates a high sensitivity of up to 496.01 GHz/RIU for the biosensor. Experimental validation involves culturing two types of glioma cells (mutant and wild-type) on the biosensor surface. The observed dependencies of frequency shifts and peak magnitude variations on cell concentrations offer a novel perspective for the molecular classification of glioma cells. Remarkably, the measured results indicate that mutant and wild-type glioma cells can be directly distinguished by monitoring both variations of EIT resonance frequency and magnitude at any cell concentration, without requiring antibody introduction. This metamaterial-based biosensor demonstrates promising potential for recognizing different molecular types of glioma cells, presenting an alternative avenue for sensitive biosensing technology [286].

Overall, biosensor technology holds immense potential for revolutionizing GBM diagnosis and treatment by enabling early detection, real-time monitoring, and precise characterization of tumor biomarkers. Continued advancements in biosensor development and integration with other diagnostic modalities promise to improve our understanding of GBM biology and ultimately enhance patient care and outcomes.

### 4.1. Advancing Organ-on-Chip Technology with Integrated Biosensors

The integration of OOC technology with biosensors represents a significant advancement in biomedical research, particularly in disease modeling and personalized medicine. OOCs, sophisticated microfluidic systems designed to mimic the physiological characteristics of living tissues and organs, provide versatile platforms for studying complex physiological processes at a microscale level. Augmented with biosensors, these microfluidic devices gain the ability to capture, analyze, and quantify molecular events and cellular responses with exceptional sensitivity and precision [287,288].

Within OOC devices, continuous monitoring of biochemical and physical parameters related to cells is essential. This monitoring encompasses aspects such as cell viability, growth, organization, differentiation, metabolism, and responses to stimuli. Of particular significance among these parameters is the monitoring of cell metabolism, as it offers crucial insights into cellular functions. For instance, measuring glucose or lactate levels, fundamental substances in cellular metabolism, allows for the assessment of cell activity. Additionally, observing how cells respond to external stimuli reveals valuable physiological and pathophysiological information about cell and organ vitality. This monitoring aids in confirming the therapeutic effects of drugs, evaluating potential side effects, and assessing immune responses [289].

Various biosensor types, including electrochemical, electrical, optical, mechanical, and strain biosensors, are commonly employed for monitoring cells within OOC platforms. Furthermore, other biosensor types, such as mechanical and strain biosensors, find applications in OOCs. The integration of multiple biosensors holds promise for uncovering additional insights into physiological processes. These biosensors facilitate automated and non-invasive analysis of cellular behavior, tissues, and organs, overcoming the limitations of traditional end-point analysis methods. They encompass a spectrum of analytical techniques, including electrochemical, optical, piezoelectric, thermal, magnetic, and micromechanical sensors. The following sections will delve into the specifics of these biosensors and their integration into organ-on-chip models, elucidating their contributions to advancing biomedical research [290,291,292,293].

#### 4.1.1. Electrochemical Biosensors

Among the various biosensors, electrochemical biosensors are prominently used in OOC platforms due to their wide dynamic linear range and low detection limits. Typically, these biosensors rely on the binding of analytes to their bioreceptors immobilized on the working electrode, leading to changes in electrical signals compared to the reference electrode [294,295]. Electrochemical biosensors can be broadly categorized into four types based on the electrical signals monitored: amperometric, voltammetric, conductometric, and potentiometric biosensors [296]. They are further classified based on the types of receptors immobilized on the electrode, including enzyme-based, DNA-based, antibody-based, and whole-cell-based biosensors [297]. Cellular metabolism and functions are commonly monitored by measuring species such as glucose, oxygen, and lactate secreted by cells. Enzyme-based electrochemical biosensors are frequently utilized for monitoring these metabolites. For instance, Misun et al. developed a modular analytical platform incorporating microfluidic hanging-drop networks with enzyme-based multi-analyte biosensors [298]. This platform facilitated the continuous monitoring of glucose consumption and lactate secretion in human colon cancer microtissues. The biosensors exhibited good sensitivities, addressing challenges related to extending biosensor lifetimes and achieving measurement reproducibility.

In a recent study conducted by Dornhof and colleagues in 2022, a microfluidic organ-on-chip platform was developed for matrix-based, heterogeneous 3D cultures. This platform featured fully integrated electrochemical chemo- and biosensor arrays designed to monitor energy metabolites such as oxygen, lactate, and glucose. The advanced microstructures of the platform facilitated straightforward integration of cell matrices with standard laboratory equipment, compartmentalization, and microfluidic access. Within this system, single patient-derived triple-negative breast cancer stem cells were observed to develop into tumor organoids in a heterogeneous spheroid culture on-chip. The platform offered unprecedented control over culture conditions, including the induction of hypoxia, while simultaneously allowing verification through integrated sensors. Notably, the study demonstrated precise and reproducible multi-analyte metabolite monitoring under dynamic conditions for over one week, surpassing previous methodologies. The study’s findings revealed insights into the responses of 3D cell cultures to alterations in culture conditions and exposure to cancer drugs, including metabolite consumption and production rates. Importantly, these responses were quantitatively and continuously monitored in real-time, offering advantages over traditional endpoint analyses. This approach underscores the significance of continuous, in situ metabolite monitoring in 3D cell cultures for standardizing and controlling culture conditions, as well as for drug screening in cancer research [299].

Unique biomarkers of cells are also employed to evaluate cell metabolism and functions. However, the biological environment within OOCs is complex, containing numerous proteins and interfering compounds alongside trace amounts of biomarkers of interest. To address this challenge, aptamers, nucleic acids with artificially designed sequences, have emerged as sensitive recognition elements for biosensors [300,301]. Aptamer-based biosensors offer advantages such as temperature stability, low cost, and reusability. Shin et al. developed an aptamer-based electrochemical biosensing platform for monitoring the creatine kinase (CK)-MB biomarker secreted from damaged cardiac tissue on a heart-on-a-chip platform [302]. This biosensor exhibited ultrahigh sensitivity, a wide dynamic range, and a long shelf life compared to antibody-based biosensors.

In recent years, the application of nanocomposites has become prevalent in electrochemical biosensing technologies due to their exceptional properties, including heightened sensitivities and specificities, rapid responses, and cost-effectiveness. For instance, Chen et al. fabricated a novel Au@MnO_2_/MoO_3_ nanocomposite applied to modify screen-printed electrodes for the detection of the heat shock protein 70 biomarker in a lung adenocarcinoma cell line [303]. The electrochemical immunosensor demonstrated remarkable accuracy, sensitivity, and selectivity, highlighting its potential for cell monitoring applications within OOC platforms. Electrochemical sensors play a pivotal role in investigating cellular responses to stimuli, particularly focusing on monitoring physiological changes induced by various external cues such as electrical and mechanical stimuli. This approach predominantly examines chemical alterations within OOC systems. For instance, Ortega et al. proposed an electrochemical impedance spectroscopy (EIS)-based monitoring platform for detecting interleukin 6 (IL-6) and tumor necrosis factor α (TNF-α) following electrical stimulation of skeletal muscle tissue [304]. They immobilized IL-6 and TNF-α antibodies on screen-printed gold electrodes and detected the targets using enzyme-linked secondary antibodies. This method enabled the detection of IL-6 and TNF-α with Limits of Detection (LOD) of 8 ng mL^−1^ and 2 ng mL^−1^, respectively. Additionally, to monitor mechanically induced biochemical signals, Jin et al. developed a stretchable electrochemical sensor integrated into a vascular chip model to monitor biochemical signals during vascular mechanotransduction in real-time [305]. The sensor, comprising conductive polymer-coated carbon nanotubes, exhibited mechanical compliance against deformation and excellent electrochemical performance. They observed and recorded stretch-induced nitric oxide (NO) and reactive oxygen species (ROS) signals under normal and hypertensive conditions. The results suggested that ROS signaling is more readily induced by circumferential strain, potentially elucidating endothelial damage and dysfunction under hypertension.

Electrochemical biosensors have been utilized to monitor cytotoxicity and disease pathology on organ-on-a-chip platforms. For instance, Shang et al. developed an interconnected organ-on-a-chip system comprising a “molecular–electronic” sensor along with other modules like mixers and quality-control units. In their study, the sensor was designed to detect a specific substance by integrating multiple modules, enabling the assessment of cytotoxicity in an organ-on-a-chip containing an intestinal epithelial culture. The study conducted a cytotoxicity test using a molecular-electronic-sensor-based chip system. Upon treatment of human epithelial colorectal adenocarcinoma cells (Caco-2, ATCC, Manassas, VA) with a drug mimic (Triton X-100), leading to damage of the cell membrane, lactate dehydrogenase (LDH) was released from the cell cytosol, serving as a marker for cytotoxicity. LDH is rapidly released upon plasma membrane damage and is commonly used as an indicator of cell damage. The electrochemical redox capacitor sensor employed in the study utilizes catechol-modified chitosan to store and exchange electrons. It enables real-time measurement of ion exchange following the conversion of lactate and NAD+ to pyruvate and NADH, respectively, catalyzed by the released LDH. NADH acts as an electron donor, charging the film in the sensing module, and the sensor measures LDH concentration based on the signal generated during the electrochemical discharge of NADH. The concentration of LDH is directly proportional to the number of reduced catechol moieties in the sensing module [306].

#### 4.1.2. Optical Biosensors

Optical biosensors operate by detecting interactions between a target analyte and a recognition element and then converting these interactions into optical signals proportional to the concentration of the analyte. They utilize various biomaterials as recognition elements, including enzymes, antibodies, antigens, receptors, nucleic acids, whole cells, and tissues [298]. Optical biosensors offer numerous advantages for monitoring biological parameters in OOC systems, including high sensitivity and specificity, low limits of detection, minimal background noise, compact size, and cost-effectiveness [307].

These biosensors are commonly employed to monitor cell metabolism and functions within OOC systems and can be categorized into label-free and label-based models. Label-free biosensors detect signals directly generated by the analyte’s interaction with the transducer, relying on inherent material properties such as atomic mass, size, surface charge, electrical impedance, and index of refraction for detection [308]. In contrast, label-based biosensors utilize labeling reagents that produce optical signals detectable by colorimetric, fluorescent, or luminescent methods [309].

Light scattering is a common, label-free optical biosensing method. Rothbauer et al. developed a three-dimensional synovium-on-a-chip with a non-invasive light-scattering biosensor to study tissue-level remodeling during inflammatory arthritis. A collimated laser beam irradiated the device, and an embedded organic photodetector array collected the scattered portion of the beam to monitor tissue responses [310].

Surface Plasmon Resonance (SPR) is a widely used label-free optical biosensing method that involves the resonant oscillation of incident light at the interface of two media. It enables real-time detection of target substances and has been applied to protein and nucleic acid detection. For instance, Ortega et al. integrated a localized surface plasmon resonance (LSPR) sensing module into an OOC to monitor in situ insulin secretion under external glucose stimulation. They achieved an LOD of 0.85 ± 0.13 μg mL^−1^ by fabricating gold nanorod arrays functionalized with capture antibodies [311].

Schneider et al. proposed a method for constructing organ-on-a-chip systems by integrating optical sensors and electrical stimulation electrodes into a chip containing human-induced pluripotent stem cell-derived cardiomyocytes and primary fibroblasts. Real-time oxygen measurements were performed via a 4-channel phase fluorometer and connected optical fibers, revealing changes in oxygen consumption under different electrical stimulation conditions [312].

In terms of label-based biosensors, fluorescent biosensors are frequently used due to their high sensitivity, selectivity, and stability [313].

Kobuszewska and colleagues utilized a fluorescent cationic carbocyanine dye to detect oxidation levels during cellular respiration and assess the influence of stem cells on cardiac cell functions. These biosensors rely on the binding of oxygen molecules to a fluorescent dye to reflect the relationship between oxygen concentration and fluorescence intensity [314].

#### 4.1.3. Electrical Biosensors

Electrical biosensors are sophisticated devices designed to detect changes in the electrical properties of their sensing elements upon exposure to an analyte. Unlike electrochemical sensors, which undergo chemical reactions during detection, the electrode surface of electrical sensors typically remains unaltered. This feature makes them valuable tools for monitoring biological processes, especially within OOC systems, where their integration is facilitated by expertise in microelectronics, particularly in miniaturized electrode integration. Through measurements of voltage changes at the electrodes, these biosensors can provide detailed insights into various cell properties, such as the formation of tight junctions in cell barriers or alterations in cell morphology. Furthermore, their application extends to monitoring cellular responses to stimuli, thereby contributing to the comprehensive analysis of biological processes within OOC systems.

Cellular barriers are fundamental structures in various physiological systems, encompassing critical organs like the skin, intestines, lungs, liver, and kidneys. They serve as crucial interfaces regulating the exchange of substances and play indispensable roles in drug absorption and distribution within the body. Assessing the integrity of these barriers is essential for understanding cellular states and evaluating the efficacy of drug treatments. Transepithelial/endothelial electrical resistance (TEER) measurements are widely recognized as rapid, conventional, and non-invasive assays for monitoring barrier integrity. These measurements involve quantifying the electrical impedance across epithelial or endothelial layers, which correlates directly with the formation and maintenance of tight junctions between neighboring cells. Tight junctions are integral to barrier function as they regulate the passage of ions, molecules, and cells through the intercellular space [315].

Henry et al. pioneered the application of TEER measurements to assess barrier functions within OOC models. They developed a sophisticated microfabrication approach to construct complex microfluidic chips with integrated electrodes, enabling precise monitoring of barrier properties. Their innovative technique utilized polycarbonate (PC) as a substrate material due to its favorable optical properties, biocompatibility, and ease of modification. The integrated electrodes allowed for the measurement of TEER across cell layers, providing valuable insights into barrier integrity and function [316].

The blood–brain barrier (BBB), a dynamic interface between the circulatory system and the central nervous system, presents unique challenges for barrier assessment due to its complex structure and function. Vatine et al. addressed these challenges by developing a human BBB chip using induced pluripotent stem cells (iPSCs) and gold electrodes. Their innovative chip design allowed for the measurement of physiologically relevant TEER values, even under conditions of high shear stress [317]. To overcome the technical limitations associated with TEER measurements, Giampetruzzi et al. developed an intestinal barrier-on-chip (IBoC) with a dedicated electrical model. Their approach utilized a Topas polymer platform modified with transparent indium tin oxide (ITO) electrodes and a porous membrane, facilitating accurate TEER calculation and continuous monitoring of barrier formation and maturation [318]. In addressing challenges related to barrier movements, Mermoud et al. incorporated a flexible printed circuit board (PCB) with a micro-impedance tomography (MITO) system into a lung-on-a-chip microfluidic platform. This innovative system allowed for the monitoring of resistivity changes within biological barriers, providing insights into barrier dynamics under conditions mimicking respiratory motion [319].

Electrical biosensors play a crucial role in directly investigating the electrical properties of cells under various stimuli, providing a more accurate model for drug screening and disease research. These biosensors not only assist in recording local field potentials and action potentials but also offer stimulating functions involving current and voltage. A study established a low-volume pump-free organ-on-a-chip platform for cardiotoxicity analysis, integrating iPSC-derived cardiomyocytes and primary hepatocytes. They measured the electrical activity of cardiomyocytes using a custom multi-electrode array (cMEA) and maintained the full functionality of both cell types for 7–28 days. Subsequently, they proposed a four-organ system, including the liver, heart, nervous system, and skeletal muscle modules, allowing for long-term monitoring of cellular functions under serum-free conditions [320].

In another study, researchers developed an optically transparent thin-film transistor array (TFT) biosensor for neuronal ensemble investigation, offering six modalities for the TFT electrodes, including recording of both local field potential (LFP) and action potential (AP), chemical stimulation, current or voltage stimulation, electrical impedance measurements, and optical imaging. This biosensor, comparable to state-of-the-art microelectrode array (MEA) devices, featured a large sensing area with high spatial and temporal resolution [321].

#### 4.1.4. Multiple Biosensors

Combining OOCs with multiple biosensors holds promise for unveiling comprehensive insights into cellular behavior. The integration of multiple biosensors with OOC models represents a significant advancement in biomedical research, offering a multifaceted approach to studying cellular behavior and complex physiological processes. This innovative combination allows for the simultaneous monitoring of various biomarkers, cellular responses, and environmental conditions within a controlled microfluidic environment. In recent years, numerous studies have explored the integration of multiple biosensors into OOC platforms to enhance their functionality and analytical capabilities. These biosensors can be designed to detect a wide range of parameters, including but not limited to biochemical markers, cellular electrical activity, mechanical forces, and environmental factors. By incorporating multiple biosensors, researchers can obtain comprehensive data on cellular function, signaling pathways, and interactions in real-time. Electrical and optical biosensors, utilized in HuMiX OOC systems, provide insights into oxygen levels (O_2_) and transepithelial electrical resistance (TEER), crucial for understanding cell growth, differentiation, and tissue integrity [322]. In gut-on-a-chip models, electrochemical biosensors play a pivotal role in monitoring O_2_ levels using dissolved oxygen (DO) and reactive oxygen species (ROS) sensors, shedding light on cellular metabolism and oxidative stress [323]. Additionally, electrochemical biosensors and electrical biosensors employed in distal tubule-on-a-chip models enable assessment of TEER, glucose levels, and monitoring of reactive oxygen species (ROS), contributing to the study of cell morphology, differentiation, and tissue integrity [324]. For drug research, proximal tubule-on-a-chip models utilize electrochemical biosensors, electrical biosensors, and microscopes for monitoring TEER, pH, and glucose levels, crucial in assessing nephrotoxicity and drug efficacy [325]. In hiPSC-derived cardiac and human liver cancer models on a dual-organ platform, different electrochemical biosensors provide insights into cardiac and hepatic function by monitoring multiple biomarkers like albumin, GST-α, and CK-MB [326]. Furthermore, electrochemical biosensors and electrical biosensors integrated into colorectal perfused tumoroid platforms monitor pH and cell proliferation, essential for evaluating drug cytotoxicity and tissue viability [327]. Additionally, optical biosensors, electrochemical biosensors, and fluorescence microscopy in lung cancer-on-a chip models enable the assessment of pH, TEER, and biomarkers, aiding in the evaluation of drug-induced organ toxicity [328]. Further, optical biosensors and electrochemical biosensors integrated into cancer heart and liver models enable multi-biosensor integration, facilitating comprehensive toxicity assessment [329].

The pharmaceutical industry faces challenges due to high failure rates and rising costs in drug development [330]. Multiple biosensor-integrated OOC platforms offer biomimetic microenvironments and dynamic cues, enhancing the prediction of drug efficacy and safety in humans. Integrated biosensors facilitate reliable toxicity assessment and drug screening in OOCs. In one study, a liver-on-a-chip device was engineered to incorporate biosensors capable of measuring transepithelial electrical resistance (TEER) and pH levels in real-time. This innovative approach allowed researchers to evaluate toxicity levels in a dynamic and continuous manner, providing valuable insights into the physiological responses of liver cells within the microfluidic system [331].

#### 4.1.5. Other Types of Biosensors

In exploring other types of biosensors, Sciurti and colleagues developed a cell culture platform integrated with sensors for real-time measurements of TEER and ion levels [332]. Furthermore, Azizgolshani et al. developed a micro-pump sensor array (MPSA) system supporting various human tissue models, including liver, vascular, gastrointestinal, and kidney chips [333]. This system integrated TEER measurement components and oxygen sensors, enabling assessment of barrier function, tissue integrity, and oxygen transport and consumption. Bavli et al. developed a liver chip integrating sensors to monitor glucose, lactate, and oxygen concentrations simultaneously [334]. By immobilizing corresponding oxidases on platinum electrodes, alterations in current were observed, while oxygen levels were measured using tissue-embedded particles equipped with a ruthenium-based dye.

The cellular microenvironment plays a crucial role in governing cell growth, development, and function. A variety of biosensors have been developed to monitor key parameters within this microenvironment, aiding in a deeper understanding of cellular behavior within OOC systems. Oxygen concentration profoundly influences cell growth and differentiation. Liebisch et al. developed a zero-consumption Clark microsensor for oxygen monitoring within OOC, benefiting from chronoamperometric protocols for long-term operation [335]. Another study by Moya and colleagues proposed a liver-on-a-chip device with integrated electrochemical dissolved oxygen sensors for local oxygen concentration monitoring [336]. Electrochemical biosensors, utilizing techniques such as amperometry and voltammetry, are frequently employed for pH monitoring within the microenvironment. Tajeddin et al. introduced a microfluidic chip embedded with sensors for pH measurement, employing free flow electrophoresis to isolate ions and capture H+ ions using a micro-cantilever [337]. Liang et al. developed a modular microfluidic system integrated with a light addressable sensor (LAPS) for real-time extracellular acidification detection [338].

Temperature fluctuations within OOC systems can exert a significant impact on cellular physiology. In a study, researchers integrated smart temperature sensors onto silicon-based OOC devices, aiming to ensure precise temperature monitoring and system robustness. This innovative approach allowed for real-time monitoring of temperature changes within the OOC platform, facilitating better control of experimental conditions and minimizing potential variations in cellular behavior [339]. Similarly, in another study, resistive temperature sensors (RTDs) were developed specifically for accurate temperature monitoring in organ chips. By implementing RTDs within the OOC system, researchers could precisely measure and regulate temperature fluctuations, thereby optimizing the culture conditions for cells within the chip. This meticulous temperature control is essential for maintaining cellular viability, function, and reproducibility of experimental results in OOC studies [340]. Osmotic pressure affects cellular behavior and differentiation. Fernandes et al. developed an osmotic hydration sensor based on a semi-permeable membrane integrated with a piezoelectric resistor for osmotic pressure monitoring [341]. This sensor demonstrated low power consumption and high sensitivity, making it suitable for integration into microfluidic systems.

In conclusion, the integration of advanced biosensors into OOC devices enables comprehensive monitoring of the cellular microenvironment, providing valuable insights into cellular behavior and facilitating the development of innovative biomedical applications. The integration of various biosensors within OOC platforms offers unprecedented insights into dynamic cellular responses and microenvironmental changes. However, the complexity of modern organ systems, such as multiple interconnected OOCs, often generates diverse signals, posing challenges for detection and quantification using a single sensing technology. Consequently, the utilization of diverse biosensor combinations becomes essential for simultaneous monitoring of cells and their microenvironment, overcoming existing limitations and advancing the field. Table 7 presents a comprehensive overview of various biosensing detection methods, including their principles, targets, advantages, and disadvantages. Table 8 depicts an overview of biosensors used in various research contexts and their functions within organ-on-chip (OOC) systems.

### 4.2. Glioblastoma with Biosensor-Integrated Organ-on-Chip Models

Incorporating biosensors into GBM chip models provides a unique opportunity to elucidate the dynamic tumor dynamics within the TME. Through real-time monitoring of specific biomarkers, cellular activities, and microenvironmental factors, biosensor-integrated chip models facilitate a thorough analysis of GBM progression and response to therapy. GBM-specific biomarkers, such as tumor-associated proteins or genetic mutations, serve as discerning targets for biosensors, offering valuable insights into tumor dynamics by quantifying changes in biomarker expression levels over time. This enables researchers to track tumor growth, invasion, and treatment response within the chip model. For example, biosensors can measure proliferation markers to assess tumor growth rates or detect invasive markers to monitor tumor spread into surrounding tissues. Furthermore, biosensors integrated into GBM chip models can continuously monitor cellular activities associated with tumor progression, including proliferation, apoptosis, and metabolic activity. This ongoing measurement allows researchers to assess dynamic changes within the tumor cell population, such as the emergence of drug-resistant subpopulations or the effects of therapeutic interventions on tumor cell viability and function. In addition to cellular dynamics, biosensors enable real-time monitoring of key microenvironmental factors influencing tumor behavior, such as oxygen levels, pH, and nutrient availability. By measuring these factors within the chip model, researchers can recreate and study the complex TME, gaining insights into how environmental cues regulate tumor behavior and response to therapy. The integration of biosensors into GBM chip models also facilitates high-throughput drug screening to identify novel therapeutic agents or evaluate existing drugs’ efficacy. Biosensor data on cellular responses and microenvironmental conditions can inform the selection of candidate drugs and optimize treatment regimens based on individual tumor biology and drug sensitivities (Figure 8).

Developing a biosensor-enhanced GBM-on-chip model with relevance to the tumor microenvironment necessitates a systematic approach encompassing various key steps. Initially, appropriate biosensors must be carefully selected based on their capability to detect specific biomarkers or signaling molecules pertinent to the GBM microenvironment. These biosensors may include electrochemical, optical, or microfluidic-based sensors, chosen for their sensitivity and selectivity. Subsequently, a microfluidic chip must be designed to mimic crucial aspects of the GBM tumor microenvironment. This involves the incorporation of compartments for tumor cells, stromal cells, immune cells, and extracellular matrix components, thereby enabling interactions akin to those observed in vivo. The integration of selected biosensors into the microfluidic chip design follows, strategically placing them within the chip to facilitate real-time monitoring of targeted biomarkers or signaling molecules involved in GBM progression and microenvironmental interactions.

Following chip design and biosensor integration, GBM cells are seeded onto the microfluidic chip in a manner that replicates the spatial organization observed in GBM tumors. This may entail establishing gradients of signaling molecules or varying cell densities to mimic the heterogeneous nature of GBM tumors. Additionally, stromal cells such as fibroblasts and immune cells such as microglia and macrophages are co-cultured with GBM cells within the microfluidic chip, positioning them in close proximity to enable bidirectional interactions influencing GBM progression and immune response. Validation and calibration of the biosensors within the microfluidic chip are crucial steps, involving functional testing with known concentrations of analytes to ensure accurate detection and quantification. Parameters for detection are optimized accordingly. Subsequently, the developed GBM-on-chip model undergoes characterization to assess its fidelity in replicating key features of the GBM tumor microenvironment. This involves microscopy, immunostaining, and gene expression analysis to evaluate cell behavior, cell–cell interactions, and molecular profiles within the chip.

Functional studies and drug testing are then conducted using the biosensor-enhanced GBM-on-chip model. Real-time monitoring of changes in biomarker levels or signaling dynamics in response to different treatments, including chemotherapy, targeted therapies, or immunotherapies, provides valuable insights. Continuous iterative optimization of the GBM-on-chip model based on experimental findings and feedback is essential to improving its relevance and reliability for studying GBM biology and therapeutic responses. Finally, data generated from biosensor-enhanced GBM-on-chip experiments is analyzed to gain insights into GBM pathophysiology and therapeutic strategies. The results are interpreted in the context of the complex interactions within the tumor microenvironment, identifying potential targets for intervention based on the observed responses to different treatments. This integrated approach allows for a comprehensive understanding of GBM biology and the development of more effective therapeutic strategies for this challenging disease.

Optical biosensors provide a versatile platform for studying the dynamic microenvironment of GBM tumors through light-based detection methods. Fluorescence-based probes are instrumental in revealing oxygen distribution within GBM tumors with remarkable spatial resolution. When integrated into chip models, these probes allow precise visualization of oxygen gradients across different tumor regions, particularly highlighting hypoxic areas crucial for tumor progression and therapy resistance. Advancements in fluorescence imaging techniques, such as multiphoton microscopy and confocal microscopy, further enhance spatial resolution and depth penetration, providing a comprehensive understanding of oxygen dynamics within GBM microenvironments. Similarly, pH-sensitive dyes or nanoparticles serve as powerful tools for monitoring extracellular pH dynamics, a hallmark feature of the acidic microenvironment in GBM. Integrated into chip models, these sensors offer real-time insights into pH variations within the tumor, aiding researchers in delineating spatial and temporal patterns of acidosis. Correlating pH dynamics with tumor aggressiveness and treatment response, optical pH sensors contribute to deciphering the metabolic adaptation of GBM cells and their interactions with the surrounding microenvironment. Furthermore, optical glucose sensors offer an opportunity to investigate the metabolic dependencies of GBM cells by monitoring glucose concentration gradients within tumors. Leveraging optical techniques like surface plasmon resonance (SPR) or fluorescence resonance energy transfer, these sensors enable continuous monitoring of glucose levels with high sensitivity and specificity. Integrating SPR technology into GBM chip models offers several potential applications. By functionalizing the SPR chip surface with specific antibodies or receptors targeting GBM biomarkers, researchers can detect and quantify these biomarkers in patient samples or cell culture media. This enables non-invasive or minimally invasive monitoring of disease progression and response to treatment. Additionally, SPR facilitates label-free screening of potential therapeutic compounds or drug candidates for their binding affinity to GBM-associated targets, aiding in the identification and optimization of novel drug candidates. Furthermore, SPR provides real-time kinetic information about molecular interactions occurring on the chip surface, allowing researchers to study interactions with ligands, drugs, or other cellular components and gain insights into GBM cell behavior and microenvironmental dynamics. This technology also contributes to the development of personalized treatment strategies by profiling patient-derived samples for specific molecular signatures or drug sensitivities. Lastly, by monitoring changes in SPR signals over time, researchers can investigate the molecular mechanisms underlying GBM progression, invasion, and response to therapy within chip models, guiding the development of novel therapeutic approaches targeting key signaling pathways or cellular processes.

Transitioning to electrochemical biosensors, rooted in fundamental electrochemical principles, these sensors offer high sensitivity and rapid response times, providing a unique avenue for investigating the intricate dynamics of the GBM microenvironment. Electrochemical oxygen sensors, utilizing oxygen-sensitive electrodes or sensors, have become indispensable tools for quantifying oxygen concentrations within tumors. These sensors allow researchers to delineate oxygen gradients, shedding light on the extent of hypoxia and its implications for tumor growth, angiogenesis, and therapy resistance. Insights into the heterogeneous distribution of oxygen within the tumor highlight responses from different cell types, including GBM cells, immune cells, and stromal cells, to varying oxygen levels. Similarly, pH-sensitive electrodes or ion-selective electrodes integrated into electrochemical biosensors continuously monitor extracellular pH dynamics within GBM microenvironments. Particularly valuable for characterizing tumor acidosis, these sensors track pH variations in real-time, elucidating the role of acidosis in driving tumor aggressiveness, immune evasion, and therapy resistance. Additionally, they offer insights into the spatial and temporal dynamics of pH regulation within the tumor, revealing interactions among GBM cells, extracellular matrix components, and other stromal cells. Enzymatic reactions or glucose-sensitive electrodes within electrochemical biosensors facilitate continuous monitoring of glucose levels within GBM tumors. This offers critical insights into the metabolic dependencies of different cell types within the tumor microenvironment. Examining glucose concentration gradients and metabolic fluxes helps decipher how cells adapt their energy metabolism under varying conditions of hypoxia and acidosis. Integration of multiplexed detection platforms, comprising arrays of electrochemical sensors, enables simultaneous analysis of multiple biomarkers associated with GBM progression and treatment response. This approach allows comprehensive profiling of the molecular landscape within GBM tumors, including growth factors, cytokines, genetic mutations, and metabolic biomarkers, providing valuable insights into tumor heterogeneity and potential therapeutic vulnerabilities.

Furthermore, multimodal and multiplexed sensor systems could be used to comprehensively monitor the microenvironment and physiological parameters within GBM, enabling a deeper understanding of tumor behavior and potential therapeutic interventions. This involves the utilization of microfluidic OOC platforms equipped with integrated sensors to monitor various aspects of GBM physiology. These platforms can include a combination of optical, electrochemical, and biochemical sensors capable of detecting key metabolites, drugs, and chemical parameters within the tumor microenvironment simultaneously. 

The patch-clamp technique offers unique opportunities to study the functional activity of GBM cells within chip models. By recording intracellular electrical signals, researchers can characterize the electrophysiological properties of GBM cells within chip models. This information aids in identifying specific signaling pathways and neural networks involved in tumor growth and invasion, which is crucial for understanding GBM pathology and developing targeted therapies. In GBM-on-chip models, patch-clamp electrophysiology allows for the direct measurement of electrical currents across the cell membrane of individual GBM cells. This technique provides valuable insights into the membrane potential, ion channel activity, and synaptic responses of these malignant cells within a more physiologically relevant microenvironment. By studying the electrical properties of GBM cells in these chip models, researchers can better mimic the in vivo tumor microenvironment and understand the intricacies of GBM progression. The integration of patch-clamp recordings with other molecular and cellular assays within GBM-on-chip models enables a comprehensive understanding of GBM cell function and behavior. This approach facilitates the identification of novel therapeutic targets and the development of more effective treatment strategies for GBM.

Microelectrode arrays (MEAs) play a crucial role in studying the electrophysiological activity of GBM cells within chip models. These devices, consisting of multiple microelectrodes, allow for the non-invasive recording and stimulation of neuronal activity, providing valuable insights into the behavior of GBM cells. In GBM chip models, MEAs enable researchers to monitor the spatiotemporal electrophysiological activity of GBM cells over time. By recording action potentials and analyzing spike waveforms, firing rates, and inter-spike intervals, researchers can gain insights into the functional properties and communication patterns of GBM cells within the tumor microenvironment.

Furthermore, MEAs facilitate the study of neuronal network formation and organization in GBM chip models. By assessing the synchronization of spikes and the connectivity between GBM cells, researchers can elucidate the dynamics of tumor cell interactions and the development of aberrant neural networks within the tumor. The integration of MEAs with GBM chip models also allows for the development of novel therapeutic strategies. By investigating the effects of pharmacological microelectrode array (MEA) agents or electrical stimulation on GBM cell activity, researchers can identify potential therapeutic targets and assess the efficacy of candidate treatments in real-time.

The development of aptamer-based integrated electrochemical biosensing platforms holds significant importance for glioblastoma chip models. These nucleic acids with artificially designed sequences demonstrate high affinity and specificity for target molecules, including nucleic acids, proteins, and cells. By immobilizing aptamers onto electrochemical biosensing platforms, researchers can create highly sensitive and selective detection systems tailored for GBM biomarkers. The advantages of aptamer-based biosensing platforms are manifold. Firstly, aptamers exhibit temperature stability, allowing for reliable detection under various environmental conditions. This feature is particularly advantageous for biosensing applications, where maintaining consistent assay conditions is crucial for accurate results. Additionally, aptamers are relatively inexpensive to produce compared to antibodies, making them cost-effective options for large-scale biosensing applications. Furthermore, the reusability of aptamers adds to the cost-effectiveness of these biosensing platforms. Unlike antibodies, which may degrade or lose activity after multiple uses, aptamers can be regenerated and reused multiple times without significant loss of performance. This reusability not only reduces the overall cost of biosensing assays but also minimizes waste and environmental impact. In the context of GBM chip models, aptamer-based integrated electrochemical biosensing platforms offer several advantages. These platforms enable the sensitive and specific detection of GBM-related biomarkers directly from complex biological samples, such as blood or cerebrospinal fluid. This capability is crucial for early diagnosis, monitoring treatment response, and predicting disease progression in GBM patients.

The fabrication of nanocomposites and screen-printed electrodes (SPEs) through the drop-casting method presents an innovative approach for developing electrochemical biosensors in glioblastoma chip models. By utilizing the drop-casting technique, the nanocomposite is precisely deposited onto the electrode surface, enabling the integration of functional materials for enhanced sensing capabilities. For GBM chip models, this approach offers several advantages. Firstly, the nanocomposite modification enhances the sensitivity and selectivity of the electrodes, allowing for the detection of specific biomarkers associated with GBM progression. Additionally, the use of screen-printed electrodes facilitates the mass production of biosensors, making them suitable for high-throughput screening applications in drug discovery and personalized medicine for glioblastoma patients. Furthermore, the drop-casting method enables the facile fabrication of biosensors with tailored properties, such as increased stability and reproducibility, which are crucial for long-term monitoring of GBM cells within chip models. By leveraging these advanced electrochemical biosensors, researchers can gain valuable insights into the dynamics of glioblastoma progression, including tumor growth, invasion, and response to therapeutic interventions.

In GBM research, the integration of biosensors into various platforms has been extensively explored. However, a notable gap exists in studies specifically evaluating biosensor-enhanced GBM chip models. Organ-on-chip models offer a promising avenue for replicating the intricate microenvironment of GBM tumors in vitro, providing a more accurate representation for disease modeling and drug screening. Despite notable advancements, the incorporation of biosensors into such models remains relatively unexplored.

A compelling direction emerges with the incorporation of biosensors into blood–brain barrier-on-a-chip (BBB-oC) models. This innovative approach holds significant potential for studying not only GBM but also other neurodegenerative diseases (NDDs) with heightened precision and efficiency. GBM, known for its aggressiveness, presents unique challenges in treatment, largely due to the restrictive nature of the blood–brain barrier (BBB). The integration of biosensors into BBB-oC models provides a real-time monitoring system for disease-specific biomarkers associated with GBM progression. This capability becomes instrumental in evaluating drug permeability and efficacy in overcoming the barriers presented by the BBB. The selective detection ability of these biosensors offers deeper insights into the dynamic interplay of GBM within the microenvironment of the BBB. Crucially, by identifying and quantifying disease-specific biomarkers, BBB-oC models contribute valuable insights into the underlying mechanisms of GBM. This not only enhances our understanding of the disease but also paves the way for the development of targeted therapies. Moreover, the integration of patient-derived cells into BBB-oC models opens up avenues for personalized medicine in GBM treatment. Tailored therapeutic interventions and optimized treatment regimens based on individual patient characteristics become feasible through this approach. The automation and high-throughput screening capabilities inherent in biosensor-enhanced BBB-oC models further accelerate the drug discovery process. This expedites the translation of preclinical findings into clinical applications, potentially hastening the development of effective treatments for GBM and other NDDs. In essence, the synergistic integration of biosensors into BBB-oC models not only enriches our understanding of GBM dynamics but also holds immense promise for transforming these insights into tangible clinical benefits.

Blood–brain barrier-on-a-chip (BBB-OC) designs are crucial for replicating physiological conditions accurately. These designs are typically categorized into three configurations: stack, flank, and tubular. The stack configuration involves two channels stacked with a membrane between them, allowing separate cultures of endothelial cells (ECs) and neuronal cells. This method requires precise assembly but allows for the use of various membranes. Commercial membranes are often used, but some researchers fabricate their own to control material, thickness, and pore size. However, excessively thick membranes can impede cell–cell interactions and hinder observation. Some implementations include transendothelial electrical resistance (TEER) systems using porous membranes and integrated electrodes for real-time monitoring. Flank configurations feature horizontally arranged channels with pillars separating them, often using 3D hydrogels instead of commercial membranes. This approach is easier to industrialize and allows for high-throughput experimentation. Tubular configurations mimic cylindrical brain capillaries, typically using sacrificial layers to define cylindrical shapes or employing 3D printing techniques. These designs enable more biomimetic representations of the BBB.

TEER sensing plays a crucial role in BBB-OC, providing insights into barrier integrity. Electrode position, material, and shape influence TEER measurements. Precise positioning near the cell monolayer minimizes noise and improves accuracy. Uniform current density is vital for reliable measurements, often achieved with specific electrode shapes and configurations. Electrode materials such as gold, indium tin oxide (ITO), and Ag/AgCl are commonly used, each with its advantages and limitations. Factors like temperature and ion concentration affect TEER values and must be carefully controlled for accurate measurements.

Integration of biosensors in BBB-OC holds promise for real-time monitoring of disease-specific biomarkers and drug permeability. Optical detection methods like surface plasmon resonance and fluorescence are widely used but may not be suitable for continuous monitoring. Electrochemical transducers offer advantages such as label-free detection and affordability for integration into microfluidic channels. Biosensors can monitor various analytes, including drugs, cytotoxicity markers, ions, neuroinflammation biomarkers, and reactive oxygen species (ROS). Aptamers, enzymatic sensors, and ion-selective electrodes are common bioreceptors for biosensors, offering high selectivity and sensitivity [342,343,344].

Despite significant advancements in blood–brain barrier-on-a-chip (BBB-OC) development, these platforms are still in their early stages, representing simplified adaptations of BBB physiology. Currently, monitoring cell culture development in BBB-OC relies heavily on fluorescence labeling of specific proteins with confocal microscopy. However, this method is time-consuming, costly, and lacks real-time monitoring capabilities, impeding widespread adoption. TEER is the only integrated sensor in BBB-OC, offering advantages like real-time monitoring and automation. Various electrode materials, shapes, sizes, and configurations have been described for TEER measurement, with recommendations for efficient configurations. However, TEER measurements have limitations, including sensitivity to variables like temperature and electrolyte concentration, leading to conflicting results. Despite these challenges, some BBB-OC models have been commercialized, focusing on barrier integrity.

Future BBB-oC models are expected to enable continuous monitoring of inflammation and cell damage markers, necessitating the integration of biosensor technology. This integration opens up possibilities for continuous monitoring of various analytes, providing valuable insights into BBB permeability-related diseases and drug testing. A forward-looking approach to BBB-oC development involves creating fully automated analyzers capable of measuring and monitoring these biomarkers in real-time. This vision entails designing BBB-oC as user-friendly analyzer tools for drug testing in hospitals or the pharmaceutical industry, facilitating wider market adoption [342,343].

Su et al. have developed a novel platform termed “Digital Tissue-BArrier-CytoKine-counting-on-a-chip (DigiTACK)”, which integrates digital immunosensors into a tissue chip system. This innovative platform enables multiplexed and ultrasensitive profiling of cytokine secretion from cultured brain endothelial barrier tissues in a longitudinal manner.

The digital sensors incorporated into the DigiTACK platform employ a unique beadless microwell format to rapidly detect analytes. This approach facilitates “digital fingerprinting” of cytokines with remarkable sensitivity, achieving a low limit of detection (LoD) ranging from 100 to 500 fg/mL for key cytokines such as mouse MCP1 (CCL2), IL-6, and KC (CXCL1). Importantly, the versatility of the DigiTACK platform extends beyond brain endothelial barrier tissues, making it applicable for profiling temporal cytokine secretion in other barrier-related organ-on-a-chip systems. By enabling sequentially controlled experiments, this platform offers valuable insights into the dynamic secretory dynamics of the blood–brain barrier (BBB) and other barrier tissues. The DigiTACK platform developed by Su et al. offers a promising avenue for advancing research on GBM. By leveraging this innovative technology, researchers can gain deeper insights into the complex interplay between cytokine secretion dynamics and glioblastoma progression. Utilizing the platform, they can characterize the glioblastoma microenvironment, identify key dysregulated cytokines, assess drug efficacy, tailor personalized treatment strategies, investigate immunotherapy approaches, and integrate data with other omics technologies. Overall, the DigiTACK platform provides a powerful tool for dissecting the intricate cytokine signaling networks within the glioblastoma microenvironment and accelerating the development of novel therapeutic strategies to improve patient outcomes [345].

Exosomes, small extracellular vesicles secreted by various cell types, have garnered significant attention in cancer research due to their potential role as biomarkers for disease diagnosis and prognosis. In the context of GBM, the detection and analysis of exosomes hold particular promise for understanding disease progression and treatment response. In a recent study conducted by Yang et al. in 2024, the development of a printed divisional optical biochip for multiplex visualizable exosome analysis at point-of-care was presented, marking a significant advancement in the detection of exosomes with specific implications for GBM. The novel biochip design integrates the self-assembly of nanochains and precise surface patterning to enable multiplex exosome analysis, specifically tailored to the unique characteristics of GBM. Leveraging resonance-induced near-field enhancement, the nanochains exhibit distinct color changes upon capturing target exosomes, enabling direct visual detection. Through a meticulously designed approach, the biochip is conjugated with specific antibodies, creating a series of divisional nanochain-based biochips with hydrophilic and hydrophobic patterns. This strategic design facilitates the precise self-splitting of a single sample droplet into multiple microdroplets, enabling the simultaneous identification of multiple target exosomes within a rapid timeframe of 30 min. Of particular significance is the biochip’s exceptional sensitivity, capable of detecting exosomes at a concentration as low as 6 × 10^7^ particles mL^−1^, surpassing the capabilities of conventional enzyme-linked immunosorbent assay (ELISA) methods by approximately two orders of magnitude. This heightened sensitivity holds immense promise for the early detection of GBM-related exosomes, aiding in timely intervention and treatment. Moreover, the biochip’s semiquantitative capability is demonstrated in distinguishing clinical exosomes derived from GBM patients from those of healthy individuals. This capability offers invaluable insights into disease progression and treatment response, facilitating personalized patient care strategies. In conclusion, Yang et al.’s innovative approach presents a transformative tool for the rapid and sensitive detection of GBM-related exosomes. By offering a simple, versatile, and highly efficient method, this technology holds immense potential as a diagnostic aid for GBM, advancing the field of liquid biopsy and enhancing patient outcomes in the fight against this devastating disease [346].

Research in biosensor-enhanced organ-on-a-chip models for investigating GBM tumor microenvironment dynamics has faced limitations primarily due to the intricate nature of the GBM tumor microenvironment. The complexity of this environment, characterized by a multitude of cell types, extracellular matrix constituents, and signaling molecules, presents substantial challenges in accurately replicating its intricacies within organ-on-a-chip platforms. Technical hurdles further impede progress in this domain, as the development of biosensor-enhanced organ-on-a-chip models necessitates the integration of biosensors with microfluidic chip systems, cell culturing methodologies, and advanced imaging modalities. Challenges encompass various aspects, including biosensor design, fabrication, calibration, and integration, all demanding meticulous attention to overcome. Another impediment lies in the lack of standardized protocols and methodologies across studies, hampering the comparability and reproducibility of results. The absence of uniformity in experimental approaches obstructs scientific progress and the establishment of a cohesive body of knowledge. Resource constraints, including limited funding and access to essential equipment and personnel, also hinder advancements in this field. Moreover, ethical considerations surrounding the use of human-derived cells and tissues, as well as animal models, add another layer of complexity and scrutiny to research endeavors.

Validation and translational challenges further complicate matters. It is critical to ascertain the clinical relevance and applicability of biosensor-enhanced organ-on-a-chip models for predicting patient outcomes and guiding therapeutic strategies. Bridging the gap between bench findings and clinical utility necessitates rigorous validation and meticulous attention to clinical translation. However, despite these challenges, there’s a growing recognition of the immense potential of biosensor-enhanced organ-on-a-chip models in elucidating the dynamics of the GBM tumor microenvironment. Future directions are likely to involve concerted efforts to address technical hurdles, standardize methodologies, secure adequate funding and resources, navigate ethical considerations, and validate models for clinical application. Collaborative endeavors among researchers, clinicians, industry stakeholders, and funding agencies will be pivotal in propelling this field forward and ultimately improving outcomes for patients grappling with GBM.

### 4.3. Future Directions

The future trajectory of biosensor-enhanced organ-on-a-chip models for probing the dynamics of the GBM tumor microenvironment involves an intricate integration of cutting-edge technologies and methodologies. These models are poised to revolutionize our understanding of GBM pathophysiology and enhance therapeutic strategies by providing unprecedented insights into the complex interplay between tumor cells, stromal components, and the immune system.

One key direction for advancing biosensor-enhanced organ-on-a-chip models is the integration of biosensors with multi-omics technologies. This approach allows for comprehensive profiling of GBM tumor microenvironment dynamics at the molecular level. By combining genomics, transcriptomics, proteomics, and metabolomics data with biosensor readouts, researchers can gain a holistic view of the molecular landscape within the GBM microenvironment. This comprehensive understanding can unveil novel biomarkers, signaling pathways, and metabolic alterations driving GBM progression and therapy resistance.

Moreover, the utilization of advanced imaging modalities alongside biosensors enables real-time visualization and quantification of cellular behaviors and interactions within the GBM microenvironment. Techniques such as live-cell imaging, multiphoton microscopy, and single-cell imaging provide invaluable insights into dynamic processes such as cell migration, proliferation, and interaction with the extracellular matrix. Coupled with biosensor data, these imaging modalities offer a multidimensional view of GBM tumor microenvironment dynamics, allowing for precise characterization and analysis.

Personalized medicine approaches are another frontier in the development of biosensor-enhanced organ-on-a-chip models for GBM research. By incorporating patient-derived organoids, or tumoroids, into chip platforms, researchers can replicate the heterogeneity and molecular diversity observed in individual GBM tumors. These patient-specific models enable personalized drug screening and therapeutic optimization, facilitating the identification of tailored treatment strategies based on individual tumor characteristics.

Furthermore, the integration of immune cells into GBM-on-a-chip models is crucial for studying immune cell–GBM interactions and immunotherapy responses. These models provide a platform to investigate the complex interplay between tumor cells and immune cells within the GBM microenvironment, elucidating mechanisms of immune evasion and identifying strategies to enhance anti-tumor immune responses. By incorporating biosensors, researchers can monitor immune cell activation, cytokine secretion, and other immune-related biomarkers in real-time, providing insights into the efficacy of immunotherapies and potential targets for intervention.

Dynamic modulation of the GBM tumor microenvironment is another area of interest for biosensor-enhanced organ-on-a-chip models. By controlling parameters such as oxygen tension, pH, nutrient availability, and mechanical forces, researchers can simulate the dynamic nature of the GBM microenvironment and its impact on tumor growth and therapy response. Integration of biosensors for real-time monitoring of microenvironmental parameters enables precise spatiotemporal manipulation, facilitating the study of microenvironmental dynamics and their influence on GBM progression and therapy outcome.

In summary, the future of biosensor-enhanced organ-on-a-chip models for investigating GBM tumor microenvironment dynamics lies in their ability to integrate advanced technologies, incorporate patient-specific features, and provide insights into the molecular and cellular mechanisms driving GBM pathogenesis and therapy resistance. These models hold immense potential for advancing personalized medicine approaches and developing more effective therapies for GBM patients.

## Figures and Tables

**Figure 1 sensors-24-02865-f001:**
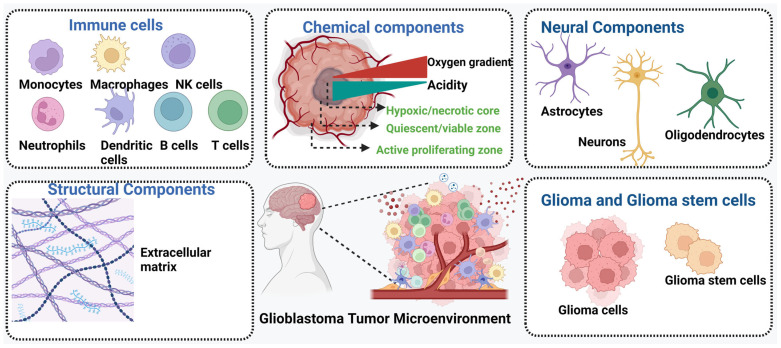
Glioblastoma tumor microenvironment.

**Figure 2 sensors-24-02865-f002:**
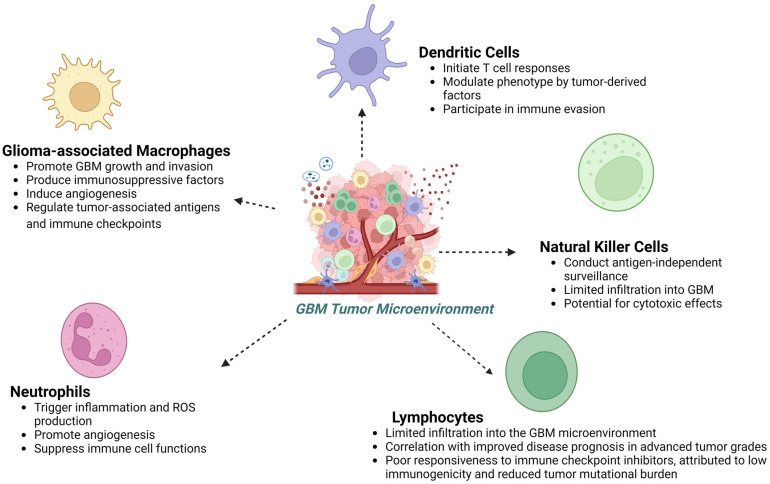
Dynamic interplay of immune cells in glioblastoma: facilitating tumor progression and evading immune surveillance (created with Biorender).

**Figure 3 sensors-24-02865-f003:**
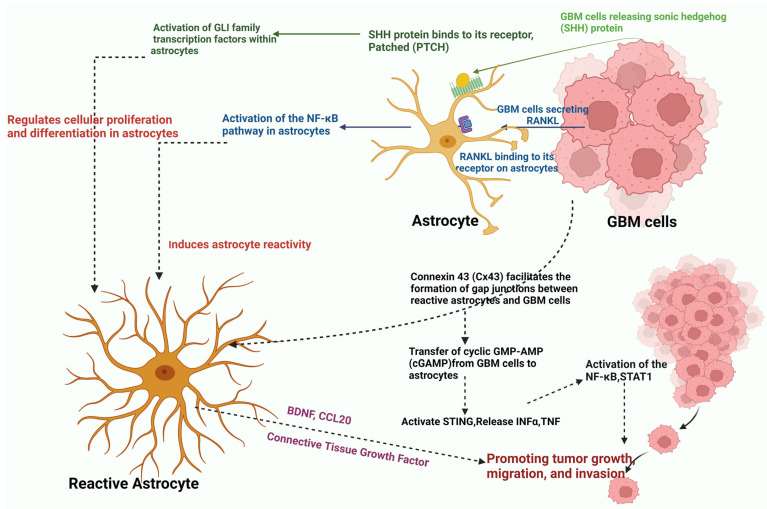
Crosstalk between astrocytes and glioblastoma cells in the tumor microenvironment (created with Biorender).

**Figure 4 sensors-24-02865-f004:**
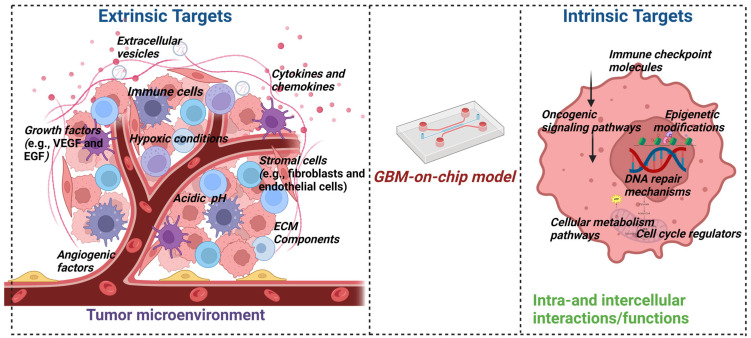
Extrinsic and intrinsic targets within glioblastoma (GBM) biology are vital for tumor-on-chip models. Extrinsic targets involve elements of the tumor microenvironment such as extracellular matrix (ECM) components, immune cells (e.g., macrophages and T cells), stromal cells (e.g., fibroblasts and endothelial cells), cytokines, chemokines, angiogenic factors, hypoxic conditions, acidic pH, extracellular vesicles, growth factors (e.g., VEGF and EGF), and matrix metalloproteinases (MMPs). Intrinsic targets include intra- and intercellular interactions/functions like oncogenic signaling pathways (e.g., EGFR, PI3K/AKT, and MAPK), tumor suppressor genes (e.g., p53 and PTEN), DNA repair mechanisms, epigenetic modifications (e.g., DNA methylation, histone modifications), cell cycle regulators (e.g., cyclins and CDKs), apoptotic pathways, cellular metabolism pathways, cancer stem cells (CSCs), drug efflux pumps (e.g., ABC transporters), and immune checkpoint molecules (e.g., PD-1 and CTLA-4). Incorporating these targets into GBM-on-chip models facilitates comprehensive studies elucidating their roles in tumor progression, drug response, and therapeutic resistance, contributing significantly to our understanding and treatment of GBM (created with Biorender).

**Figure 5 sensors-24-02865-f005:**
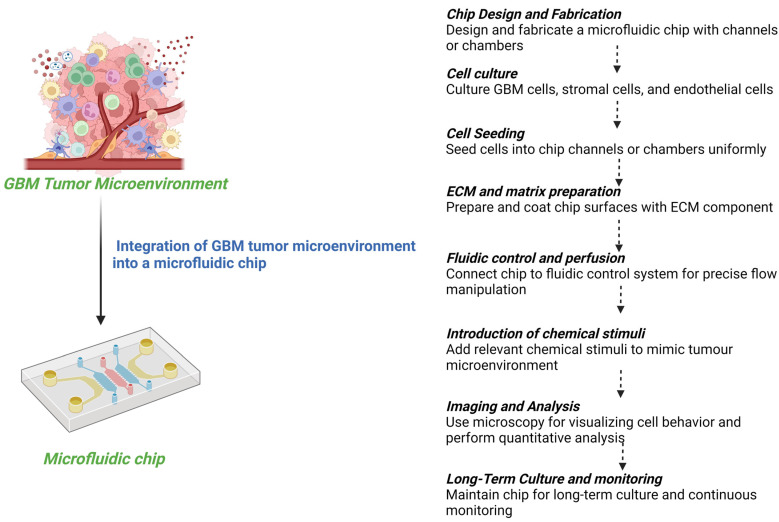
Workflow for creating a glioblastoma tumor microenvironment in microfluidic chips. This figure illustrates the step-by-step process for preparing a glioblastoma tumor microenvironment using traditional microfluidic chips. Starting from chip design and fabrication, the workflow progresses through cell culture, seeding, preparation of extracellular matrix (ECM), fluidic control, introduction of chemical stimuli, imaging, analysis, and long-term culture and monitoring. Each step is crucial for accurately mimicking the complex tumor microenvironment and enables researchers to study tumor biology, drug responses, and therapeutic interventions in a controlled laboratory setting (created with Biorender).

**Figure 6 sensors-24-02865-f006:**
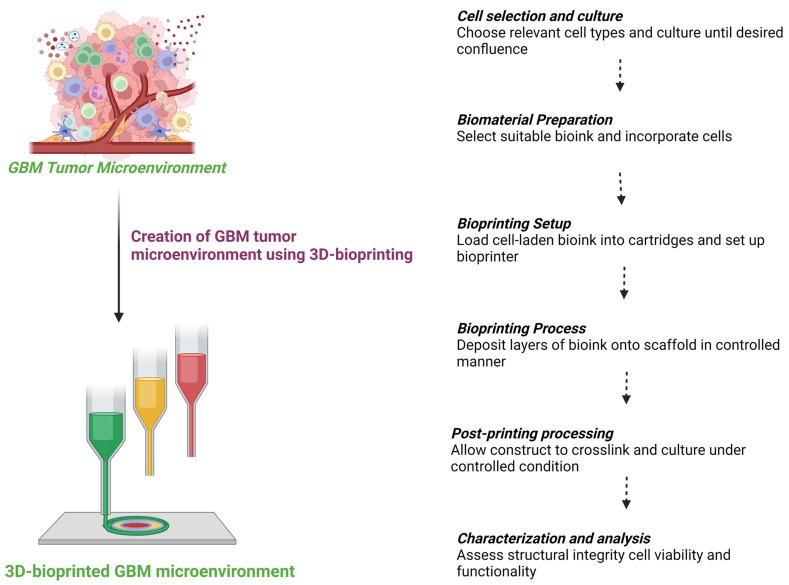
Schematic representation of the 3D bioprinting process for creating the tumor microenvironment of glioblastoma. This illustration depicts the step-by-step procedure involved in 3D bioprinting to simulate the tumor microenvironment typical of glioblastoma. The process encompasses cell selection and culture, biomaterial preparation, bioprinting setup, and post-printing processes. Initially, relevant cell types, including glioblastoma cells, stromal cells, and endothelial cells, are cultured separately to achieve optimal confluence and viability. Subsequently, a suitable bioink, capable of supporting cell viability and mimicking the extracellular matrix composition, is chosen and combined with the cultured cells. The bioprinter is then set up, with the cell-laden bioink loaded into cartridges or syringes. Guided by computer-aided design models, the bioprinter deposits layers of bioink onto a substrate or scaffold in a controlled, layer-by-layer fashion. Printing parameters, such as nozzle size, speed, and pressure, are adjusted to ensure the desired resolution and cell viability. Post-printing processing involves allowing the construct to undergo crosslinking or gelation for stabilization, followed by culture under controlled conditions to promote tissue maturation. The resulting bioprinted model reflects the intricate architecture and cellular composition of the glioblastoma microenvironment, facilitating characterization and analysis for research and therapeutic applications (created with Biorender).

**Figure 7 sensors-24-02865-f007:**
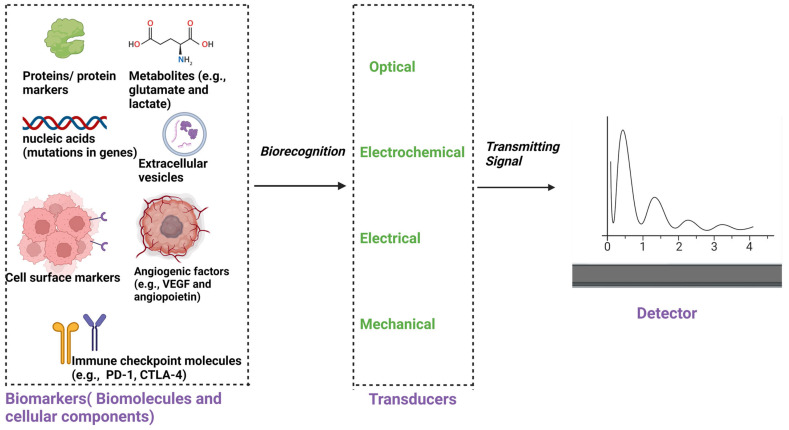
Biosensor detection of GBM biomarkers: sequential process. The process by which biosensors detect biomarkers associated with GBM begins with the recognition of specific biomarkers present within the GBM microenvironment. These biomarkers include proteins, nucleic acids, metabolites, extracellular vesicles, cell surface markers, angiogenic factors, and immune checkpoint molecules. Following biorecognition, a signal is either generated or modulated, which subsequently undergoes transmission and detection by the biosensor system. This systematic sequence of events, involving biorecognition, signal transmission, and detection, facilitates the identification and quantification of GBM biomarkers. Ultimately, this process yields valuable insights into GBM tumor biology, aiding in the assessment of diagnosis, prognosis, and treatment response (created with Biorender).

**Figure 8 sensors-24-02865-f008:**
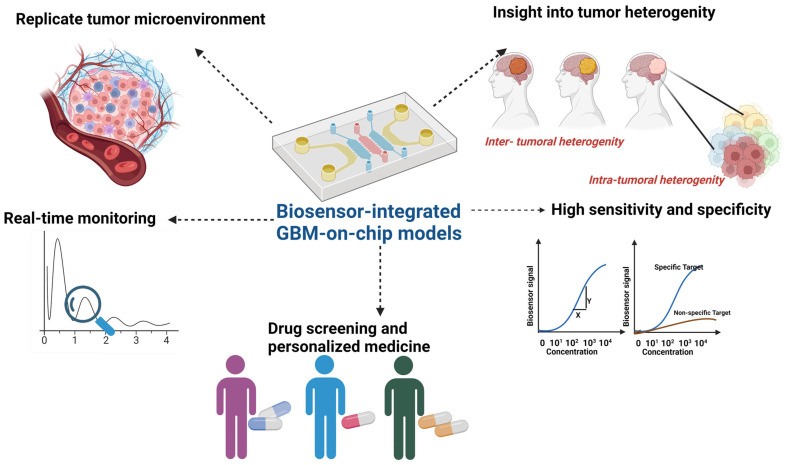
Integrated biosensors in GBM-on-chip models revolutionize glioblastoma research by enabling real-time monitoring of tumor dynamics and treatment responses. This schematic illustrates the multifaceted benefits of biosensor-integrated GBM-on-chip models, including the replication of tumor microenvironmental cues, real-time monitoring of biomarker expression and cellular responses, high sensitivity and specificity in biomarker detection, facilitation of drug screening and personalized medicine approaches, insights into tumor heterogeneity, and reduced reliance on animal models for preclinical studies. These advancements pave the way for improved understanding of GBM biology and the development of more effective therapeutic strategies (created with Biorender).

**Table 1 sensors-24-02865-t001:** Extracellular matrix (ECM) components and their roles in GBM pathogenesis.

ECM Component	Role in GBM Pathogenesis	References
Hyaluronic acid (HA)	Elevated expression levels correlate inversely with glioma patient prognosis.	[45]
Fibronectin	Enhances adherence of glioma stem-like cells, promotes glioma progression and immunosuppression.	[54,55,56,57,58,59,60,61]
Tenascin-C (TN-C)	Facilitates glioma cell invasion and migration through the ECM, modulated by the interleukin-33 (IL-33)-ST2-NFκB pathway.	[42,48,49,50,51,52,53]
Laminins	Promote glioma cell adhesion and invasion, implicated in glioma progression.	[64,65,66,67]
Collagen	Upregulated in GBMs, correlated with poor progression-free survival and overall survival, modulated by various receptors and pathways.	[69,70,71,72,73,74,75]
Matrix metalloproteinases (MMPs)	Degrade ECM proteins, promote cell migration, associated with glioma grading and development.	[76,77,78,79,80,81,82,83]
Other ECM proteins	Drives tumor progression by modulating Notch and NF-κB signaling pathways, correlates with poor patient survival.	[62,63]

**Table 2 sensors-24-02865-t002:** Immune cell types in the GBM microenvironment: functions and targeted therapeutic approaches.

Immune Cell Type	Role in GBM Microenvironment	Mechanisms/Functions	Targeted Therapeutic Approaches
Microglia	Predominant immune cell populations within the glioma microenvironment. Promote glioma proliferation and invasion [86].	Exhibit dynamic phenotypes under different cytokine and chemokine conditions. Produce anti-inflammatory cytokines (IL-4, IL-10, and TGF-β) and tumor-promoting factors (IGF-1, EGF, and PDGF) [88,89,90].	Depletion, repolarization, enhancing phagocytosis, and reducing recruitment strategies explored in preclinical models [101,105,106,107,108,109,110,111].
Glioma-associated macrophages (GAMs)	Constitute up to 30–40% of the bulk tumor mass in GBM. Exert pro-tumorigenic activities by releasing various factors [86,87].	Polarization into M1 or M2 phenotypes. Typically exhibit an immunosuppressive profile. Produce anti-inflammatory cytokines, tumor-promoting factors, angiogenesis mediators, and metabolic disruptors [88,89,90].	Depletion, repolarization, enhancing phagocytosis, and reducing recruitment strategies explored. Therapeutic targets include CSF-1/CSF-1R, the CD47-SIRPα axis, and the chemokine axes (CX3CL1/CX3CR1 and SDF-1/CXCR4) [99,100,101,102,103,110,111,112].
Neutrophils	Subset of myeloid-derived suppressor cells exhibiting pro-tumorigenic properties. Contribute to tumor initiation, proliferation, and dissemination [113,114].	Trigger inflammatory responses through phagocytosis, ROS and nitrogen species production, and the release of cytotoxic granules. Promote angiogenesis and suppress immune responses [113,115,116].	Targeting neutrophil recruitment, blocking immunosuppressive functions, and utilization as drug delivery vehicles are under investigation [113,116,120].
Dendritic cells	Orchestrate the activation and regulation of immune effector cells. Likely participate in recognizing and presenting tumor antigens [121,123].	Modulated by various molecules hindering their activation and driving toward a suppressive phenotype. Induce regulatory T-cell activation and dampen cytotoxic T-cell recruitment [124,125,126,127].	Dendritic cell vaccines (DCVs) aim to enhance tumor antigen presentation. Targeting molecular pathways (e.g., Nrf2 and YTHDF1) to restore DC maturation and effector T-cell activation [129,130].
Natural killer (NK) cells	Exhibit innate anti-tumor activity. Sparse presence in the glioma microenvironment [132].	Functionality modulated by classic and nonclassic HLA-I molecules. Limited by immunosuppressive factors (e.g., TGFβ) in TME [133,134].	CAR NK cell therapy and adoptive NK cell therapy are under investigation in clinical trials. Strategies aim to enhance anti-tumor efficacy by overcoming inhibitory mechanisms [135,136].
Lymphocytes	Scarcity of infiltration in gliomas. CD8+ T cells associated with an improved prognosis [85,132].	Limited responsiveness to immune checkpoint inhibitors. Challenges in antigen presentation and T-cell priming in brain parenchyma [139,140].	Immunotherapeutic strategies (e.g., DC vaccines and STING agonists) aim to enhance glioma immunogenicity and promote cytotoxic T-cell activity. CAR T-cell therapies target specific tumor antigens [141,142].

**Table 3 sensors-24-02865-t003:** Neural component interactions in the glioblastoma microenvironment: roles, mechanisms, and implications.

Neural Component	Role in GBM Microenvironment	Mechanisms/Interactions	References
Astrocytes	Disruption of blood–brain barrier integrity, secretion of RANKL, activation of SHH pathway, production of CTGF, release of CCL20.	Displacement of astrocytic end-feet processes. Activation of NF-κB signaling pathway, regulation of cell proliferation, contribution to glioblastoma growth.	[148,149,150,151,152,153]
Neurons	Regulation of PD-L1 signaling pathway, promotion of glioblastoma cell survival and proliferation.	Activation of Ras protein via PD-L1 binding; secretion of EGFR, interferons, toll-like receptor ligands.	[157,158]
Oligodendrocytes	Upregulation of GBM invasiveness through the angiopoietin-2 signaling pathway, facilitation of glioblastoma migration and invasion.	Binding of angiopoietin-2 to endothelial cell receptors. Enhancement of GBM migratory capabilities.	[159,160]
Paracrine interactions	Exchange of BDNF and NLGN3-mediated signals, regulation of BDNF expression by miRNAs, promotion of glioblastoma growth and survival.	Conversion of pro-BDNF to mature BDNF, upregulation of synapse-related gene expression in GBM cells. Activation of the PI3K-mTOR pathway.	[161,162,163,164,165]

**Table 4 sensors-24-02865-t004:** Interplay of hypoxia and acidosis in glioblastoma multiforme (GBM): key aspects and implications.

Aspect	Hypoxia	Acidosis
Definition	Condition characterized by low oxygen levels within the tumor microenvironment [169].	Condition marked by an acidic pH environment surrounding cancer cells [191,192,193,194].
Etiology	Result of inadequate oxygen delivery due to abnormal tumor vasculature, high metabolic demand, and limited diffusion [169].	Driven by metabolic reprogramming, notably the Warburg effect, leading to increased lactic acid production and accumulation of H^+^ ions [191,192,193,194].
Cellular effects	Influence angiogenesis, invasion, drug resistance, immune evasion, and cellular adaptation [169].	Induces stem-like properties, enhances invasiveness, alters cellular metabolism, affects tumor microenvironment interactions [195].
Molecular mechanisms	Upregulation of HIF-mediated pathways, modulation of gene expression, activation of survival pathways, promotion of invasion [169].	Activation of HIF-mediated pathways, modulation of gene expression, upregulation of cytoprotective proteins, promotion of cellular survival and aggressiveness [195,196].
Therapeutic implications	Target HIF signaling pathways, hypoxia-targeting agents in clinical trials [169].	Utilization of carbonic anhydrase inhibitors like acetazolamide alongside standard-of-care therapy, under clinical investigation for efficacy in GBM treatment [198,199].
Clinical trials	Ongoing trials investigating hypoxia-targeting agents such as MBM-02 (Tempol) and belzutifan (NCT04874506 and NCT02974738).	Phase I trial evaluating acetazolamide in adjuvant temozolomide treatment for newly diagnosed MGMT-methylated malignant glioma (ACZ in GBM, NCT02026024) [199].
Promising outcomes	Improved treatment outcomes, potential disruption of hypoxia signaling, enhanced efficacy of existing therapies.	Well-tolerated addition of ACZ to adjuvant temozolomide treatment, potential improvement in progression-free survival and overall survival in GBM patients [199].

**Table 5 sensors-24-02865-t005:** A comparative analysis of the basic characteristics, advantages, and disadvantages of various fabrication techniques for glioblastoma-on-chip models.

Biosensors	Methods	Advantages	Disadvantages	Applications
Soft lithography	Microcontact printing (µCP)	Enables precise patterning of extracellular matrix (ECM) proteins and cell-adhesive molecules. Mimics spatial cues inherent in native tissues, promoting cell behavior resembling in vivo conditions. Cost-effective and straightforward process, suitable for high-throughput applications.	Limited to the patterning of specific biomolecules, lacking the ability to replicate complex tissue structures.	Facilitating controlled cell adhesion and organization within glioblastoma-on-chip models.
	Molding-based soft lithography	Allows for the fabrication of microfluidic channels and chambers with precision and reproducibility. Offers control over channel dimensions and geometries, facilitating customization for specific experimental needs.	Compatible with various elastomeric materials, such as polydimethylsiloxane (PDMS).	Emulating vascular networks and tissue compartments within glioblastoma-on-chip models.
	Micromolding in capillaries (MIMIC)	Provides a straightforward approach for fabricating microfluidic channels with defined dimensions and geometries. Offers the ability to create gradients of nutrients or signaling molecules within microchannels.	Limited to generating simple channel structures, necessitating precise control over the filling process.	Cultivating glioblastoma cells within controlled microenvironments, allowing for precise manipulation of signaling gradients.
	Embossing and replica molding	Facilitates the creation of microscale features and topographical cues within soft elastomeric materials. Enables the replication of patterns onto soft elastomeric substrates, ensuring consistency and reproducibility. Offers flexibility in design and customization, allowing for the tailoring of chip structures to specific experimental requirements.	Requires meticulous control over the embossing process and may involve multiple steps for replication.	Constructing barrier structures and intricate microenvironments within glioblastoma-on-chip models.
3D bioprinting	Extrusion-based bioprinting	Versatile, allows the use of various bioinks, including hydrogels loaded with cells and growth factors.	Limited resolution compared to other 3D bioprinting techniques, may result in cell damage during printing.	Fabrication of 3D GBM models by printing layers of bioinks containing GBM cells and supportive matrices.
	Inkjet-based bioprinting	High precision and scalability.	Limited to low-viscosity bioinks, may cause cell damage, limited to small-scale printing.	Creation of GBM models by printing bioinks containing GBM cells, ECM components, and other relevant cell types.
	Laser-based bioprinting	High resolution and minimal damage to biomaterials, capable of printing complex structures with high fidelity to native tissue architecture.	Complex setup and high cost, limited to certain biomaterials, limited to small-scale printing.	Generating GBM models with precise spatial arrangements to mimic the tumor microenvironment using cell deposition, fabrication of tissue scaffolds for organ-on-a-chip and drug screening applications.

**Table 6 sensors-24-02865-t006:** Comparison of platforms for creating glioblastoma (GBM)-on-chip models: characteristics, advantages, and disadvantages.

Platform	Characteristics	Advantages	Disadvantages
Microfluidic cell culture platforms	Provide precise control over cell culture conditions.	Mimic in vivo tumor microenvironment. Study of cell response to various microenvironmental factors.	Limited scalability.
3D microfluidic culture systems	Enable the growth of GBM cells in a 3D environment.	Recapitulate tumor architecture. Drug screening and testing in a more physiologically relevant model.	Complex fabrication processes. Limited compatibility with certain assays.
Hydrogel-based microfluidic platforms	Mimic extracellular matrix of tumors.	Provide a supportive environment for GBM cells.	Hydrogel properties may affect cell behavior.
organ-on-chip systems	Mimic physiological functions of the brain.	Allow study of tumor-brain interactions. Require integration of multiple cell types and functions.	Integration complexity may affect device performance.
Microscale tissue engineering platforms	Allow for the creation of microscale tissue constructs.	Investigating cell–cell and cell–matrix interactions. Drug screening in a controlled microenvironment.	Require optimization for mimicking tumor microenvironment. Limited scalability for high-throughput applications
Droplet-based microfluidic platforms	Enable high-throughput screening of compounds, biomolecules, and cells by encapsulating them in individual droplets, allowing for parallel analysis.	Allow for precise control over droplet size, composition, and content. Facilitate single-cell encapsulation and analysis. Support multiplexed assays. Reduce sample and reagent consumption due to small volumes of droplets, making it cost-effective and suitable for rare or expensive materials. Provide accurate and reproducible experimental results. Enable studies of cellular heterogeneity and rare cell populations within GBM samples.	Complexity of device fabrication. Limited compatibility with high-viscosity samples. Risk of cross-contamination between droplets. Limited compatibility with certain assays. Challenges in scaling up to larger throughput applications.
3D bioprinting platforms	Utilize computer-aided design (CAD) software to create digital models of tissue constructs, which are then translated into printable instructions. Employ various printing techniques, such as extrusion-based, inkjet-based, and laser-based bioprinting, to deposit bioinks with cellular components in precise spatial arrangements through layer-by-layer deposition, allowing for the creation of complex tissue structures. Utilize bioinks containing cells and biomaterials to generate tissue constructs with desired properties. Can incorporate multiple cell types and growth factors into printed structures.	Enable customization of scaffold properties, including porosity, mechanical strength, and degradation rate, to mimic the GBM microenvironment. Allow for the engineering of complex tissue models for drug testing and personalized medicine applications. Provides control over tissue architecture, facilitating the study of cell–cell and cell–matrix interactions.	Limited resolution and printing speed, leading to challenges in accurately reproducing complex tissue structures and vasculature. Difficulties in bioprinting vasculature, limiting the ability to mimic the vascularized GBM microenvironment. Require optimization of bioink formulations and printing parameters for specific applications.

**Table 7 sensors-24-02865-t007:** Comparative analysis of biosensing detection methods for integration with organ-on-chip models: principles, target analytes, advantages, and limitations.

Biosensing Detection Method	Principle	Targets	Advantages	Disadvantages
Electrochemical biosensors [294,295,296,297]	Detection based on electrical signal changes includes alterations in current, voltage, impedance, or capacitance, depending on the nature of the electrochemical transduction mechanism resulting from analyte binding to bioreceptors immobilized on electrodes.	Enzymes, DNA, antibodies, whole cells.	Wide dynamic linear range. Low detection limits. Real-time monitoring. High sensitivity and specificity. Diverse receptor immobilization. Integration with microfluidic systems.	Requires careful surface modification for receptor immobilization. Susceptible to interference from non-specific binding. Electrode fouling may occur over time.
Optical biosensors [293,307]	Detection relies on optical signals generated by interactions between target analytes and recognition elements. These biosensors can be categorized into label-free and label-based models. Label-free models rely on inherent material properties for detection, such as light scattering and SPR. In contrast, label-based models utilize labeling reagents to produce detectable optical signals, such as fluorescence.	Enzymes, antibodies, antigens, nucleic acids, whole cells, tissues.	High sensitivity and specificity. Low limits of detection. Minimal background noise. Compact size. Multiplexing capability. Non-destructive. Wide dynamic range. Compatibility with microfluidic systems. Stability.	Label-based methods require labeling reagents. May be influenced by sample turbidity or autofluorescence. Optical alignment may be challenging.
Electrical biosensors [315]	Detection based on changes in the electrical properties of sensing elements upon exposure to analytes.	Voltage changes, electrical impedance.	Non-invasive and rapid assessment of barrier integrity. Real-time monitoring of cellular functions. Direct investigation of cellular electrical properties. Long-term monitoring within multi-organ systems. Versatile design options for integrating various modalities.	Limited to the electrical properties of analytes. Requires expertise in microelectronics for integration. Sensitivity may be affected by environmental conditions.
Mechanical biosensors [292]	Detection based on mechanical changes induced by analyte binding or cellular activities.	Mechanical stress.	High sensitivity to mechanical stimuli. Potential for label-free detection. Compatible with microfluidic systems. Versatility. Non-invasiveness.	Complex fabrication processes. Limited to specific mechanical properties of analytes. Signal interpretation may be challenging.

**Table 8 sensors-24-02865-t008:** Overview of biosensors used in various research and their functions in organ-on-chip (OOC) systems.

Biosensor		Analyte	Main Function	Type	Reference
Electrochemical		Glucose, lactate	Continuous monitoring of glucose consumption and lactate secretion	Colon cancer microtissues	[298]
		Oxygen, lactate, and glucose	Monitoring of energy metabolites and culture conditions in 3D cell cultures	Matrix-based, heterogeneous 3D cultures	[299]
		Creatine kinase (CK)-MB biomarker	Monitoring biomarkers secreted from damaged cardiac tissue	Heart-on-a-chip platform	[302]
		Heat shock protein 70 (HSP-70) biomarker	Immobilizing antiHSP70 antibodies on SPE	OOC platform	[303]
		Interleukin 6 (IL-6) and tumor necrosis factor α (TNF-α)	Detection of IL-6 and TNF-α following electrical stimulation of skeletal muscle tissue	Integrated platform for muscle cell stimulation	[304]
		Biochemical signals (nitric oxide, reactive oxygen species)	Monitoring biochemical signals during vascular mechanotransduction in real-time	Vascular chip model	[305]
		Lactate dehydrogenase (LDH)	Measurement of LDH concentration, indicating cytotoxicity and cell damage		[306]
Optical		Tissue responses	Non-invasive monitoring of tissue-level remodeling	Synovium-on-a-chip	[307]
		Insulin secretion in response to glucose stimulation	Real-time detection of insulin secretion	Islets-on-a-chip	[311]
		Oxygen consumption	Monitoring O_2_ consumption under electrical stimuli	SpheroFlow HoC system	[312]
		Oxygen consumption	Monitoring O_2_ consumption to evaluate cardiac metabolism	Co-culture of mesenchymal stem cells and cardiac tissues	[314]
Electrical		TEER	Measuring TEER and cell layer capacitance	Human lung airway chip; human gut chip	[316]
		TEER	Real-time measurement for BBB-on-a-chip	hiPSC-derived blood–brain barrier chips	[317]
		TEER	Monitoring reliably in the presence of microbubbles	Intestinal barrier-on-chip	[318]
		Resistivity	Real-time monitoring of resistivity changes	A breathing lung-on-chip	[319]
		Electrical signals	Measuring the electrical activity from neurons	TFT systems applicable to neuron culture	[320]
		Contractile force, electrical conductivity	Monitoring the electrical activity of cardiomyocytes	Multi-organ human-on-a-chip system	[321]
Multiple biosensors	Electrical/optical	Oxygen, TEER	Monitoring oxygen levels and TEER	HuMiX OOC systems	[322]
	Electrical/electrochemical	Oxygen, ROS	Monitoring oxygen levels and oxidative stress	Gut-on-a-chip models	[324]
	Electrical/electrochemical	TEER, glucose levels, ROS	Assessment of TEER, glucose levels, and ROS	Distal tubule-on-a-chip models	[326]
	Electrochemical/electrical	pH, cell proliferation	Monitoring pH and cell proliferation	Colorectal Perfused Tumoroid Platforms	[327]
	Optical/electrochemical	pH, TEER, biomarkers	Assessment of pH, TEER, and biomarkers	Lung cancer on a chip model	[328]
	Optical/electrochemical	Various biomarkers	Comprehensive toxicity assessment	Cancer, heart, and liver models	[329]
Other types of biosensors	Electrochemical stripping analysis sensors	TEER, ion levels	Real-time measurements of TEER and ion levels	Cell culture platform	[332]
	Micro-pump sensor array (MPSA) system	Various tissue parameters	Support various tissue models, including liver, vascular, gastrointestinal, and kidney chips	Various tissue models	[333]
	Phosphorescent microprobes and electrochemical sensors	Glucose, lactate, oxygen	Real-time monitoring of mitochondrial respiration and metabolic shifts from oxidative phosphorylation to anaerobic glycolysis	Liver-on-chip	[334]
	Clark-type oxygen microsensors	Oxygen	Long-term oxygen monitoring, implementation of the Ross principle	Various	[335]
	Electrochemically dissolved oxygen sensors	Oxygen	Local online monitoring of oxygen concentrations	Liver-on-chip	[336]
	Microfluidic pH sensor-embedded chip	pH levels	Real-time pH measurement using micro-cantilever	Microfluidic chip	[337]
	Microfluidic system with light addressable sensor (LAPS)	Extracellular acidification	Real-time detection of acidification	Modular microfluidic system	[338]
	Smart temperature sensors	Temperature	Real-time temperature monitoring	Silicon-based organ-on-a-chip device	[339]
	Resistive temperature sensors (RTDs)	Temperature	Accurate temperature monitoring and regulation	Organ chips	[340]
	Osmotic hydration sensor	Osmotic pressure	Monitor osmotic pressure	Microfluidic systems	[341]

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
