# Peer review of "Biosensor-Enhanced Organ-on-a-Chip Models for Investigating Glioblastoma Tumor Microenvironment Dynamics"

_sensors, 2024, doi:10.3390/s24092865_

Round 1

Reviewer 1 Report

Comments and Suggestions for Authors

This paper is very successful and public able, but some problems with figures. For example i did not understand the name of figure 1. Drawings i think should be performed with higher resolution and more cleared details in figures 2 and 3 about chips. On figure 2 the letters "GBM-on-chip model" is coincident with the left part. The same problem on figure 5 with medium part (Green letters "Electrochenical). 

Author Response

Dear Reviewer,

Thanks for your review and feedback.

Name of Figure 1 is included.

The issue with the alignment of the "GBM-on-chip model" in figure 2 and 3, coinciding with the left part have bee corrected. Similarly, in Figure 5, there was an alignment problem with the medium part, where the green letters "Electrochemical" were affected. The green letters "Electrochemical"  has been corrected  with the newly constructed figure 5.

Reviewer 2 Report

Comments and Suggestions for Authors

The manuscript by Thenuwara and Tian addresses a topic that will attract the attention of a diverse audience as it provides information on translational research into the most aggressive brain tumor using nanobiology technology. While the article is quite interesting and is incredibly well written, it is very long and needs to be edited to make the reading more enjoyable and more focused. The introduction seems like a big advertisement for the use of the chip and provides little relevant information to the reader. The focus of the article is not the hallmarks of glioblastoma or its characteristics that make treatment difficult or even impossible. The first half of the article (page 1 to page 26) details several aspects that make glioblastoma an extremely aggressive and incurable tumor. Topics 1 and 2 must be included in the introduction, which should not exceed two pages. The real manuscript starts on topic 3 (page 26). Additionally, all figures require improved resolution for clarity.

Author Response

Dear Reviewer,

Thanks for reviewing and feedback. 

The topic 1 and topic 2 have shorted.   8 tables have been introduced to summary  the significant contribution in the field.

The quality of the five figures have been improved with high resolution.

Furong

Reviewer 3 Report

Comments and Suggestions for Authors

This manuscript investigated recent research activities on Organ-on-a-Chip to explore the microenvironmental dynamics of primary brain tumors, particularly gliomas. The authors examined the microenvironment of glioma tumors and confirmed the potential applicability of Organ-on-a-Chip to mimic this environment. Therefore, it is suitable for publication in the "Sensor" journal as it contains interesting topics and results. However, some revisions are necessary before publication. Recommendations for revisions are as follows:

Major Revisions:

  1. 1. In sections "2.2. Immune Composition in the Microenvironment of Schwannomas," "2.3. Neurological Components in the Microenvironment of Gliomas," and "2.4. Chemical Components in the Microenvironment of GBMs," include figures that can more easily explain the content. Enhance clarity.

  2. 2. This manuscript describes various research activities related to the biosensor GBM tumor-on-a-chip model. However, it seems necessary to add a summary table or brief overview of these research activities.

Minor Revisions:

  1. 1. In the abstract, use the abbreviation "OOC" for Organ-on-a-Chip consistently throughout the text, instead of using "OoC."

Author Response

Dear Reviewer,

Thanks for reviewing and feedback.

1. In sections "2.2. Immune Composition in the Microenvironment of Schwannomas," "2.3. Neurological Components in the Microenvironment of Gliomas," and "2.4. Chemical Components in the Microenvironment of GBMs," include figures that can more easily explain the content. Enhance clarity.

Reply: The new table 2, table 3 and table 4  have been included to clarify the content of section 2.3, 2,3 and 2.4 , respectively. 

2. This manuscript describes various research activities related to the biosensor GBM tumor-on-a-chip model. However, it seems necessary to add a summary table or brief overview of these research activities.

Reply:  Table 8 has been included to describe various research activities related to the biosensor GBM tumor-on-a-chip model.

Minor Revisions:

 In the abstract, use the abbreviation "OOC" for Organ-on-a-Chip consistently throughout the text, instead of using "OoC."

Reply:

All of 'OoC' have been replace by 'OOC' through the manuscript.

Reviewer 4 Report

Comments and Suggestions for Authors

Organ-on-a-chip is a highly cutting-edge research field, and chips integrated with biosensors possess more prominent advantages, making them significant for physiological and disease research. This article provides a comprehensive overview with certain reference value. However, there are areas that require further optimization, such as:

  1. It would be best to include a list of abbreviations, as the length of the article is quite long and there are numerous abbreviations used.
  2. Some sections, such as 2.1, are too long and could be divided into shorter subsections for better readability.
  3. There are a few formatting errors, such as missing a space on line 293, an extra space on line 305, and indentation issues on line 538.
  4. A comparative analysis of the basic characteristics, advantages, and disadvantages of various GBM-on-a-Chip construction methods should be presented in a table.
  5. Similarly, a table comparing the basic characteristics, advantages, and disadvantages of various biosensing detection methods (based on different principles and targets) should also be included.

Author Response

Dear Reviewer,

Thanks for reviewing and feedback.

It would be best to include a list of abbreviations, as the length of the article is quite long and there are numerous abbreviations used.

Reply: The text of manuscript has been shorted and a list of abbreviations has been include at end of text of manuscript.

2. Some sections, such as 2.1, are too long and could be divided into shorter subsections for better readability.

Reply: section 2.1 has been shorted from 5 pages to 2 pages and table 1 has been included to clarify the contribution in the field.

3. There are a few formatting errors, such as missing a space on line 293, an extra space on line 305, and indentation issues on line 538.

Reply: The line spaces have been justified. 

4.A comparative analysis of the basic characteristics, advantages, and disadvantages of various GBM-on-a-Chip construction methods should be presented in a table.

Reply: A comparative analysis of the basic characteristics, advantages, and disadvantages of various GBM-on-a-Chip construction methods has been include in figure 7. 

5. Similarly, a table comparing the basic characteristics, advantages, and disadvantages of various biosensing detection methods (based on different principles and targets) should also be included.

Reply: Table 5 has been included basic on characteristics, advantages, and disadvantages of various GBM-on-a-Chip construction methods on different principle.

Table  6 is summarised  Various Fabrication Techniques for Glioblastoma-on-Chip Models to summary the different platforms  in section 3.2. Bioprinting GBM-on-a-Chip.